# Controlled Collaboration Geometry for Personalized Federated Learning

Hongbo Yin [1]  Jichun Wu [1]  Zhou Yang [2]  Chi Jiang [1]  Yin Zhang [1]  Yan Zhang [1]

## Abstract

In personalized federated learning (PFL), collaboration graphs specify model aggregation among clients. However, without constraints on the collaboration geometry, training can drift into two degenerate regimes: global consensus or spontaneous clustering. This paper provides a unified dynamical analysis: under the same budget of representative models, collaborative PFL is more expressive and achieves higher-order approximation accuracy than clustered PFL. An upper bound on disagreement further reveals two degeneration mechanisms—overly strong collaboration drives consensus (reducing to standard federated learning), while similarity-driven weight updates make the graph nearly reducible and induce self-clustering (collapsing to clustered PFL). Motivated by these findings, we propose pFedCCG. pFedCCG preserves the expressivity advantage via controlled collaboration geometry (CCG): it builds a static similarity-based collaboration template decoupled from training, optimizes a Markovian collaboration matrix with a prescribed stationary distribution via reversible parameterization and Euclidean projection, and schedules collaboration strength to avoid self-clustering. Experiments across diverse heterogeneity settings show consistent personalization gains and markedly reduced collapse and self-clustering. Code is available at https://github.com/YinHonb/pFedCCG.

## 1. Introduction

Federated learning (FL) (McMahan et al., 2017) trains models collaboratively without sharing raw data, but heterogeneous client distributions often make a single global model

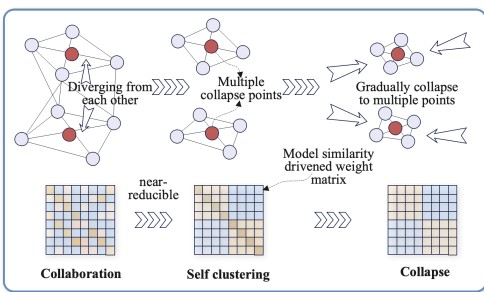

*Figure 1.* Collaboration based on model similarity to induce collapse into clustered PFL.

suboptimal (Zhu et al., 2021; Arbaoui et al., 2024; Karimireddy et al., 2020). Personalized federated learning (PFL) remedies this by learning client-specific models while retaining cross-client collaboration (Tan et al., 2022).

A common perspective is that PFL algorithms are largely distinguished by the aggregation geometry that governs inter-client collaboration. Regularization or decomposition based methods personalize on top of an FL-style backbone (Li et al., 2021; T Dinh et al., 2020; Fallah et al., 2020; Collins et al., 2021), while another line explicitly designs collaboration structures. At a coarse level, clustering-based PFL partitions clients into $K$ groups and shares one prototype per group (CFL-style) (Sattler et al., 2020; Licciardi et al., 2025; Huang et al., 2021). At a finer level, graph-based PFL learns a client-specific aggregation row and performs weighted mixing on a collaboration graph (Ye et al., 2023; Zhang et al., 2024; Chen et al., 2024; Liao et al., 2023). Thus, the collaboration kernel effectively defines the hypothesis class each client can reach via mixing.

Conceptually, clustering can be viewed as a coarse special case of collaboration in which each client is tied to a single prototype among $K$ anchors, whereas collaboration supports fine-grained, client-specific convex mixing over anchors, yielding a more flexible aggregation mechanism(Smith et al., 2017). However, in practice, the expressivity advantage can be undermined by the coupled evolution of optimization and aggregation, since the collaboration kernel directly shapes how inter-client differences are propagated across rounds.

In this paper, we identify two degeneration modes. First,

---

[1]School of Information and Communication Engineering, University of Electronic Science and Technology of China, China [2]School of Computer Science and Technology, Xi'an Jiaotong University, China. Correspondence to: Yin Zhang <yin.zhang.cn@ieee.org>.

*Proceedings of the 43 rd International Conference on Machine Learning*, Seoul, South Korea. PMLR 306, 2026. Copyright 2026 by the author(s).

when the kernel is strongly contractive on disagreement directions, aggregation repeatedly attenuates inter-client diversity; together with the contractive effect of local training, the dynamics drift toward near-consensus and **degenerates to standard FL**(Olfati-Saber et al., 2007). Second, in similarity-driven designs that update weights from current model proximity (Ye et al., 2023; Zhang et al., 2024; Chen et al., 2024; Liao et al., 2023), the feedback loop amplifies existing closeness: similar clients increasingly mix with each other, while cross-group interactions gradually vanish. This progressively weakens global connectivity and yields an almost block-separated collaboration graph(Sharma et al., 2025), resulting in spontaneous self-clustering and a multi-point collapse that **degenerates to $K$-clustered FL**, as shown in Fig. 1.

Consequently, without explicit control of the collaboration geometry, collaborative PFL may either collapse to a single consensus model or fragment into multiple block-wise consensus points, in both cases erasing its expressivity advantage.

Motivated by these insights, we show that collaboration is strictly more expressive than clustering under the same $K$-anchor budget(Yarotsky, 2017), and further establish that this expressivity can be lost through two degeneration modes, namely **degenerating to standard FL** or **degenerating to $K$-clustered FL**. Guided by these findings, we propose **pFedCCG**, a controlled collaboration geometry framework that explicitly regulates the collaboration kernel across rounds to preserve expressivity while avoiding degeneration. The contributions of this paper are summarized as follows:

- **Collaboration strictly dominates clustering.** We establish a hypothesis-class comparison showing that collaborative PFL strictly contains clustered PFL under the same $K$-anchor budget, and achieves a strictly faster approximation rate under intrinsic-dimensional continuous heterogeneity.

- **Consensus-collapse theory.** We analyze the coupled training–aggregation dynamics and derive a disagreement upper bound showing that overly contractive collaboration kernels suppress inter-client diversity and drive the system toward near-consensus, thereby degenerating to standard FL.

- **Multi-point collapse theory.** We further show that similarity-driven feedback can make the collaboration graph nearly reducible, induce self-clustering, and lead to block-wise consensus, thereby degenerating to a $K$-prototype clustered FL solution.

- **Controlled collaboration geometry.** Guided by the above theory, we propose pFedCCG, which designs

a controlled collaboration geometry to stabilize aggregation over training. pFedCCG breaks similarity feedback with a fixed affinity template, and produces valid time-varying kernels through a simple projection scheme, effectively preventing collapse in collaborative PFL.

## 2. Related work

The previous PFL methods predominantly include regularization (pFedME (T Dinh et al., 2020), Ditto (Li et al., 2021)), FedAMP (Huang et al., 2021)), model splitting (FedRep (Collins et al., 2021), FedRoD (Chen & Chao, 2021)), meta learning (perFedAvg (Fallah et al., 2020)), etc. However, these methods lack fine-grained collaboration at the client level, leading to drift in personalized models.

### 2.1. Cluster-based PFL

Clustered personalized federated learning (CFL) (Sattler et al., 2020) partitions clients into clusters and performs aggregation within each cluster to mitigate non-IID effects. Early methods used parameter or gradient similarity for hard clustering and trained one model per cluster, but are sensitive to similarity thresholds and may yield unstable assignments and privacy risks. Later works improve intra-cluster collaboration and knowledge sharing: FedCollab (Bao et al., 2023) adjusts intra-cluster weights; FedGWC (Licciardi et al., 2025) separates global and local weights for cross-cluster sharing; CFLGP (Kim et al., 2024) stabilizes clustering via refined gradient similarity; pFedCS (Wu et al., 2025) fine-tunes locally after clustering; FedSPD (Lin et al., 2025) adopts decentralized soft clustering; LCFed (Zhang et al., 2025) uses model partitioning and low-rank similarity for updates; and IFCA (Ghosh et al., 2022) alternates assignment estimation and model optimization. Nonetheless, most CFL methods rely on few prototypes, resulting in coarse granularity and limited modeling of fine-grained heterogeneity.

### 2.2. Collaborative PFL

In recent years, graph structures have been widely adopted in personalized federated learning (PFL) to construct client collaboration topologies and guide aggregation by dynamically adjusting graph connections. Early methods such as SFL (Chen et al., 2022), BiG-Fed (Xing et al., 2022), GCRO (Yu et al., 2024), and pFedGAT (Zhou et al., 2025) learn collaboration via graph modeling, but often rely on prior graphs or external knowledge, which can limit generalization in real deployments. Later works build graphs from model similarity or learnable mechanisms. For example, pFedGraph (Ye et al., 2023) and GPFedRec (Zhang et al., 2024) infer relationships and weights using model or embedding similarity; FedAGHN (Song et al., 2025) and FedAMP (Huang

et al., 2021) use attention to capture fine-grained collaboration; and GraphRx (Wang et al., 2025) applies message passing to integrate intra and inter cluster information. However, in highly heterogeneous or large-scale settings, these methods may still exhibit self-clustering, where the graph splits into internally mixed groups with weak cross-group interaction, reducing effective collaboration and limiting personalization.

# 3. Theoretical analysis

In this section, we first compare clustering and collaboration from an expressivity viewpoint and show that collaboration strictly dominates clustering under the same $K$-anchor budget. Then, we analyze the coupled training to aggregation dynamics and characterize a collaboration-induced collapse to standard FL. Finally, we show that similarity-driven feedback can push the dynamics to the opposite extreme—self-clustering into multiple block-wise consensus points—so that collaboration degenerates to $K$-clustered FL.

## 3.1. Problem Setup and Notation

Consider a PFL setting with $N$ clients. At the beginning of communication round $t$, the model parameters of client $i$ are denoted by $\boldsymbol{\theta}_i^t \in \mathbb{R}^d, i = 1, \ldots, N$. For the convenience of analysis, stacking all client models into $\boldsymbol{S}^t \triangleq [(\boldsymbol{\theta}_1^t)^\top \ldots (\boldsymbol{\theta}_N^t)^\top]^\top \in \mathbb{R}^{Nd}$. The client $i$ performs local training on its local objective $L_i$ and obtains the updated model $\widetilde{\boldsymbol{\theta}}_i^t$. After local updates, each client $i$ aggregates (a weighted combination of) other clients' updated models through a collaboration matrix $\boldsymbol{W}^t \in \mathbb{R}^{N \times N}$ to form its next-round initialization: $\boldsymbol{\theta}_i^{t+1} = \sum_{j=1}^N W_{ij}^t \widetilde{\boldsymbol{\theta}}_j^t$. The aggregation process can be represented in the form of Kronecker-product:

$$\boldsymbol{S}^{t+1} = (\boldsymbol{W}^t \otimes \boldsymbol{I}_d) \widetilde{\boldsymbol{S}}^t. \tag{1}$$

Let $\boldsymbol{\theta}_i^\star \in \arg\min_{\boldsymbol{\theta} \in \mathbb{R}^d} L_i(\boldsymbol{\theta})$ denote a optimal model for client $i$. We measure personalization error by the $q$-weighted excess risk $\mathcal{R}(\{\boldsymbol{\theta}_i\}) \triangleq \sum_{i=1}^N q_i \left( L_i(\boldsymbol{\theta}_i) - L_i(\boldsymbol{\theta}_i^\star) \right)$, where $q_i > 0, \sum_{i=1}^N q_i = 1$ is the proportion of dataset size.

## 3.2. Why Collaboration rather than clustering?

Clustering partitions clients into $K$ groups and represents each group by a single prototype. Intuitively, Collaboration is more expressive in principle since each client may mix models with client-specific weights rather than selecting a single clustering prototype.

More importantly, even under the same budget of only $K$ representative models (anchors), collaboration admits a strictly richer approximation family than clustering, and

thus can achieve a higher-order approximation of a continuous set of heterogeneous optima. In particular, clustering (piecewise-constant) incurs an irreducible quantization bias, whereas collaboration (piecewise-linear in latent space) achieves a strictly faster approximation rate. The next theorem formalizes this gap under a common $K$-anchor budget.

**Theorem 3.1** (Expressivity gap between collaboration and clustering). *Let $\mathcal{R}_{\mathrm{cl}}^\star(K)$ and $\mathcal{R}_{\mathrm{col}}^\star(K)$ denote the best achievable risks within the $K$-clustered class and the $K$-collaborative class (Appendix A). Under mild regularity assumptions (local quadratic growth and intrinsic $r_{\mathrm{int}}$-dimensional continuous heterogeneity):*

1. ***Dominance:*** $\mathcal{R}_{\mathrm{col}}^\star(K) \leq \mathcal{R}_{\mathrm{cl}}^\star(K)$ *for all $K \geq 1$.*

2. ***Rate gap:*** *there exist constants $c, C > 0$ independent of $K$ such that*

$$\mathcal{R}_{\mathrm{cl}}^\star(K) \geq c\, K^{-2/r_{\mathrm{int}}}, \quad \mathcal{R}_{\mathrm{col}}^\star(K) \leq C\, K^{-4/r_{\mathrm{int}}}.$$

The proof is provided in Appendix A. Theorem 3.1 is a hypothesis-class comparison: even with perfect optimization, $K$-clustered personalization is fundamentally limited by quantization bias. Client-specific collaboration strictly enlarges the representable solution set and can approximate a smooth continuum of heterogeneous optima with higher order accuracy.

In practice, however, whether this expressivity advantage is realized critically depends on how the collaboration kernel $\{W^t\}$ evolves during training. We next analyze the coupled Training and Aggregation dynamics, and show that improper collaboration geometry can induce collapse, thereby erasing its expressivity advantage.

## 3.3. Collaboration-induced collapse to consensus

This subsection studies a collaboration-induced collapse tendency: if the aggregation kernel is contractive on the non-consensus directions, then each round suppresses inter-client differences and pushes the system toward near-consensus, thereby gradually degenerates toward standard FL.

We make this precise by tracking the disagreement energy. $\boldsymbol{W}^t$ is primitive row-stochastic. Let $\boldsymbol{\pi}^t$ denote the stationary distribution: $(\boldsymbol{\pi}^t)^\top \boldsymbol{W}^t = (\boldsymbol{\pi}^t)^\top$, $\pi_i^t > 0$, $\sum_i \pi_i^t = 1$. Define the consensus projector and disagreement projector:

$$\boldsymbol{P}^t \triangleq \left(\mathbf{1}(\boldsymbol{\pi}^t)^\top\right) \otimes \boldsymbol{I}_d, \quad \boldsymbol{M}^t \triangleq \boldsymbol{I}_{Nd} - \boldsymbol{P}^t. \tag{2}$$

Then the projection onto the disagreement subspace (disagreement component) could be defined as: $\boldsymbol{S}_{\mathrm{dis}}^t \triangleq \boldsymbol{M}^t \boldsymbol{S}^t$. We measure disagreement via

$$\Delta(t) \triangleq \|\boldsymbol{S}_{\mathrm{dis}}^t\|_F^2. \tag{3}$$

Since the consensus direction corresponds to the top singular mode ($\sigma_1 = 1$), the remaining contraction on disagreement is governed by the second singular value. Define the disagreement contraction factor

$$\rho_W^t \triangleq \sigma_2\Big((\boldsymbol{D}_{\pi^t})^{1/2} \boldsymbol{W}^t (\boldsymbol{D}_{\pi^t})^{-1/2}\Big). \quad (4)$$

where $\boldsymbol{D}_{\pi^t} \triangleq \mathrm{diag}(\pi_1^t, \ldots, \pi_N^t)$. The $\rho_W^t$ could capture the worst-case contraction of inter-client differences induced by one aggregation step.

Based on this, we characterize how collaboration and local training jointly compress personalization by tracking the disagreement energy $\Delta(t)$,

**Theorem 3.2** (Disagreement Upper Bound). *Assume that for all rounds $t$: (i) $\boldsymbol{W}^t$ is primitive row-stochastic and admits a disagreement contraction factor $\rho_W^t < 1$; (ii) local training induces a contractive operator with $\|\boldsymbol{G}\|_2 \leq \rho_G < 1$; (iii) the per-round forcing on the disagreement subspace is bounded by a scale $B_{\mathrm{het}}$. Let $\overline{\rho}_W \triangleq \sup_t \rho_W^t < 1$, and assume $2(\overline{\rho}_W \rho_G)^2 < 1$. Then there exists a constant $C \geq 2$ such that*

$$\Delta(t) \;\leq\; \big(\sqrt{2}\,\overline{\rho}_W \rho_G\big)^{2t} \Delta(0) \;+\; \frac{C\,\overline{\rho}_W^2\,B_{\mathrm{het}}^2}{1 - 2(\overline{\rho}_W \rho_G)^2}. \quad (5)$$

*In particular, if $\boldsymbol{W}^t = \mathbf{1}(\boldsymbol{\pi}^t)^\top$ is rank-one consensus for all $t$ (so $\rho_W^t = 0$), then $\Delta(t) = 0$ for all $t \geq 1$.*

The proof is provided in Appendix C. Theorem 3.2 shows that the combined effect of collaboration and local training drives the disagreement $\Delta(t)$ into a small steady-state region. Under the coupled recursion, disagreement follows a transient–steady-state behavior: the transient term decays geometrically at rate $\overline{\rho}_W \rho_G$, while the steady-state level is set by the heterogeneity forcing $B_{\mathrm{het}}$. Here $\overline{\rho}_W$ summarizes how strongly aggregation contracts non-consensus differences, $\rho_G$ captures the contraction effect of local training, and $B_{\mathrm{het}}$ is an irreducible floor induced by mismatched client optima. Consequently, any contractive mixing ($\rho_W^t < 1$) progressively suppresses inter-client differences, making the dynamics increasingly FL-like. See Appendix C.1 for a more detailed interpretation.

A direct mitigation strategy is to weaken collaboration by reducing the aggregation intensity. However, weaker collaboration also reduces the effective hypothesis class available to each client and thus forfeits the expressivity advantage established in Theorem 3.1. This motivates controlling the collaboration kernel $W^t$ rather than simply turning collaboration off.

Many existing schemes control $W^t$ through similarity-driven feedback. However, such positive feedback can push the dynamics to the opposite extreme: the collaboration graph becomes self-separating and the system collapses to multiple consensus points (degenerating to clustered FL). Building on Theorem 3.2 (which captures the $K = 1$ consensus collapse), we next extend the analysis to the block-consensus regime and show how similarity-driven feedback leads to self-clustering and multi-point collapse.

### 3.4. Similarity-induced degeneration to CFL

This section explains a complementary phenomenon: when the collaboration matrix is dynamically determined by model similarity, collaboration can self-discretize and degenerate to clustered FL (CFL), i.e., a multi-point collapse.

In many collaborative PFL schemes (Ye et al., 2023), the weight matrix is dynamically generated from the current models:

$$\boldsymbol{W}^t = \Phi(\boldsymbol{S}^t), \quad (6)$$

where $\Phi(\cdot)$ tends to assign larger weights to more similar models. Such rules create a feedback loop: similar clients mix more and become even more similar; dissimilar clients mix less and are increasingly governed by their distinct local objectives.

Similarity-based weighting suppresses cross-group connections as cross-group distances grow. In spectral terms, the global disagreement contraction factor $\rho_W^t$ approaches 1. Importantly, $\rho_W^t \to 1$ does not indicate parameter divergence; it indicates that global consensus is no longer enforced and the correct limiting geometry is block consensus: each emergent group mixes internally, while different groups exchange negligible mass.

**multi-point collapse (degeneration to CFL).** Once the dynamics enters the block-consensus regime, the analysis from Theorem 3.2 applies within each group, driving intra-group disagreement into a small steady-state ball governed by the heterogeneity scale $B_{\mathrm{het}}$. Meanwhile, cross-group mixing vanishes, so group means remain separated near their distinct optima. Therefore, the collaboration degenerates to CFL. We formalize the above self-separation phenomenon and quantify the resulting block-wise collapse in the following theorem.

**Theorem 3.3** (Similarity-induced self-clustering). *Assume that there exist $t_0$ and a fixed partition $\mathcal{C} = \{C_1, \ldots, C_K\}$ ($K \geq 1$) such that the cross-block leakage*

$$\varepsilon_t \triangleq \max_k \max_{i \in C_k} \sum_{\substack{j \notin C_k}} \boldsymbol{W}_{ij}^t \quad (7)$$

*satisfies $\varepsilon_t \to 0$, and the within-block restriction $\overline{\boldsymbol{W}}^t$ has a uniform contraction $\rho_{\mathrm{in}}^t \leq \overline{\rho}_{\mathrm{in}} < 1$ for all $t \geq t_0$ (definition in Appendix D). Suppose moreover that the same local-regime conditions as in Theorem 3.2 hold, so that $\|\boldsymbol{G}\|_2 \leq \rho_G < 1$ and the forcing is uniformly bounded by $B_{\mathrm{het}}$, and*

*assume $2(\overline{\rho}_{\text{in}}\rho_G)^2 < 1$. Then there exists a constant $C > 0$ such that for all $t \geq t_0$,*

$$\sum_{k=1}^{K} \sum_{i \in C_k} \pi_i^t \left\| \theta_i^t - \bar{\theta}_k^t \right\|_2^2 \leq (\sqrt{2}\,\overline{\rho}_{\text{in}}\rho_G)^{2(t-t_0)} \Delta_{\text{in}}(t_0)$$

$$+ \frac{4\,\overline{\rho}_{\text{in}}^2 B_{\text{het}}^2}{1 - 2(\overline{\rho}_{\text{in}}\rho_G)^2} + C \sup_{s \geq t_0} \varepsilon_s^2, \tag{8}$$

*where $\bar{\theta}_k^t \triangleq \sum_{j \in C_k} \pi_{k,j}^t \theta_j^t$ is the block representative. Consequently, as $\varepsilon_t \to 0$, cross-block mixing vanishes and the iterates behave as a $K$-prototype clustered limit (degeneration to $K$-CFL); the case $K = 1$ reduces to the one-point consensus degeneration.*

Proof See Appendix D. Theorem 3.3 formalizes a self-discretization effect of similarity-driven collaboration. When cross-block leakage $\varepsilon_t \to 0$, the collaboration graph becomes near-reducible. In short, consensus becomes group-wise rather than global. Inside each block, the normalized restriction $\overline{W}^t$ admits a uniform mixing gap $\rho_{\text{in}}^t \leq \overline{\rho}_{\text{in}} < 1$, so the composition of within-block aggregation and contractive local training ($\|G\| \leq \rho_G < 1$) exponentially suppresses historical within-block discrepancies at rate $\overline{\rho}_{\text{in}}\rho_G$.

The remaining two terms in the bound quantify the steady-state sources of within-block disagreement: the heterogeneity forcing $B_{\text{het}}$ induced by mismatched client optima, and the residual cross-block leakage $\varepsilon_t$. Consequently, as $\varepsilon_t$ becomes negligible, clients concentrate around $K$ block representatives $\{\bar{\theta}_k^t\}_{k=1}^K$, meaning that similarity-based collaboration degenerates to a clustered FL (CFL) solution with $K$ clusters, with the special case $K = 1$ recovering the one-point consensus collapse. The Fig. 6 in Appendix F illustrates these collapses.

We have established that collaboration strictly dominates clustering as a hypothesis class (Theorem 3.1). However, Theorems 3.2 and 3.3 show that this representational advantage is fragile: improper collaboration geometry can either over-contract personalization toward (near-)consensus or self-separate into blocks and degenerate to $K$-prototype CFL. These two failure modes suggest that the key is not whether to collaborate, but how to choose the collaboration kernel $\{W^t\}$ across rounds.

In the next section, we develop a feedback-free and stage-controlled design of $W^t$ that preserves collaborative expressivity while avoiding both consensus collapse and similarity-induced self-clustering.

## 4. Methodology

On the one hand, overly strong mixing (very small $\rho_W^t$) contracts disagreement too aggressively and flattens personalization towards near-consensus (Theorem 3.2). On the other hand, similarity-feedback rules $W^t = \Phi(S^t)$ can suppress cross-group flow and make the induced Markov chain near-reducible; in the long run, the dynamics self-discretizes and degenerates to a $K$-prototype CFL solution (Theorem 3.3).

Motivated by the two failure modes, we seek controlled collaboration geometry (CCG). Accordingly, we propose pFedCCG, a feedback-free and stage-controlled scheme that designs the time-varying collaboration kernel $\{W^t\}$ to preserve collaborative expressivity while avoiding both consensus collapse and similarity-induced self-clustering. Specifically,

- **Feedback-free affinity template.** We fix a static client-wise template $\Psi$ (independent of current models) to break similarity feedback and prevent self-clustering.

- **Objective-aligned geometry.** We enforce $q$ as the stationary distribution of $W^t$ to align collaboration geometry with the $q$-weighted objective (define in 3.1).

- **Projection-based kernel design.** At each round, we obtain a valid kernel by projecting $\alpha_t \Psi$ onto the Markov set (nonnegativity, row-stochasticity, and $\pi = q$), where $\alpha_t$ controls the aggregation intensity, keeping $W^t$ as close as possible to the target geometry.

- **Collaboration geometry scheduling.** We schedule $\alpha_t$ to regulate aggregation intensity over time: stronger aggregation in early rounds to quickly enter a favorable optimization region, and weaker aggregation in later rounds to preserve personalization and avoid collapse.

### 4.1. Feedback-free affinity template

To avoid the self-reinforcing dynamics in Theorem 3.3, we cut the dependence of $W^t$ on current model parameters and use a static client-wise similarity proxy that is fixed across rounds.

For each client $i$, let $\zeta_i$ denote any lightweight, client-local descriptor that is available a priori and is stable across training rounds (e.g., data statistics, feature summaries, reference embeddings, metadata-based profiles, etc.). We define a generic pairwise similarity indicator

$$\psi_{ij} \triangleq \text{sim}(\zeta_i, \zeta_j), \tag{9}$$

which is intended to correlate with the proximity of client-wise optima: larger $\psi_{ij}$ suggests that $\|\theta_i^\star - \theta_j^\star\|$ is smaller and that collaboration between $i$ and $j$ is more beneficial.

### 4.2. Objective-aligned geometry $\pi = q$

We measure personalization by the $q$-weighted excess risk $\sum_i q_i(\cdot)$, and our disagreement analysis is naturally expressed in the stationary geometry $\|\cdot\|_\pi$ induced by $W^t$

(Section 3.3). To align them, we set the target stationary distribution to the dataset proportions:

$$q^\top \boldsymbol{W}^t = q^\top, \quad q_i > 0, \sum_{i=1}^N q_i = 1. \tag{10}$$

Let $D_q \triangleq \mathrm{diag}(q)$. We use the symmetric flow matrix

$$F^t \triangleq D_q \boldsymbol{W}^t, \tag{11}$$

so that $W^t = D_q^{-1} F^t$. Imposing detailed balance $F^t = (F^t)^\top$ guarantees that $q$ is stationary. Moreover, row-stochasticity of $W^t$ is equivalent to

$$W^t \mathbf{1} = \mathbf{1} \quad \Longleftrightarrow \quad F^t \mathbf{1} = q. \tag{12}$$

In addition, since $q_i > 0$, the Markov nonnegativity constraint is equivalently expressed in $K$:

$$W^t \geq 0 \quad \Longleftrightarrow \quad F^t \geq 0. \tag{13}$$

Thus we design $W^t$ by optimizing over the convex feasible set

$$\mathcal{F}(q) \triangleq \left\{ F \in \mathbb{R}_+^{N \times N} : F = F^\top, F\mathbf{1} = q \right\}, \tag{14}$$

and then $W^t = D_q^{-1} F^t$.

### 4.3. Projection-based kernel design

The remaining question is how to turn $\Psi$ into a valid collaboration kernel. A feasible collaboration matrix must (i) be nonnegative and row-stochastic, and (ii) match the risk-weight geometry used in our theory. We therefore restrict $W$ to the Markov set with prescribed stationary distribution $q$:

$$\mathcal{W}(q) \triangleq \{ W \in \mathbb{R}_+^{N \times N} : W\mathbf{1} = \mathbf{1}, q^\top W = q^\top \}. \tag{15}$$

Since $\Psi$ is only a template (it may not be stochastic nor $q$-stationary), we obtain $W^t$ by projecting the scaled template $\alpha_t \Psi$ onto $\mathcal{W}(q)$. Accordingly, we measure the mismatch between $W$ and the template in the $q$-weighted Frobenius norm,

$$\|D_q^{1/2}(W - \alpha_t \Psi)\|_F^2 = \sum_{i=1}^N q_i \|w_i - \alpha_t \psi_i\|_2^2, \tag{16}$$

which penalizes deviations of each row $w_i$ proportionally to its dataset weight. This yields the following projection objective with a single stage-wise knob $\alpha_t$.

$$\min_{W \geq 0} \left\| D_q^{1/2}(W - \alpha_t \Psi) \right\|_F^2 \quad \text{s.t.} \quad W\mathbf{1} = \mathbf{1}, \ q^\top W = q^\top. \tag{17}$$

Substituting $W = D_q^{-1} F$ with $F \in \mathcal{F}(q)$, (17) becomes the convex quadratic program

$$\min_{F \in \mathcal{F}(q)} \left\| D_q^{-1/2} F - \alpha_t D_q^{1/2} \Psi \right\|_F^2, \quad W^t = D_q^{-1} F^t. \tag{18}$$

As $\alpha_t$ increases, the optimizer is pushed to track the affinity structure in $\Psi$ as much as possible, while still satisfying nonnegativity, stochasticity, and $\pi = q$ by construction.

**Solver.** We solve (18) using projected gradient descent with alternating projections onto (i) nonnegativity and (ii) the symmetric-marginal constraint set $\{F = F^\top, \ F\mathbf{1} = q\}$. The second projection admits a closed-form Euclidean solution, so each iteration is $O(N^2)$ and converges rapidly in practice.

### 4.4. Collaboration geometry scheduling

Our theory suggests a simple stage-wise control principle for $\{W^t\}$. Early in training, heterogeneous local updates can rapidly separate client trajectories. If clients enter different attraction basins, cross-client averaging becomes detrimental: it tends to pull the iterate out of each basin, so the mixed model no longer improves any client reliably. Consequently, useful mixing persists mainly among clients that remain close, while weights assigned to far-apart clients effectively vanish, matching the self-separation pathway that leads to the block-consensus (CFL-like) degeneration in Theorem 3.3.

Therefore, we favor stronger global mixing in the early stage to keep trajectories in a nearby region and reduce the chance of premature fragmentation. In the middle/late stage, however, persistent strong mixing over-contracts disagreement directions and pushes the system toward a near-consensus, FL-like solution (Theorem 3.2), so mixing should be gradually weakened to preserve personalization.

In our projection-based design (17)–(18), $\alpha_t$ could adjusts the mixing strength while maintaining $W^t \in \mathcal{W}(q)$. Smaller $\alpha_t$ yields a denser kernel closer to $\mathbf{1}q^\top$, whereas larger $\alpha_t$ makes $W^t$ adhere more strongly to the affinity template $\Psi$, thereby reducing global mixing. Although our analysis characterizes the contraction and separation tendencies through quantities such as $\rho_W^t$ and cross-block leakage, these terms are not directly accessible for closed-form control under the projection ((17)–(18)) and can be expensive to estimate during training.

We therefore adopt simple monotone schedules for $\alpha_t$ as practical surrogates that implement the above stage-wise principle with minimal additional computation.

$$\alpha_t = \alpha_{\min} + (\alpha_{\max} - \alpha_{\min}) \cdot \frac{1}{1 + \exp\left(-\frac{t - t_s}{\tau}\right)}, \tag{19}$$

where $t_s$ is the transition round and $\tau$ controls the smooth-

*Table 1.* Unified comparison of personalized accuracy averaged over clients under varying heterogeneity. Numbers are the mean (std) of personalized performance across 50 clients after 50 rounds. CIFAR-10 and Yahoo report accuracy with Dirichlet $\beta \in \{0.2, 0.5, 1\}$, while MovieLens reports Recall@20 and HR@20 with $\beta \in \{0.5, 1\}$.

| Model | Cifar10 (ResNet10) | | | Yahoo (TextCNN) | | | Movielens (NCF) | | | |
| --- | --- | --- | --- | --- | --- | --- | --- | --- | --- | --- |
| | $\beta$=0.2 | $\beta$=0.5 | $\beta$=1 | $\beta$=0.2 | $\beta$=0.5 | $\beta$=1 | $\beta$=0.5 | | $\beta$=1 | |
| | Accuracy | Accuracy | Accuracy | Accuracy | Accuracy | Accuracy | Recall@20 | HR@20 | Recall@20 | HR@20 |
| Local | 77.40 (10.33) | 65.97 (9.41) | 59.97 (7.43) | 74.16 (11.27) | 62.50 (9.26) | 55.69 (7.16) | 84.48 (5.96) | 88.81 (5.46) | 82.11 (6.12) | 86.72 (5.78) |
| FedAvg-FT | 83.61 (7.83) | 78.38 (5.76) | 76.20 (3.58) | 78.76 (8.95) | 69.86 (7.18) | 64.91 (5.80) | 87.06 (4.92) | 91.19 (4.01) | 85.33 (5.05) | 89.45 (4.23) |
| FedProx-FT | 83.79 (7.72) | 78.39 (5.76) | 76.27 (3.60) | 78.71 (9.01) | 69.74 (7.02) | 64.88 (5.79) | 86.65 (4.86) | 90.63 (4.01) | 84.98 (5.11) | 88.97 (4.29) |
| FedAMP | 76.00 (10.77) | 65.51 (8.37) | 59.83 (6.20) | 76.81 (10.29) | 69.27 (7.05) | 64.16 (5.87) | 86.96 (4.99) | 90.81 (4.26) | 85.27 (5.08) | 89.11 (4.35) |
| CFL | 81.36 (8.90) | 74.81 (6.84) | 68.07 (5.74) | 78.76 (8.95) | 69.86 (7.19) | 64.89 (5.79) | 86.73 (4.90) | 90.73 (4.14) | 85.03 (5.09) | 89.03 (4.31) |
| pFedGraph | 82.64 (8.42) | 77.32 (5.92) | 75.40 (4.17) | 77.81 (9.75) | 69.03 (7.28) | 64.87 (6.10) | 87.84 (4.70) | 91.66 (3.85) | 86.05 (4.86) | 89.95 (4.07) |
| FedCollab | 82.48 (8.40) | 77.51 (5.85) | 76.21 (3.35) | 76.01 (10.55) | 69.38 (7.16) | 64.91 (5.80) | 86.27 (4.93) | 90.59 (4.03) | 84.58 (5.15) | 88.83 (4.33) |
| GraphRx | 83.39 (7.91) | 78.45 (5.73) | 76.14 (3.55) | 78.86 (8.87) | 69.87 (7.17) | 64.89 (5.73) | 86.38 (5.09) | 90.50 (4.26) | 84.69 (5.22) | 88.74 (4.41) |
| FedGWC | 83.88 (8.08) | 78.25 (5.58) | 75.73 (4.06) | 78.46 (9.41) | 69.83 (7.10) | 64.81 (5.41) | 87.32 (4.77) | 91.32 (3.79) | 85.56 (4.92) | 89.61 (3.98) |
| CFLGP | 82.68 (8.17) | 77.00 (6.40) | 74.38 (4.76) | 77.82 (9.92) | 69.27 (7.62) | 64.09 (6.02) | 87.83 (4.68) | 91.68 (3.78) | 86.04 (4.84) | 89.97 (4.04) |
| pFedCS | 82.95 (8.16) | 76.02 (6.51) | 72.03 (5.11) | 76.01 (10.32) | 65.43 (8.15) | 59.62 (6.63) | 86.29 (5.06) | 90.42 (4.22) | 84.59 (5.19) | 88.65 (4.39) |
| FedAGHN | 83.97 (7.62) | 78.56 (5.78) | 75.39 (4.41) | 78.02 (9.56) | 69.80 (7.34) | 65.00 (5.79) | 87.72 (4.73) | 91.61 (3.85) | 85.93 (4.88) | 89.90 (4.09) |
| pFedCCG | **85.20 (6.94)** | **80.36 (4.97)** | **78.35 (3.44)** | **79.76 (8.95)** | **70.61 (7.01)** | **65.87 (5.48)** | **88.81 (4.38)** | **92.50 (3.41)** | **86.31 (5.56)** | **90.69 (3.72)** |

ness. Compared to the linear ramp, the sigmoid ramp keeps $\alpha_t$ closer to $\alpha_{\min}$ for longer (strong early mixing), then transitions more decisively to $\alpha_{\max}$ (weaker late mixing).

# 5. Experiments

## 5.1. Experimental Setup

**Tasks & datasets.** We organize the experiments on four tasks: image classification on CIFAR-10 (Krizhevsky et al., 2009) dataset with ResNet-10 (He et al., 2016), NLP on Yahoo Answers (Zhang et al., 2015) dataset with TextCNN (Zhu et al., 2020), recommendation on MovieLens (Harper & Konstan, 2015) dataset with NCF model, and LLM supervised fine-tuning (LLM-SFT) on Aya (Singh et al., 2024) dataset with a LoRA-adapted LLM.

**Baselines.** We compared 12 baseline methods: local training; traditional FL algorithms local fine-tuning including FedAvg-FT (McMahan et al., 2017) and FedProx-FT (Li et al., 2020);Cluster based PFL algorithms including CFL (Sattler et al., 2020), FedGWC (Licciardi et al., 2025), CFLGP (Kim et al., 2024) and pFedCS (Wu et al., 2025); and the collaboration-based PFL algorithms including FedAMP (Huang et al., 2021), pFedGraph (Ye et al., 2023), FedCollab (Bao et al., 2023), GraphRx (Wang et al., 2025) and FedAGHN (Song et al., 2025).

**Training setting.** We considered non independent and identically distributed (non-IID) data settings with Dirichlet coefficient $\beta$. We conduct experiments over 50 communication rounds with 50 clients. Additional implementation details are provided in Appendix E. All personalized performance metrics are evaluated on the client-specific test set that follows the same local data distribution as the corresponding client. We evaluate $\Psi$ using JS divergence of data distribution and cosine similarity of initial model updates, respectively.

## 5.2. Experimental Results

**Main results with non-IID setting on different tasks.** Table 1 summarizes the personalized performance (mean±std over 50 clients) on three representative tasks: image classification (CIFAR-10), text classification (Yahoo), and recommendation (MovieLens). Across all heterogeneity levels and metrics, our method consistently achieves the best average personalized performance. These consistent improvements across modalities indicate that the proposed controlled collaboration geometry is not task-specific. Table 2 reports LLM supervised fine-tuning results on Aya under language heterogeneity. We select the top-10 most frequent languages in dataset and construct 20 clients, where each client contains $k \in \{2, 3\}$ languages. Our method achieves the best performance under both heterogeneity regimes. These consistent gains indicate that the proposed controlled collaboration geometry transfers effectively to LLM adaptation. In addition to improving the mean, our method exhibits competitive cross-client dispersion. Our method maintains comparable (often lower) dispersion while achieving the best mean performance, which is consistent with the goal of controlled collaboration: enabling beneficial transfer without collapsing personalization into overly uniform solutions. For more details, please refer to Fig. 5.

**Visualization on collaboration matrix.** Figure 2 visualizes the collaboration kernels $W^t$ at Round 15 on CIFAR-10 with 20 clients. Clustered PFL baselines exhibit pronounced block/partition patterns in $W^t$ (e.g., CFL, FedGWC, CFLGP, pFedCS), where large weights concentrate on a restricted subset of clients and cross-group weights remain sparse. Similarity-driven collaborative schemes

*Table 2.* LLM LoRA SFT on Aya dataset. Report the mean±std of client-wise personalized performance across 20 clients after 30 rounds, where each client contains $k \in \{2, 3\}$ languages.

| Method | Aya (LoRA-adapted LLM) | | | |
| | $k = 2$ | | $k = 3$ | |
| | PPL ↓ | Acc@5 ↑ | PPL ↓ | Acc@5 ↑ |
|---|---|---|---|---|
| Local | 7.12 (4.10) | 79.82 (7.42) | 6.62 (2.02) | 79.61 (4.55) |
| FedAvg-FT | 7.20 (4.28) | 79.77 (7.68) | 6.46 (1.96) | 79.96 (4.56) |
| FedProx-FT | 7.34 (4.40) | 79.46 (7.79) | 6.53 (1.99) | 79.78 (4.61) |
| FedAMP | 7.68 (4.71) | 78.99 (7.90) | 6.85 (2.14) | 79.16 (4.71) |
| CFL | 7.14 (4.17) | 79.79 (7.64) | 6.45 (1.96) | 79.94 (4.55) |
| pFedGraph | 6.90 (3.83) | 80.07 (7.36) | 6.45 (1.96) | 79.95 (4.56) |
| FedCollab | 7.11 (4.20) | 79.91 (7.68) | 6.43 (1.94) | 80.04 (4.51) |
| GraphRx | 7.19 (4.25) | 79.74 (7.67) | 6.44 (1.95) | 79.97 (4.58) |
| FedGWC | 6.99 (4.02) | 79.95 (7.56) | 6.43 (1.95) | 80.00 (4.57) |
| CFLGP | 7.15 (4.16) | 79.76 (7.67) | 6.49 (1.98) | 79.82 (4.59) |
| pFedCS | 7.01 (4.02) | 79.99 (7.43) | 6.55 (1.97) | 79.69 (4.57) |
| FedAGHN | 7.00 (3.98) | 79.90 (7.50) | 6.41 (1.94) | 79.99 (4.50) |
| pFedCCG | **6.84 (3.88)** | **80.29 (7.29)** | **6.35 (1.92)** | **80.12 (4.53)** |

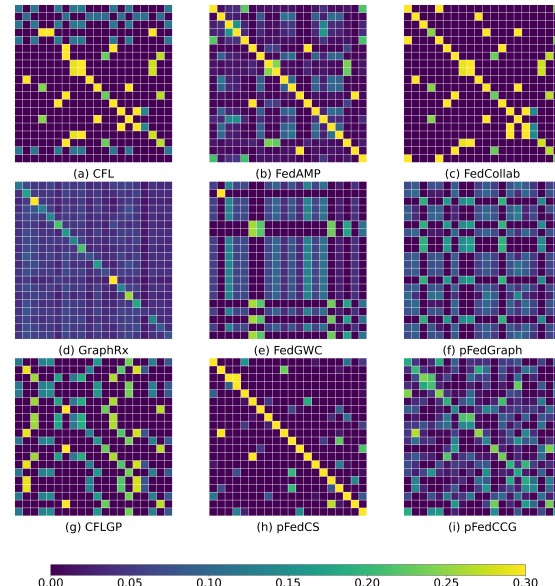

*Figure 2.* Collaboration-matrix $W$ heatmaps at Round 15 on CIFAR-10 ($N = 20, \beta = 0.2$).

(pFedGraph, FedAMP, GraphRx) show clear collapse already at Round 15. Specifically, pFedGraph produces highly structured row/column "stripe" patterns, indicating that many clients place comparable weights on a small set of preferred peers, which amplifies similarity feedback and accelerates self-separation of the collaboration graph. FedAMP and GraphRx further show strong diagonal concentration with only narrow or weak off-diagonal support, implying that aggregation becomes predominantly local (or nearly identity-like), thereby reducing effective cross-client transfer. In contrast, the proposed method remains non-trivially dense with moderated, structured off-diagonal weights, suggesting that cross-client information flow is preserved without collapsing into a few isolated components. For more details, please refer to Fig. 3.

**Ablation.** To assess the contribution of each component, we conduct ablation studies on four tasks with five variants: *SF*, which replaces the feedback-free template with similarity-feedback dynamic updates; *w/o GC*, which removes collaboration geometry scheduling and uses static collaboration; *w/o q*, which removes the prescribed stationary distribution and thus discards distribution-aware projection; and *pFedCCG-cos*, which constructs the affinity template using cosine similarity of initial model updates instead of JS divergence. As shown in Table 3, the feedback-free variants consistently outperform *SF*, especially under highly heterogeneous settings such as CIFAR-10 and Yahoo with $\beta = 0.2$. This indicates that decoupling the collaboration kernel from real-time model parameters helps avoid consensus collapse or self-clustering and better preserves personalization expressivity.

Moreover, removing geometry control or the prescribed stationary distribution leads to performance degradation,

suggesting that both scheduling-based geometry control and objective-aligned projection contribute to stable collaboration. In particular, *w/o q* shows that ignoring client-specific data proportions during kernel construction weakens knowledge transfer. The full pFedCCG achieves the best or most competitive performance across tasks, indicating that geometry scheduling and distribution-aware projection effectively balance consensus and personalization. In addition, *pFedCCG-cos* remains close to the full JS-based variant, showing that initial model updates can serve as an effective proxy when direct data-distribution statistics are unavailable.

*Table 3.* Ablation study. For LLM-SFT, lower PPL is better; for the other tasks, higher accuracy or Recall@20 is better.

| Settings | CIFAR-10 ($\beta = 0.2$) | Yahoo ($\beta = 0.2$) | RecSys ($\beta = 0.5$) | LLM-SFT ($k = 3$) |
|---|---|---|---|---|
| SF | 82.64 (8.42) | 77.81 (9.75) | 87.84 (4.70) | 6.45 (1.96) |
| w/o GC | 84.72 (7.35) | 78.95 (14.00) | 88.62 (4.42) | 6.39 (1.93) |
| w/o $q$ | 85.01 (7.11) | 79.27 (8.99) | 88.75 (4.42) | 6.36 (1.92) |
| pFedCCG-cos | 85.22 (6.97) | 79.46 (9.03) | 88.56 (5.00) | 6.39 (1.94) |
| pFedCCG | 85.20 (6.94) | 79.76 (8.95) | 88.81 (4.38) | 6.35 (1.92) |

**Robustness to weaker affinity proxies.** We further evaluate whether the performance gain relies on an oracle-quality template constructed from true client data distributions. We compare the default JS-divergence template with several weaker proxy templates on CIFAR-10 under different heterogeneity levels. Specifically, *cos-JS* computes JS similarity over proxies derived from initial update directions; *full-cos* directly uses the cosine similarity of full-model initial updates; *full-cos-normalization* further normalizes

*Table 4.* Robustness to different affinity proxy templates on CIFAR-10. We report the mean personalized accuracy across 50 clients under different Dirichlet heterogeneity levels.

| Method | $\beta = 0.2$ | $\beta = 0.5$ | $\beta = 1$ |
|---|---|---|---|
| pFedCCG-JS | 85.20 | 80.36 | 78.35 |
| cos-JS | 85.08 | 79.52 | 77.15 |
| full-cos | 85.22 | 80.19 | 77.73 |
| full-cos-normalization | 85.07 | 80.08 | 77.78 |
| classifier-cos | 85.24 | 79.90 | 77.43 |
| softmax-JS | 84.86 | 79.88 | 77.89 |
| softmax-cos | 84.59 | 79.38 | 77.49 |
| Best baseline | 83.97 | 78.56 | 76.27 |

*Table 5.* Additional collaboration overhead under different numbers of clients (unit: seconds).

| Method | 20 | 50 | 100 | 150 | 200 | 300 | 500 |
|---|---|---|---|---|---|---|---|
| FedAMP | 0.10 | 0.61 | 2.42 | 5.42 | 9.71 | 21.77 | 60.16 |
| CFL | 0.11 | 0.65 | 2.51 | 5.57 | 9.87 | 21.80 | 60.78 |
| pFedGraph | 46.83 | 281.60 | 349.71 | – | – | – | – |
| FedCollab | 1.23 | 43.21 | 697.20 | – | – | – | – |
| GraphRx | 12.27 | 77.95 | 237.63 | 512.35 | – | – | – |
| FedGWC | 0.01 | 0.02 | 0.16 | 0.25 | 0.45 | 1.10 | 3.05 |
| CFLGP | 1.37 | 4.14 | 9.23 | 18.34 | 26.16 | 31.58 | 63.45 |
| pFedCS | 0.11 | 0.47 | 1.21 | 2.59 | 4.28 | 9.52 | 27.93 |
| FedAGHN | 106.14 | 121.05 | 163.65 | 239.91 | 332.83 | – | – |
| pFedCCG | 0.02 | 0.04 | 0.15 | 1.23 | 5.27 | 7.58 | 13.34 |

the full-model update vectors before computing cosine similarity; *classifier-cos* uses only the classifier-layer initial updates; and *softmax-JS* and *softmax-cos* derive similarity from model softmax outputs on a small public anchor set of 256 samples. As shown in Table 4, all proxy-based variants consistently outperform the strongest baseline across three heterogeneity levels. This indicates that pFedCCG does not require an oracle-like affinity template. A coarse but stable similarity proxy is sufficient to provide useful collaboration priors. Meanwhile, the performance gap among different proxies also shows that the template quality still matters: stronger proxies, such as JS-based and full-model cosine similarities, generally achieve better results than output-based proxies. These results support the design principle of pFedCCG: the key is not to obtain a perfect similarity estimate, but to use a stable feedback-free template and transform it into a valid controlled collaboration geometry.

**Additional collaboration overhead.** Since pFedCCG introduces a projection-based kernel construction step, we further evaluate its additional collaboration overhead. We report the wall-clock runtime of the extra collaboration computation, excluding local training and standard model aggregation, under different numbers of clients. As shown in Table 5, pFedCCG introduces only a small overhead in the main experimental scale. For example, with 50 clients, the additional overhead of pFedCCG is only 0.04 seconds, which is negligible compared with local model training. Moreover, pFedCCG is substantially more efficient than adaptive graph-based methods such as pFedGraph, FedCollab, GraphRx, and FedAGHN, whose overhead increases rapidly with the number of clients. Although simple weighting-based methods such as FedGWC can be faster in some large-scale cases, pFedCCG remains efficient and scales to 500 clients with a moderate overhead. These results indicate that the projection step does not constitute a computational bottleneck in practice.

## 6. Conclusions

In this paper, we identified two degeneration regimes in collaborative PFL: consensus collapse under overly strong mixing and self-clustering under similarity-feedback updates. Guided by a unified dynamical analysis, we proposed pFedCCG to control collaboration geometry by decoupling similarity from training, constructing valid Markovian collaboration kernels via projection, and scheduling collaboration strength to avoid collapse. Experiments across diverse heterogeneity settings consistently show improved personalization and markedly reduced consensus collapse and self-clustering. Future work will explore adaptive, data-driven geometry control and extend the framework to more complex task scenarios. Moreover, stronger privacy-preserving mechanisms for constructing the affinity template will also be investigated.

**Limitations** First, pFedCCG relies on a fixed affinity template as a stable collaboration prior. Our robustness experiments show that the method tolerates noisy and weaker proxy templates, but a severely misspecified template may still weaken collaboration quality. Second, while the projection step introduces only moderate overhead in our experiments, its cost still grows with the number of clients and may require further acceleration for extremely large-scale federations. Third, our theoretical analysis is based on local linearization and bounded-iterate assumptions, which provide useful insight into collapse dynamics but may not fully characterize all phases of non-convex training. Finally, stronger privacy-preserving mechanisms for constructing affinity templates, such as secure similarity computation or differentially private statistics, are left for future work.

## Acknowledgements

This work was supported in part by the National Natural Science Foundation of China under Grant 62572099; in part by the National Natural Science Foundation of China under Grant U23A20310; in part by the Center for HPC, University of Electronic Science and Technology of China.

## Impact Statement

This paper presents work whose goal is to advance the field of Machine Learning. There are many potential societal consequences of our work, none which we feel must be specifically highlighted here.

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

# A. Proof of Theorem 3.1

For any hypothesis class $\Theta$ (a set of feasible $\{\boldsymbol{\theta}_i\}_{i=1}^N$), define

$$\mathcal{R}^\star(\Theta) \triangleq \inf_{\{\boldsymbol{\theta}_i\}\in\Theta} \mathcal{R}(\{\boldsymbol{\theta}_i\}), \tag{20}$$

is the smallest achievable or arbitrarily approachable excess risk within $\Theta$. Let $\boldsymbol{\theta}_i^\star$ denotes a theoretical reference optimum for client $i$.

$K$**-clustered class.** Clustering restricts personalization to choose one of $K$ shared prototypes, and each client uses a single prototype parameter. Formally,

$$\Theta_{\mathrm{cl}}(K) \triangleq \Big\{\boldsymbol{\theta}_i = \boldsymbol{c}_{k(i)} \text{ for some } \{\boldsymbol{c}_k\}_{k=1}^K \subset \mathbb{R}^d \text{ and assignment } k(\cdot)\Big\}, \quad \mathcal{R}_{\mathrm{cl}}^\star(K) \triangleq \mathcal{R}^\star(\Theta_{\mathrm{cl}}(K)). \tag{21}$$

$K$**-collaborative class.** Collaboration instead allows each client to mix the same $K$ representative models (anchors) with client-specific weights, so clients can interpolate between anchors rather than selecting a single one. Let $\mathcal{A} \subseteq \{1, \dots, N\}$ with $|\mathcal{A}| = K$ be an anchor set and define

$$\Theta_{\mathrm{col}}(K; \mathcal{A}) \triangleq \Big\{\boldsymbol{\theta}_i = \sum_{j\in\mathcal{A}} w_{ij}\boldsymbol{\theta}_j^\star, \sum_{j\in\mathcal{A}} w_{ij} = 1, \ w_{ij} \geq 0, \ \forall i\Big\}, \tag{22}$$

and we allow the best choice of anchors:

$$\mathcal{R}_{\mathrm{col}}^\star(K) \triangleq \inf_{\mathcal{A}:\, |\mathcal{A}|=K} \mathcal{R}^\star(\Theta_{\mathrm{col}}(K; \mathcal{A})). \tag{23}$$

Then, measure how far personalization deviates from the ideal client optima in parameter space via

$$\mathcal{E}(\{\boldsymbol{\theta}_i\}) \triangleq \sum_{i=1}^N q_i\|\boldsymbol{\theta}_i - \boldsymbol{\theta}_i^\star\|_2^2. \tag{24}$$

**Assumption A (local quadratic growth).** There exist constants $0 < \mu \leq L$ such that for all $i$ and all $\boldsymbol{\theta}$ in a neighborhood of $\boldsymbol{\theta}_i^\star$,

$$\frac{\mu}{2}\|\boldsymbol{\theta} - \boldsymbol{\theta}_i^\star\|_2^2 \ \leq \ L_i(\boldsymbol{\theta}) - L_i(\boldsymbol{\theta}_i^\star) \ \leq \ \frac{L}{2}\|\boldsymbol{\theta} - \boldsymbol{\theta}_i^\star\|_2^2. \tag{25}$$

**Lemma A.1** (Risk–distance equivalence near local optima)**.** *Under Assumption A, for any $\{\boldsymbol{\theta}_i\}$ staying in the above neighborhoods,*

$$\frac{\mu}{2}\mathcal{E}(\{\boldsymbol{\theta}_i\}) \ \leq \ \mathcal{R}(\{\boldsymbol{\theta}_i\}) \ \leq \ \frac{L}{2}\mathcal{E}(\{\boldsymbol{\theta}_i\}). \tag{26}$$

*Proof.* By Assumption A, multiplying by $q_i$ and summing over $i$ yields

$$\frac{\mu}{2}\sum_{i=1}^N q_i\|\boldsymbol{\theta}_i - \boldsymbol{\theta}_i^\star\|_2^2 \ \leq \ \sum_{i=1}^N q_i\big(L_i(\boldsymbol{\theta}_i) - L_i(\boldsymbol{\theta}_i^\star)\big) \ \leq \ \frac{L}{2}\sum_{i=1}^N q_i\|\boldsymbol{\theta}_i - \boldsymbol{\theta}_i^\star\|_2^2, \tag{27}$$

which is exactly $\frac{\mu}{2}\mathcal{E} \leq \mathcal{R} \leq \frac{L}{2}\mathcal{E}$. $\square$

When each $L_i$ is locally well-approximated by a quadratic around $\boldsymbol{\theta}_i^\star$ (local quadratic growth), the loss gap $L_i(\boldsymbol{\theta}_i) - L_i(\boldsymbol{\theta}_i^\star)$ scales linearly with $\|\boldsymbol{\theta}_i - \boldsymbol{\theta}_i^\star\|_2^2$ up to constants (Lemma A.1). Therefore, under the local quadratic-growth condition, the excess risk around $\{\boldsymbol{\theta}_i^\star\}$ is proportional to the squared distance to $\{\boldsymbol{\theta}_i^\star\}$. As a result, any bound on the best achievable distortion $\inf \mathcal{E}$ immediately translates to a bound of the same order on the best achievable excess risk $\inf \mathcal{R}$.

**Lemma A.2** (Clustering is a special case of collaboration)**.** *For any $K \geq 1$,*

$$\mathcal{R}_{\mathrm{col}}^\star(K) \leq \mathcal{R}_{\mathrm{cl}}^\star(K). \tag{28}$$

*Proof.* Fix any anchor set $\mathcal{A}$ with $|\mathcal{A}| = K$. Consider the restricted clustered class that can only use these anchors as prototypes:

$$\Theta_{\mathrm{cl}}(K; \mathcal{A}) \triangleq \left\{ \boldsymbol{\theta}_i = \boldsymbol{\theta}_{j(i)}^\star \text{ for some } j(i) \in \mathcal{A} \right\}. \tag{29}$$

This is a special case of $\Theta_{\mathrm{col}}(K; \mathcal{A})$ by choosing one-hot weights: for each $i$, set $w_{i,j(i)} = 1$ and $w_{ij} = 0$ for $j \neq j(i)$, which yields $\boldsymbol{\theta}_i = \sum_{j \in \mathcal{A}} w_{ij} \boldsymbol{\theta}_j^\star = \boldsymbol{\theta}_{j(i)}^\star$. Hence $\Theta_{\mathrm{cl}}(K; \mathcal{A}) \subseteq \Theta_{\mathrm{col}}(K; \mathcal{A})$, and therefore

$$\inf_{\Theta_{\mathrm{col}}(K; \mathcal{A})} \mathcal{R} \leq \inf_{\Theta_{\mathrm{cl}}(K; \mathcal{A})} \mathcal{R}. \tag{30}$$

Since the above inequality holds for every anchor set $\mathcal{A}$ with $|\mathcal{A}| = K$, taking $\inf_{|\mathcal{A}|=K}$ on both sides gives the desired result. $\qquad\square$

**Lemma A.3** (Clustered quantization lower bound (Zador-type)). *Assume the client optima $\{\boldsymbol{\theta}_i^\star\}$ lie on a smooth $r_{\mathrm{int}}$-dimensional latent manifold: there exist $\boldsymbol{\zeta}_i \in [0,1]^{r_{\mathrm{int}}}$ and a $C^2$ map $h$ with bounded curvature such that $\boldsymbol{\theta}_i^\star = h(\boldsymbol{\zeta}_i)$. Moreover, the latent points $\boldsymbol{\zeta}_i$ are non-degenerate (they have a density bounded away from $0$ and $\infty$ on $[0,1]^{r_{\mathrm{int}}}$). There exists $c_0 > 0$ independent of $K$ such that*

$$\inf_{\Theta_{\mathrm{cl}}(K)} \mathcal{E} \geq c_0 K^{-2/r_{\mathrm{int}}}. \tag{31}$$

*Proof.* This is the standard Zador-type lower bound for the optimal $K$-point quantization (equivalently, optimal $K$-means) distortion of an $r_{\mathrm{int}}$-dimensional absolutely continuous distribution (Gray & Neuhoff, 1998). $\qquad\square$

**Lemma A.4** (Collaborative piecewise-linear interpolation upper bound). *Assuming there exists an anchor set $\mathcal{A}$ of size $K$ whose latent points $\{\boldsymbol{\zeta}_j\}_{j \in \mathcal{A}}$ form a shape-regular triangulation (or quasi-uniform simplicial complex) of $[0,1]^{r_{\mathrm{int}}}$ with mesh size $h \asymp K^{-1/r_{\mathrm{int}}}$. Under Assumptions in Lemma A.3, there exists $\{\boldsymbol{\theta}_i\} \in \Theta_{\mathrm{col}}(K; \mathcal{A})$ such that*

$$\mathcal{E}(\{\boldsymbol{\theta}_i\}) \leq C_0 K^{-4/r_{\mathrm{int}}} \tag{32}$$

*for a constant $C_0$ independent of $K$.*

*Proof.* For each $i$, let $\Delta$ be the simplex in the triangulation containing $\boldsymbol{\zeta}_i$ with vertices $\{\boldsymbol{\zeta}_{j_0}, \ldots, \boldsymbol{\zeta}_{j_{r_{\mathrm{int}}}}\} \subseteq \{\boldsymbol{\zeta}_j\}_{j \in \mathcal{A}}$. Let $\{w_{ij_\ell}\}_{\ell=0}^{r_{\mathrm{int}}}$ be the barycentric coordinates of $\boldsymbol{\zeta}_i$ in $\Delta$, so $w_{ij_\ell} \geq 0$, $\sum_\ell w_{ij_\ell} = 1$, and $\boldsymbol{\zeta}_i = \sum_\ell w_{ij_\ell} \boldsymbol{\zeta}_{j_\ell}$. Define $\boldsymbol{\theta}_i := \sum_\ell w_{ij_\ell} \boldsymbol{\theta}_{j_\ell}^\star$, which is feasible in $\Theta_{\mathrm{col}}(K; \mathcal{A})$. By second-order Taylor expansion of $h$ at $\boldsymbol{\zeta}_i$,

$$h(\boldsymbol{\zeta}_{j_\ell}) = h(\boldsymbol{\zeta}_i) + J_h(\boldsymbol{\zeta}_i)(\boldsymbol{\zeta}_{j_\ell} - \boldsymbol{\zeta}_i) + \boldsymbol{R}_{i,\ell}, \quad \|\boldsymbol{R}_{i,\ell}\|_2 \leq \frac{M}{2} \|\boldsymbol{\zeta}_{j_\ell} - \boldsymbol{\zeta}_i\|_2^2, \tag{33}$$

where $J_h(\boldsymbol{\zeta}_i) \in \mathbb{R}^{d \times r_{\mathrm{int}}}$ is the Jacobian of $h$ at $\boldsymbol{\zeta}_i$. Weighting by $w_{ij_\ell}$ and summing, the first-order term cancels since $\sum_\ell w_{ij_\ell}(\boldsymbol{\zeta}_{j_\ell} - \boldsymbol{\zeta}_i) = \boldsymbol{0}$. Thus

$$\|\boldsymbol{\theta}_i^\star - \boldsymbol{\theta}_i\| = \left\| h(\boldsymbol{\zeta}_i) - \sum_\ell w_{ij_\ell} h(\boldsymbol{\zeta}_{j_\ell}) \right\| \leq \sum_\ell w_{ij_\ell} \|\boldsymbol{R}_{i,\ell}\|_2 \leq \frac{M}{2} \max_\ell \|\boldsymbol{\zeta}_{j_\ell} - \boldsymbol{\zeta}_i\|_2^2. \tag{34}$$

Shape-regularity gives $\max_\ell \|\boldsymbol{\zeta}_{j_\ell} - \boldsymbol{\zeta}_i\| \lesssim h$, hence $\|\boldsymbol{\theta}_i^\star - \boldsymbol{\theta}_i\| \lesssim h^2$ and $\|\boldsymbol{\theta}_i^\star - \boldsymbol{\theta}_i\|^2 \lesssim h^4$. Summing with $\sum_i q_i = 1$ yields $\mathcal{E} \lesssim h^4 \asymp K^{-4/r_{\mathrm{int}}}$. $\qquad\square$

*Proof of Theorem 3.1.* Dominance follows from Lemma A.2. For the rates, Lemma A.3 and Lemma A.4 bound $\inf \mathcal{E}$ for clustering and collaboration, respectively. Finally, convert distortion bounds to excess-risk bounds via Lemma A.1, which gives $\mathcal{R}_{\mathrm{cl}}^\star(K) \geq c K^{-2/r_{\mathrm{int}}}$ and $\mathcal{R}_{\mathrm{col}}^\star(K) \leq C K^{-4/r_{\mathrm{int}}}$ for constants $c, C > 0$ independent of $K$. $\qquad\square$

The clustered class considered here should be understood as a cluster-restricted personalization class under a fixed $K$-anchor budget. Hard assignment to one prototype is the clearest special case, but the comparison is not limited to a particular clustering algorithm. It also covers variants whose effective personalized models remain confined to cluster-restricted prototype sets. If a method allows unrestricted client-specific mixing across clusters, then it is no longer clustered in the sense of Theorem 3.1, but is closer to the collaborative class itself. Therefore, Theorem 3.1 compares cluster-restricted and collaboration-enabled hypothesis classes under the same anchor budget, rather than ruling out every possible soft or hybrid clustering implementation.

# B. Auxiliary lemmas and spectral derivations for Theorem 3.2

## B.1. Spectral analysis of collaboration

We fix a round $t$ and omit the superscript $t$ when no confusion arises. Let $\boldsymbol{\pi}$ denote its stationary distribution:

$$\boldsymbol{\pi}^\top \boldsymbol{W} = \boldsymbol{\pi}^\top, \quad \pi_i > 0, \quad \sum_{i=1}^N \pi_i = 1. \tag{35}$$

Let $\boldsymbol{D}_\pi \triangleq \mathrm{diag}(\pi_1, \ldots, \pi_N)$. For a stacked models vector $\boldsymbol{S} = [\boldsymbol{s}_1^\top, \ldots, \boldsymbol{s}_N^\top]^\top \in \mathbb{R}^{Nd}$, define the $\boldsymbol{\pi}$-weighted norm

$$\|\boldsymbol{S}\|_\pi^2 \triangleq \sum_{i=1}^N \pi_i \|\boldsymbol{s}_i\|_2^2 = \|(\boldsymbol{D}_\pi^{1/2} \otimes \boldsymbol{I}_d)\boldsymbol{S}\|_2^2. \tag{36}$$

This norm is equivalent to $\|\cdot\|_F$ up to constants depending only on $\pi_{\min} \triangleq \min_i \pi_i$ and $\pi_{\max} \triangleq \max_i \pi_i$:

$$\pi_{\min}\|\boldsymbol{S}\|_F^2 \leq \|\boldsymbol{S}\|_\pi^2 \leq \pi_{\max}\|\boldsymbol{S}\|_F^2. \tag{37}$$

Then, define the $\boldsymbol{\pi}$-weighted consensus projector $\boldsymbol{P}$ and the corresponding disagreement projector $\boldsymbol{M}$:

$$\boldsymbol{P} \triangleq (\mathbf{1}\boldsymbol{\pi}^\top) \otimes \boldsymbol{I}_d, \quad \boldsymbol{M} \triangleq \boldsymbol{I}_{Nd} - \boldsymbol{P}. \tag{38}$$

The disagreement subspace is

$$\mathcal{D} \triangleq \left\{ \boldsymbol{S} \in \mathbb{R}^{Nd} \; : \; (\boldsymbol{\pi}^\top \otimes \boldsymbol{I}_d)\boldsymbol{S} = \mathbf{0} \right\}. \tag{39}$$

For any $\boldsymbol{S}$, define $\boldsymbol{S}_{\mathrm{dis}} \triangleq \boldsymbol{M}\boldsymbol{S} \in \mathcal{D}$. Row-stochasticity implies $\boldsymbol{W}\mathbf{1} = \mathbf{1}$, and thus

$$(\boldsymbol{W} \otimes \boldsymbol{I}_d)\boldsymbol{P} = \boldsymbol{P}, \quad \boldsymbol{M}(\boldsymbol{W} \otimes \boldsymbol{I}_d)\boldsymbol{P} = \mathbf{0}. \tag{40}$$

Define the disagreement spectral contraction factor $\rho_W$, which controls the one-round contraction of disagreement under $\|\cdot\|_\pi$.

$$\widetilde{\boldsymbol{W}} \triangleq \boldsymbol{D}_\pi^{1/2}\boldsymbol{W}\boldsymbol{D}_\pi^{-1/2}, \quad \rho_W \triangleq \sigma_2(\widetilde{\boldsymbol{W}}), \tag{41}$$

where $\sigma_2(\cdot)$ is the second largest singular value.

**Lemma B.1** (Disagreement contraction of collaboration under $\|\cdot\|_\pi$). *For any $\boldsymbol{S} \in \mathcal{D}$,*

$$\|(\boldsymbol{W} \otimes \boldsymbol{I}_d)\boldsymbol{S}\|_\pi \leq \rho_W \|\boldsymbol{S}\|_\pi. \tag{42}$$

*Equivalently, $\|\boldsymbol{M}(\boldsymbol{W} \otimes \boldsymbol{I}_d)\boldsymbol{M}\|_{\pi \to \pi} \leq \rho_W$.*

*Proof.* Let $\boldsymbol{Z} \triangleq (\boldsymbol{D}_\pi^{1/2} \otimes \boldsymbol{I}_d)\boldsymbol{S}$. Then $\|\boldsymbol{S}\|_\pi = \|\boldsymbol{Z}\|_2$ by (36). Moreover, $\boldsymbol{S} \in \mathcal{D}$ implies $((\boldsymbol{D}_\pi^{1/2}\mathbf{1})^\top \otimes \boldsymbol{I}_d)\boldsymbol{Z} = (\boldsymbol{\pi}^\top \otimes \boldsymbol{I}_d)\boldsymbol{S} = \mathbf{0}$, so $\boldsymbol{Z}$ lies in the orthogonal complement of the top singular direction. Now,

$$\|(\boldsymbol{W} \otimes \boldsymbol{I}_d)\boldsymbol{S}\|_\pi = \|(\boldsymbol{D}_\pi^{1/2} \otimes \boldsymbol{I}_d)(\boldsymbol{W} \otimes \boldsymbol{I}_d)\boldsymbol{S}\|_2 = \|(\widetilde{\boldsymbol{W}} \otimes \boldsymbol{I}_d)\boldsymbol{Z}\|_2. \tag{43}$$

By the variational characterization of $\sigma_2$ on the orthogonal complement, $\|(\widetilde{\boldsymbol{W}} \otimes \boldsymbol{I}_d)\boldsymbol{Z}\|_2 \leq \sigma_2(\widetilde{\boldsymbol{W}})\|\boldsymbol{Z}\|_2 = \rho_W\|\boldsymbol{S}\|_\pi$. $\square$

**Lemma B.2** (Non-expansiveness of $\boldsymbol{M}$ in $\|\cdot\|_\pi$). *For all $\boldsymbol{S} \in \mathbb{R}^{Nd}$,*

$$\|\boldsymbol{M}\boldsymbol{S}\|_\pi \leq \|\boldsymbol{S}\|_\pi. \tag{44}$$

*Proof.* Write $\boldsymbol{S} = [\boldsymbol{s}_1^\top, \ldots, \boldsymbol{s}_N^\top]^\top$ and let $\bar{\boldsymbol{s}} \triangleq \sum_{i=1}^N \pi_i \boldsymbol{s}_i \in \mathbb{R}^d$. Then $(\boldsymbol{P}\boldsymbol{S})_i = \bar{\boldsymbol{s}}$ and $(\boldsymbol{M}\boldsymbol{S})_i = \boldsymbol{s}_i - \bar{\boldsymbol{s}}$. Hence,

$$\|\boldsymbol{M}\boldsymbol{S}\|_\pi^2 = \sum_i \pi_i \|\boldsymbol{s}_i - \bar{\boldsymbol{s}}\|_2^2 = \sum_i \pi_i \|\boldsymbol{s}_i\|_2^2 - \|\bar{\boldsymbol{s}}\|_2^2 \leq \sum_i \pi_i \|\boldsymbol{s}_i\|_2^2 = \|\boldsymbol{S}\|_\pi^2,$$

$\square$

## B.2. Spectral analysis of local training

Assume each local objective $L_i$ is $\mu$-strongly convex and $L$-smooth in a neighborhood of its minimizer $\boldsymbol{\theta}_i^\star$, twice continuously differentiable there, and let $\boldsymbol{H}_i \triangleq \nabla^2 L_i(\boldsymbol{\theta}_i^\star)$. A second-order expansion around $\boldsymbol{\theta}_i^\star$ gives

$$\nabla L_i(\boldsymbol{\theta}) \approx \boldsymbol{H}_i(\boldsymbol{\theta} - \boldsymbol{\theta}_i^\star), \quad \mu \boldsymbol{I} \preceq \boldsymbol{H}_i \preceq L\boldsymbol{I}. \tag{45}$$

One gradient step with step size $\eta$ satisfies

$$\boldsymbol{\theta}_i^+ = \boldsymbol{\theta}_i - \eta \nabla L_i(\boldsymbol{\theta}_i) \approx (\boldsymbol{I} - \eta \boldsymbol{H}_i)\boldsymbol{\theta}_i + \eta \boldsymbol{H}_i \boldsymbol{\theta}_i^\star. \tag{46}$$

After $K$ local steps,

$$\widetilde{\boldsymbol{\theta}}_i \approx \boldsymbol{G}_i \boldsymbol{\theta}_i + \boldsymbol{b}_i, \quad \boldsymbol{G}_i \triangleq (\boldsymbol{I} - \eta \boldsymbol{H}_i)^K, \quad \boldsymbol{b}_i \triangleq \left(\boldsymbol{I} - (\boldsymbol{I} - \eta \boldsymbol{H}_i)^K\right)\boldsymbol{\theta}_i^\star. \tag{47}$$

Stacking all clients yields

$$\widetilde{\boldsymbol{S}} \approx \boldsymbol{G}\boldsymbol{S} + \boldsymbol{b}, \quad \boldsymbol{G} \triangleq \mathrm{diag}(\boldsymbol{G}_1, \ldots, \boldsymbol{G}_N), \quad \boldsymbol{b} \triangleq [\boldsymbol{b}_1^\top, \ldots, \boldsymbol{b}_N^\top]^\top. \tag{48}$$

Moreover, if $0 < \eta \le \frac{2}{L+\mu}$, then for each $i$,

$$\|\boldsymbol{I} - \eta \boldsymbol{H}_i\|_2 \le \max\{\,|1 - \eta\mu|,\ |1 - \eta L|\,\} \le \frac{L - \mu}{L + \mu} =: \rho_1 < 1, \tag{49}$$

and hence

$$\|\boldsymbol{G}_i\|_2 = \|(\boldsymbol{I} - \eta \boldsymbol{H}_i)^K\|_2 \le \rho_1^K =: \rho_G < 1. \tag{50}$$

Because $\boldsymbol{G}$ is block-diagonal across clients, the $\pi$-weighted similarity transform leaves it unchanged: $(\boldsymbol{D}_\pi^{1/2} \otimes \boldsymbol{I}_d)\, \boldsymbol{G}\, (\boldsymbol{D}_\pi^{-1/2} \otimes \boldsymbol{I}_d) = \boldsymbol{G}$, and therefore its induced operator norm under $\|\cdot\|_\pi$ equals its Euclidean operator norm:

$$\|\boldsymbol{G}\|_{\pi \to \pi} = \|\boldsymbol{G}\|_2. \tag{51}$$

## B.3. Spectral analysis of one round operator

Combining collaboration and local training into a single linearized one-round map. At round $t$, define the effective linear operator

$$\mathcal{M}^t \triangleq (\boldsymbol{W}^t \otimes \boldsymbol{I}_d)\, \boldsymbol{G}. \tag{52}$$

To isolate the action on disagreement, consider the projected operator

$$\mathcal{T}^t \triangleq \boldsymbol{M}^t(\boldsymbol{W}^t \otimes \boldsymbol{I}_d)\boldsymbol{M}^t \boldsymbol{G}. \tag{53}$$

Using $\boldsymbol{M}^t$ on both sides removes the consensus component before mixing and after mixing; this is the operator that appears in the disagreement recursion. Then, define the one-round disagreement spectral radius

$$\rho_{\mathrm{dis}}^t \triangleq \rho\left(\mathcal{T}^t\big|_{\mathcal{D}^t}\right), \tag{54}$$

where $\rho(\cdot)$ is the spectral radius.

**Lemma B.3** (Bounds for the one-round disagreement operator). *For every round $t$,*

$$\|\mathcal{T}^t\|_{\pi^t \to \pi^t} \le \rho_W^t \rho_G, \quad \Rightarrow \quad \rho_{\mathrm{dis}}^t \le \rho_W^t \rho_G. \tag{55}$$

*Proof.* Fix $t$ and suppress the superscript. For any $\boldsymbol{S} \in \mathbb{R}^{Nd}$, by Lemma B.2 and (51),

$$\|\boldsymbol{M}\boldsymbol{G}\boldsymbol{S}\|_\pi \le \|\boldsymbol{G}\boldsymbol{S}\|_\pi \le \|\boldsymbol{G}\|_{\pi \to \pi} \|\boldsymbol{S}\|_\pi = \|\boldsymbol{G}\|_2 \|\boldsymbol{S}\|_\pi \le \rho_G \|\boldsymbol{S}\|_\pi. \tag{56}$$

Moreover, $\boldsymbol{M}\boldsymbol{G}\boldsymbol{S} \in \mathcal{D}$ by construction, hence Lemma B.1 yields

$$\|(\boldsymbol{W} \otimes \boldsymbol{I}_d)\boldsymbol{M}\boldsymbol{G}\boldsymbol{S}\|_\pi \le \rho_W \|\boldsymbol{M}\boldsymbol{G}\boldsymbol{S}\|_\pi. \tag{57}$$

According to Lemma B.2,

$$\|\mathcal{T}\boldsymbol{S}\|_\pi = \|\boldsymbol{M}(\boldsymbol{W} \otimes \boldsymbol{I}_d)\boldsymbol{M}\boldsymbol{G}\boldsymbol{S}\|_\pi \le \|(\boldsymbol{W} \otimes \boldsymbol{I}_d)\boldsymbol{M}\boldsymbol{G}\boldsymbol{S}\|_\pi \le \rho_W \|\boldsymbol{M}\boldsymbol{G}\boldsymbol{S}\|_\pi \le \rho_W \rho_G \|\boldsymbol{S}\|_\pi, \tag{58}$$

Taking the supremum over $\boldsymbol{S} \ne 0$ gives $\|\mathcal{T}\|_{\pi \to \pi} \le \rho_W \rho_G$, which implies (55). Finally, $\rho_{\mathrm{dis}} \le \|\mathcal{T}\|_{\pi \to \pi}$ for any induced norm, so $\rho_{\mathrm{dis}} \le \rho_W \rho_G$. $\qquad\square$

**B.4. Forcing bound: new disagreement injection scales with $H_\star$**

**Lemma B.4** (Forcing bound). *Let $\bar{\boldsymbol{\theta}}^\star \triangleq \frac{1}{N} \sum_{i=1}^N \boldsymbol{\theta}_i^\star$ and*

$$H_\star \triangleq \Big( \sum_{i=1}^N \|\boldsymbol{\theta}_i^\star - \bar{\boldsymbol{\theta}}^\star\|_2^2 \Big)^{1/2}. \tag{59}$$

*Let $\boldsymbol{Q} \triangleq \operatorname{diag}(\boldsymbol{Q}_1, \ldots, \boldsymbol{Q}_N)$ and $\boldsymbol{Q}_i \triangleq \boldsymbol{I} - (\boldsymbol{I} - \eta \boldsymbol{H}_i)^K$, so that $\boldsymbol{b} = \boldsymbol{Q}\boldsymbol{\Theta}^\star$ with $\boldsymbol{\Theta}^\star = [(\boldsymbol{\theta}_1^\star)^\top, \ldots, (\boldsymbol{\theta}_N^\star)^\top]^\top$. Define $\kappa_b \triangleq \|\boldsymbol{Q}\|_2$. Then for every round $t$,*

$$\big\| \boldsymbol{M}^t (\boldsymbol{W}^t \otimes \boldsymbol{I}_d)\, \boldsymbol{b} \big\|_{\pi^t} \leq \rho_W^t\, B_{\text{het}}, \tag{60}$$

*where $B_{\text{het}} \triangleq \kappa_b\, H_\star$.*

*Proof.* Fix a round $t$ and suppress superscripts. Decompose

$$\boldsymbol{\Theta}^\star = (\mathbf{1} \otimes \bar{\boldsymbol{\theta}}^\star) + \boldsymbol{\Theta}_c^\star, \quad \text{where} \quad \boldsymbol{\Theta}_c^\star = [(\boldsymbol{\theta}_1^\star - \bar{\boldsymbol{\theta}}^\star)^\top, \ldots, (\boldsymbol{\theta}_N^\star - \bar{\boldsymbol{\theta}}^\star)^\top]^\top \tag{61}$$

so that $\|\boldsymbol{\Theta}_c^\star\|_2 = H_\star$. Since $\boldsymbol{M}(\mathbf{1} \otimes v) = \mathbf{0}$ for any $v$,

$$\boldsymbol{M}\boldsymbol{b} = \boldsymbol{M}\boldsymbol{Q}\boldsymbol{\Theta}^\star = \boldsymbol{M}\boldsymbol{Q}\boldsymbol{\Theta}_c^\star, \quad \Rightarrow \quad \|\boldsymbol{M}\boldsymbol{b}\|_2 \leq \|\boldsymbol{Q}\|_2 \|\boldsymbol{\Theta}_c^\star\|_2 = \kappa_b H_\star. \tag{62}$$

Next, by (40), insert $\boldsymbol{M}$ on the right:

$$\boldsymbol{M}(\boldsymbol{W} \otimes \boldsymbol{I}_d)\boldsymbol{b} = \boldsymbol{M}(\boldsymbol{W} \otimes \boldsymbol{I}_d)\boldsymbol{M}\boldsymbol{b}. \tag{63}$$

Apply Lemma B.1 to $\boldsymbol{M}\boldsymbol{b} \in \mathcal{D}$ and then Lemma B.2:

$$\|\boldsymbol{M}(\boldsymbol{W} \otimes \boldsymbol{I}_d)\boldsymbol{M}\boldsymbol{b}\|_\pi \leq \|(\boldsymbol{W} \otimes \boldsymbol{I}_d)\boldsymbol{M}\boldsymbol{b}\|_\pi \leq \rho_W \|\boldsymbol{M}\boldsymbol{b}\|_\pi. \tag{64}$$

Finally, $\|\cdot\|_\pi \leq \|\cdot\|_2$ (since $\sum_i \pi_i = 1$), hence $\|\boldsymbol{M}\boldsymbol{b}\|_\pi \leq \|\boldsymbol{M}\boldsymbol{b}\|_2 \leq \kappa_b H_\star$, which gives (60). $\qquad \square$

## C. Proof of Theorem 3.2

*Proof.* Recall the linear approximation of one round of local training and collaborative aggregation, obtained by combining the local linearization (48) with the aggregation step (1):

$$\boldsymbol{S}^{t+1} \approx (\boldsymbol{W}^t \otimes \boldsymbol{I}_d)\,(\boldsymbol{G}\boldsymbol{S}^t + \boldsymbol{b}), \tag{65}$$

Let $\boldsymbol{\pi}^t$ be the stationary distribution of $\boldsymbol{W}^t$ and define $\boldsymbol{P}^t = (\mathbf{1}(\boldsymbol{\pi}^t)^\top) \otimes \boldsymbol{I}_d$, $\boldsymbol{M}^t = \boldsymbol{I}_{Nd} - \boldsymbol{P}^t$. Let $\boldsymbol{S}_{\text{dis}}^t \triangleq \boldsymbol{M}^t \boldsymbol{S}^t$ and $\Delta(t) \triangleq \|\boldsymbol{S}_{\text{dis}}^t\|_F^2$. We work with the $\|\cdot\|_{\pi^t}$ norm defined in (36).

**Step 1: project the update onto disagreement.** From $\boldsymbol{M}^t (\boldsymbol{W}^t \otimes \boldsymbol{I}_d)\boldsymbol{P}^t = \mathbf{0}$ ((40)), we obtain

$$\begin{aligned} \boldsymbol{S}_{\text{dis}}^{t+1} &= \boldsymbol{M}^t \boldsymbol{S}^{t+1} \\ &\approx \boldsymbol{M}^t (\boldsymbol{W}^t \otimes \boldsymbol{I}_d)\big(\boldsymbol{G}\boldsymbol{S}^t + \boldsymbol{b}\big) \\ &= \boldsymbol{M}^t (\boldsymbol{W}^t \otimes \boldsymbol{I}_d)\big(\boldsymbol{P}^t + \boldsymbol{M}^t\big)\big(\boldsymbol{G}\boldsymbol{S}^t + \boldsymbol{b}\big) \\ &= \boldsymbol{M}^t (\boldsymbol{W}^t \otimes \boldsymbol{I}_d)\boldsymbol{M}^t\big(\boldsymbol{G}\boldsymbol{S}^t + \boldsymbol{b}\big). \end{aligned} \tag{66}$$

Now decompose $\boldsymbol{S}^t = \boldsymbol{P}^t \boldsymbol{S}^t + \boldsymbol{M}^t \boldsymbol{S}^t$:

$$\begin{aligned} \boldsymbol{S}_{\text{dis}}^{t+1} &\approx \boldsymbol{M}^t (\boldsymbol{W}^t \otimes \boldsymbol{I}_d)\boldsymbol{M}^t \Big( \boldsymbol{G}\boldsymbol{M}^t \boldsymbol{S}^t + \boldsymbol{G}\boldsymbol{P}^t \boldsymbol{S}^t + \boldsymbol{b} \Big) \\ &= \underbrace{\boldsymbol{M}^t (\boldsymbol{W}^t \otimes \boldsymbol{I}_d)\boldsymbol{M}^t \boldsymbol{G}\boldsymbol{S}_{\text{dis}}^t}_{\text{historical disagreement}} + \underbrace{\boldsymbol{M}^t (\boldsymbol{W}^t \otimes \boldsymbol{I}_d)\boldsymbol{M}^t \boldsymbol{G}\boldsymbol{P}^t \boldsymbol{S}^t}_{\text{consensus-disagreement coupling}} + \underbrace{\boldsymbol{M}^t (\boldsymbol{W}^t \otimes \boldsymbol{I}_d)\boldsymbol{b}}_{\text{heterogeneity forcing}}. \end{aligned} \tag{67}$$

**Step 2: contraction of historical disagreement.** Define the one-round disagreement operator $\mathcal{T}^t = \boldsymbol{M}^t (\boldsymbol{W}^t \otimes \boldsymbol{I}_d)\boldsymbol{M}^t \boldsymbol{G}$ as in (53). Lemma B.3 gives

$$\|\mathcal{T}^t \boldsymbol{S}_{\text{dis}}^t\|_{\pi^t} \leq \rho_W^t \rho_G \|\boldsymbol{S}_{\text{dis}}^t\|_{\pi^t}. \tag{68}$$

**Step 3: control the consensus–disagreement coupling.** The term $\boldsymbol{G}\boldsymbol{P}^t\boldsymbol{S}^t$ is generally not in the consensus subspace unless all $\boldsymbol{G}_i$ coincide. Write $\boldsymbol{P}^t\boldsymbol{S}^t = \mathbf{1} \otimes \bar{\boldsymbol{\theta}}^t$ where $\bar{\boldsymbol{\theta}}^t \triangleq ((\boldsymbol{\pi}^t)^\top \otimes \boldsymbol{I}_d)\boldsymbol{S}^t$. Define the $\pi^t$-average local operator

$$\bar{\boldsymbol{G}}^t \triangleq \sum_{i=1}^{N} \pi_i^t \boldsymbol{G}_i, \tag{69}$$

and a curvature-heterogeneity coefficient

$$\delta_G^t \triangleq \left(\sum_{i=1}^{N} \pi_i^t \|\boldsymbol{G}_i - \bar{\boldsymbol{G}}^t\|_2^2\right)^{1/2}. \tag{70}$$

Then, by direct expansion,

$$\|\boldsymbol{M}^t\boldsymbol{G}\boldsymbol{P}^t\boldsymbol{S}^t\|_{\pi^t} = \left(\sum_{i=1}^{N} \pi_i^t \|(\boldsymbol{G}_i - \bar{\boldsymbol{G}}^t)\bar{\boldsymbol{\theta}}^t\|_2^2\right)^{1/2} \leq \delta_G^t \|\bar{\boldsymbol{\theta}}^t\|_2. \tag{71}$$

Applying Lemma B.1 and the non-expansiveness of $\boldsymbol{M}^t$ under $\|\cdot\|_{\pi^t}$ (Lemma B.2), we further obtain

$$\left\|\boldsymbol{M}^t(\boldsymbol{W}^t \otimes \boldsymbol{I}_d)\boldsymbol{M}^t\boldsymbol{G}\boldsymbol{P}^t\boldsymbol{S}^t\right\|_{\pi^t} \leq \rho_W^t \|\boldsymbol{M}^t\boldsymbol{G}\boldsymbol{P}^t\boldsymbol{S}^t\|_{\pi^t} \leq \rho_W^t \delta_G^t \|\bar{\boldsymbol{\theta}}^t\|_2. \tag{72}$$

In typical bounded-iterate regimes, it is standard to assume

$$\sup_t \|\bar{\boldsymbol{\theta}}^t\|_2 \leq \Theta_{\max} < \infty, \quad \sup_t \delta_G^t \leq \bar{\delta}_G < \infty, \tag{73}$$

in which case the coupling term contributes an additional uniformly bounded $\rho_W^t$ forcing:

$$\left\|\boldsymbol{M}^t(\boldsymbol{W}^t \otimes \boldsymbol{I}_d)\boldsymbol{M}^t\boldsymbol{G}\boldsymbol{P}^t\boldsymbol{S}^t\right\|_{\pi^t} \leq \rho_W^t \bar{\delta}_G \Theta_{\max}. \tag{74}$$

**Step 4: the forcing term is bounded by $\rho_W^t B_{\mathrm{het}}$ and scales with $H_\star$.** Lemma B.4 yields

$$\left\|\boldsymbol{M}^t(\boldsymbol{W}^t \otimes \boldsymbol{I}_d)\,\boldsymbol{b}\right\|_{\pi^t} \leq \rho_W^t \kappa_b H_\star =: \rho_W^t B_{\mathrm{het}}. \tag{75}$$

**Step 5: one-dimensional recursion with the coupling absorbed.** Combine (67), (68), (74), and (75). Let $x_t \triangleq \|\boldsymbol{S}_{\mathrm{dis}}^t\|_{\pi^t}$. Then

$$x_{t+1} \leq \rho_W^t \rho_G \, x_t + \rho_W^t \left(B_{\mathrm{het}} + \bar{\delta}_G \Theta_{\max}\right), \tag{76}$$

where $B_{\mathrm{het}}$ is the forcing scale from Step 4, i.e., $B_{\mathrm{het}} = \kappa_b H_\star$. For notational consistency, we absorb the consensus–disagreement coupling into a single forcing constant and redefine

$$B_{\mathrm{het}} \triangleq \kappa_b H_\star + \bar{\delta}_G \Theta_{\max}. \tag{77}$$

With (77), (76) becomes

$$x_{t+1} \leq (\rho_W^t \rho_G) \, x_t + \rho_W^t B_{\mathrm{het}}. \tag{78}$$

**Step 6: recursion.** Let $\Delta_\pi(t) \triangleq x_t^2$. Squaring (78) and using $(u + v)^2 \leq 2u^2 + 2v^2$ yields

$$\Delta_\pi(t + 1) \leq 2(\rho_W^t \rho_G)^2 \Delta_\pi(t) + 2(\rho_W^t)^2 B_{\mathrm{het}}^2. \tag{79}$$

If $\rho_W^t \leq \bar{\rho}_W$ for all $t$ and $\alpha \triangleq 2(\bar{\rho}_W \rho_G)^2 < 1$, then

$$\Delta_\pi(t) \leq \left(\sqrt{2}\,\bar{\rho}_W \rho_G\right)^{2t} \Delta_\pi(0) + \frac{2\,\bar{\rho}_W^2 B_{\mathrm{het}}^2}{1 - 2(\bar{\rho}_W \rho_G)^2}. \tag{80}$$

**Step 7: translate $\Delta_\pi(t)$ to $\Delta(t)$.** By norm equivalence (37),

$$\pi_{\min}^t \|\boldsymbol{S}\|_F^2 \leq \|\boldsymbol{S}\|_{\pi^t}^2 \leq \pi_{\max}^t \|\boldsymbol{S}\|_F^2. \tag{81}$$

Then,

$$\Delta(t) = \|\boldsymbol{S}_{\mathrm{dis}}^t\|_F^2 \leq \frac{1}{\pi_{\min}^t} \|\boldsymbol{S}_{\mathrm{dis}}^t\|_{\pi^t}^2 = \frac{1}{\pi_{\min}^t} \Delta_\pi(t), \quad \Delta_\pi(0) \leq \pi_{\max}^0 \Delta(0). \tag{82}$$

Absorbing $\sup_t(\pi_{\max}^t/\pi_{\min}^t)$ into a universal constant $C \geq 2$, we obtain

$$\Delta(t) \leq \left(\sqrt{2}\,\overline{\rho}_W \rho_G\right)^{2t} \Delta(0) + \frac{C\,\overline{\rho}_W^2\, B_{\mathrm{het}}^2}{1 - 2(\overline{\rho}_W \rho_G)^2}. \tag{83}$$

$\square$

### C.1. Remark on Theorem 3.2

The bound in Theorem 3.2 admits a clean transient–steady-state decomposition:

$$\Delta(t) \;\leq\; \underbrace{\left(\sqrt{2}\,\overline{\rho}_W \rho_G\right)^{2t} \Delta(0)}_{\text{transient decay}} \;+\; \underbrace{\frac{C\,\overline{\rho}_W^2}{1 - 2(\overline{\rho}_W \rho_G)^2}\, B_{\mathrm{het}}^2}_{\text{steady-state floor}}. \tag{84}$$

It shows that the coupled operator suppresses historical inter-client discrepancies at a geometric rate, but cannot remove the disagreement injected by heterogeneity.

**Transient term: contraction on disagreement.** Let $\boldsymbol{S}_{\mathrm{dis}}^t = \boldsymbol{M}^t \boldsymbol{S}^t$ denote the disagreement component under the $\pi^t$-weighted geometry. The proof derives a one-step recursion of the form

$$\|\boldsymbol{S}_{\mathrm{dis}}^{t+1}\|_{\pi^t} \;\lesssim\; \rho_W^t \rho_G \|\boldsymbol{S}_{\mathrm{dis}}^t\|_{\pi^t} \;+\; \rho_W^t\, B_{\mathrm{het}}, \tag{85}$$

so the effective per-round contraction factor on disagreement is $\rho_W^t \rho_G$. Where $\rho_W^t \in [0,1)$ is the mixing-induced contraction on non-consensus directions, and $\rho_G \in (0,1)$ is the training-induced contraction of the local linearized operator. Taking $\overline{\rho}_W = \sup_t \rho_W^t$ yields the geometric decay term $\left(\sqrt{2}\,\overline{\rho}_W \rho_G\right)^{2t} \Delta(0)$ in (84).

**Steady-state term: per-round disagreement injection.** Even when $\Delta(t)$ is small, fresh disagreement is injected each round through two mechanisms:

- *Heterogeneity forcing from mismatched optima.* In the linearized local update $\widetilde{\boldsymbol{S}}^t \approx \boldsymbol{G}\boldsymbol{S}^t + \boldsymbol{b}$, the bias $\boldsymbol{b} = \boldsymbol{Q}\boldsymbol{\Theta}^\star$ is driven by the dispersion of client optima. Lemma B.4 yields

$$\left\|\boldsymbol{M}^t(\boldsymbol{W}^t \otimes \boldsymbol{I}_d)\boldsymbol{b}\right\|_{\pi^t} \;\leq\; \rho_W^t\, \kappa_b\, H_\star, \quad \kappa_b \triangleq \|\boldsymbol{Q}\|_2, \quad H_\star \triangleq \left(\sum_{i=1}^N \|\boldsymbol{\theta}_i^\star - \overline{\boldsymbol{\theta}}^\star\|_2^2\right)^{1/2}. \tag{86}$$

- *Consensus–disagreement coupling from non-identical local dynamics.* When $\boldsymbol{G}_i$ differ across clients, $\boldsymbol{G}\boldsymbol{P}^t \boldsymbol{S}^t$ is not purely consensual and can leak into the disagreement subspace. With $\delta_G^t \triangleq \left(\sum_i \pi_i^t \|\boldsymbol{G}_i - \bar{\boldsymbol{G}}^t\|_2^2\right)^{1/2}$ and bounded iterates $\sup_t \|\overline{\boldsymbol{\theta}}^t\|_2 \leq \Theta_{\max}$, the coupling contributes an additional per-round injection of order $\rho_W^t\, \delta_G^t \Theta_{\max}$.

Accordingly, throughout Theorem 3.2 we use a single forcing scale

$$B_{\mathrm{het}} \;\triangleq\; \kappa_b\, H_\star + \overline{\delta}_G\, \Theta_{\max}, \quad \overline{\delta}_G \triangleq \sup_t \delta_G^t, \tag{87}$$

which upper-bounds the total disagreement injected per round after mixing and projection.

Smaller $\overline{\rho}_W$ accelerates transient decay and reduces the steady-state floor via the multiplicative factor $\overline{\rho}_W^2$, hence it drives the dynamics toward an FL-like regime. In the rank-one consensus case ($\rho_W^t = 0$), disagreement vanishes after one aggregation step. Larger heterogeneity (larger $H_\star$) raises the irreducible disagreement floor through the term $\kappa_b H_\star$. This term does not change the contraction rate, but sets the asymptotic level of personalization differences.

The constant $C$ absorbs (i) norm-equivalence factors between $\|\cdot\|_{\pi^t}$ and $\|\cdot\|_F$ and (ii) mild variability of $\{\pi^t\}$ across rounds when translating the $\pi^t$-weighted recursion to the Frobenius energy $\Delta(t) = \|\boldsymbol{S}_{\mathrm{dis}}^t\|_F^2$.

# D. Proof of Theorem 3.3

Use the linearized one-round model according to (48) and (1):

$$S^{t+1} = (W^t \otimes I_d)(GS^t + b), \quad \|G\|_2 \le \rho_G < 1. \tag{88}$$

We work under the same local regime as Theorem 3.2: $\|\cdot\|_{\pi^t}$ denotes the $\pi^t$-weighted norm (36), where $\pi^t$ is the stationary distribution of $W^t$, and we use non-expansiveness of $\pi^t$-weighted (conditional expectation) projectors as in Lemma B.2.

Fix $\tau > 0$. For each round $t$, define the directed threshold graph $\mathcal{G}_\tau^t = ([N], \mathcal{E}_\tau^t)$ with edges $(i,j) \in \mathcal{E}_\tau^t$ iff $W_{ij}^t \ge \tau$, and let $\mathcal{C}^t = \{C_1^t, \dots, C_{K_t}^t\}$ be the partition of $[N]$ induced by the strongly connected components of $\mathcal{G}_\tau^t$. Define the cross-block leakage

$$\varepsilon_t \triangleq \max_{k \in [K_t]} \max_{i \in C_k^t} \sum_{j \notin C_k^t} W_{ij}^t. \tag{89}$$

The theorem assumes that the similarity rule yields an asymptotically block-decoupled Markov kernel: there exists a time $t_0$ such that the total probability mass that any client assigns outside its own block, $\varepsilon_t \triangleq \max_k \max_{i \in C_k^t} \sum_{j \notin C_k^t} W_{ij}^t$, vanishes as $t \to \infty$, i.e., $\varepsilon_t \to 0$, while the row-normalized within-block restrictions remain uniformly mixing, with within-block contraction bounded by $\rho_{\text{in}}^t \le \bar\rho_{\text{in}} < 1$ for all $t \ge t_0$. For notational simplicity, we further assume that after $t_0$ the induced partition no longer changes, i.e., $\mathcal{C}^t \equiv \mathcal{C} = \{C_1, \dots, C_K\}$ and $K_t \equiv K$ for all $t \ge t_0$.

**Step 1: sufficient conditions for vanishing cross-block leakage.** We now give an explicit sufficient condition showing why similarity rules typically imply $\varepsilon_t \to 0$.

**Lemma D.1** (Kernel ratio bound for leakage). *Suppose $\Phi$ is constructed as follows: for each round $t$, define unnormalized affinities $a_{ij}^t = \varphi(d_{ij}^t)$ with a non-increasing kernel $\varphi : \mathbb{R}_+ \to \mathbb{R}_+$ and dissimilarity $d_{ij}^t = d(\boldsymbol{\theta}_i^t, \boldsymbol{\theta}_j^t) \ge 0$. Apply a sparsification operator $\mathcal{S}$ that never increases entries (i.e., $(\mathcal{S}(A))_{ij} \le A_{ij}$), and then row-normalize:*

$$\widehat{a}_{ij}^t \triangleq (\mathcal{S}(A^t))_{ij}, \qquad W_{ij}^t \triangleq \frac{\widehat{a}_{ij}^t}{\sum_{\ell=1}^N \widehat{a}_{i\ell}^t}. \tag{90}$$

*Then for any partition $\mathcal{C}$ and any $t$,*

$$\varepsilon_t \le \max_k \max_{i \in C_k} \frac{\sum_{j \notin C_k} \varphi(d_{ij}^t)}{\sum_{\ell=1}^N \varphi(d_{i\ell}^t)}. \tag{91}$$

*In particular, suppose that the partition exhibits a clear separation at round $t$: the within-block diameter is at most $\delta_t$ while the cross-block separation is at least $\Delta_t$, i.e.,*

$$\max_k \max_{i,\ell \in C_k} d_{i\ell}^t \le \delta_t, \quad \min_{k \ne k'} \min_{i \in C_k, \, j \in C_{k'}} d_{ij}^t \ge \Delta_t. \tag{92}$$

*Then*

$$\varepsilon_t \le (N-1) \frac{\varphi(\Delta_t)}{\varphi(\delta_t)}. \tag{93}$$

*Hence, whenever $\varphi(\Delta_t)/\varphi(\delta_t) \to 0$ (e.g. Gaussian kernel with $\Delta_t \to \infty$, or hard thresholding/top-k that eventually removes cross-block edges), we have $\varepsilon_t \to 0$.*

*Proof.* Since sparsification never increases entries, for every $i$ and $j$, $W_{ij}^t \le \varphi(d_{ij}^t)/\sum_\ell \varphi(d_{i\ell}^t)$. Summing over $j \notin C_k$ yields (91). Under the separation condition, the denominator is at least $\varphi(\delta_t)$ for each $i \in C_k$, while each cross-block term is at most $\varphi(\Delta_t)$, giving (93). $\square$

Lemma D.1 makes precise the mechanism behind "similar clients mix more and become even more similar; dissimilar clients mix less and become disconnected": once cross-block distances grow, kernel-based weights yield $\varepsilon_t \to 0$.

**Step 2: near-reducibility drives the disagreement contraction factor toward one.** Recall the global disagreement contraction factor $\rho_W^t$ defined in (41).

**Lemma D.2** (Cross-block leakage forces $\rho_W$ to be close to 1)**.** *Fix a round $t$ and omit the superscript $t$ for notational convenience, i.e., $W \equiv W^t$, $\pi \equiv \pi^t$, and $D_\pi \equiv D_{\pi^t}$; assume that with respect to a partition $\mathcal{C} = \{C_1, \ldots, C_K\}$ the cross-block leakage satisfies $\max_k \max_{i \in C_k} \sum_{j \notin C_k} W_{ij} \leq \varepsilon$; define the $\pi$-time-reversal $W^* \triangleq D_\pi^{-1} W^\top D_\pi$ and the reversibilization $R \triangleq W^* W$. Then*

$$\sigma_2\left(D_\pi^{1/2} W D_\pi^{-1/2}\right)^2 = \lambda_2(R), \tag{94}$$

*and there exists a constant $c_\pi \geq 1$ depending only on $\pi_{\min} \triangleq \min_i \pi_i$ and $\pi_{\max} \triangleq \max_i \pi_i$ such that*

$$1 - \rho_W \leq c_\pi \varepsilon. \tag{95}$$

*Consequently, if $\varepsilon_t \to 0$ then $\rho_W^t \uparrow 1$.*

*Proof.* Let $\widetilde{W} \triangleq D_\pi^{1/2} W D_\pi^{-1/2}$. The nontrivial singular values of $\widetilde{W}$ are the square-roots of the nontrivial eigenvalues of $\widetilde{W}^\top \widetilde{W}$. Moreover,

$$\widetilde{W}^\top \widetilde{W} = D_\pi^{-1/2} W^\top D_\pi W D_\pi^{-1/2} \sim D_\pi^{-1} W^\top D_\pi W = W^* W = R, \tag{96}$$

which proves (94). Since $R$ is reversible with respect to $\pi$, we may bound its spectral gap via conductance.

For any index set $A \subseteq \{1, \ldots, N\}$, define $W_{iA} \triangleq \sum_{j \in A} W_{ij}$ and let $\mathbf{1}_A \in \{0, 1\}^N$ be its indicator vector. Using $D_\pi W^* = W^\top D_\pi$, the $\pi$-flow across the cut $(A, A^c)$ under $R$ admits the identity

$$Q_R(A, A^c) \triangleq \mathbf{1}_A^\top D_\pi R \mathbf{1}_{A^c} = (W \mathbf{1}_A)^\top D_\pi (W \mathbf{1}_{A^c}) = \sum_{i=1}^N \pi_i W_{iA} W_{iA^c}. \tag{97}$$

Now take $A = \mathcal{I}$ to be any nonempty union of blocks in $\mathcal{C}$ with $\pi(\mathcal{I}) \triangleq \sum_{i \in \mathcal{I}} \pi_i \leq 1/2$. Because $\mathcal{I}$ is a union of blocks and the leakage bound holds blockwise, we have $W_{i\mathcal{I}^c} \leq \varepsilon$ for all $i \in \mathcal{I}$ and $W_{i\mathcal{I}} \leq \varepsilon$ for all $i \in \mathcal{I}^c$. Hence for every $i$, $W_{i\mathcal{I}} W_{i\mathcal{I}^c} \leq \varepsilon$, and (97) yields $Q_R(\mathcal{I}, \mathcal{I}^c) \leq \varepsilon$. Therefore the conductance of $R$ satisfies

$$\Phi(R) \triangleq \min_{0 < \pi(A) \leq 1/2} \frac{Q_R(A, A^c)}{\pi(A)} \leq \frac{Q_R(\mathcal{I}, \mathcal{I}^c)}{\pi(\mathcal{I})} \leq \frac{\varepsilon}{\pi(\mathcal{I})} \leq \frac{\varepsilon}{\pi_{\min}}. \tag{98}$$

For reversible Markov kernels, a Cheeger-type inequality gives $1 - \lambda_2(R) \leq 2\Phi(R) \leq 2\varepsilon/\pi_{\min}$. Combining with (94) and using $1 - \rho_W \leq 1 - \rho_W^2$ yields

$$1 - \rho_W \leq 1 - \rho_W^2 = 1 - \lambda_2(R) \leq \frac{2}{\pi_{\min}} \varepsilon, \tag{99}$$

which proves (95) with $c_\pi \triangleq 2/\pi_{\min}$. $\qquad\square$

**Step 3: block-consensus geometry.** Fix the stabilized partition $\mathcal{C} = \{C_1, \ldots, C_K\}$ for $t \geq t_0$. Let $\pi^t(C_k) = \sum_{i \in C_k} \pi_i^t$ and define normalized block weights $\pi_{k,i}^t = \pi_i^t / \pi^t(C_k)$ for $i \in C_k$. Define the block-consensus projector $P_{\mathcal{C}}^t$ by

$$(P_{\mathcal{C}}^t S)_i = \sum_{j \in C_{c(i)}} \pi_{c(i), j}^t s_j, \quad M_{\mathcal{C}}^t = I_{Nd} - P_{\mathcal{C}}^t. \tag{100}$$

Let $\bar{\theta}_k^t = \sum_{j \in C_k} \pi_{k,j}^t \theta_j^t$ be the block representative. Define the within-block disagreement energy

$$\Delta_{\mathrm{in}}(t) \triangleq \|M_{\mathcal{C}}^t S^t\|_{\pi^t}^2 = \sum_{k=1}^K \sum_{i \in C_k} \pi_i^t \|\theta_i^t - \bar{\theta}_k^t\|_2^2. \tag{101}$$

As $P_{\mathcal{C}}^t$ is a conditional expectation under $\pi^t$, we have the non-expansiveness (block-level extension of Lemma B.2):

$$\|M_{\mathcal{C}}^t S\|_{\pi^t} \leq \|S\|_{\pi^t}, \quad \|P_{\mathcal{C}}^t S\|_{\pi^t} \leq \|S\|_{\pi^t}. \tag{102}$$

**Step 4: decomposition of $W^t$ into within-block mixing and cross-block leakage.** For all $t \geq t_0$, define the cross-block leakage weight of row $i$ by $\ell_i^t \triangleq \sum_{j \notin C_{c(i)}} W_{ij}^t$. so $\max_i \ell_i^t \leq \varepsilon_t$. Decompose

$$(\boldsymbol{W}_{\text{in}}^t)_{ij} = W_{ij}^t \mathbf{1}\{c(i) = c(j)\}, \quad (\boldsymbol{W}_{\text{out}}^t)_{ij} = W_{ij}^t \mathbf{1}\{c(i) \neq c(j)\}. \tag{103}$$

Each row of $\boldsymbol{W}_{\text{out}}^t$ sums to $\ell_i^t \leq \varepsilon_t$. Renormalize the in-block part to be row-stochastic:

$$\overline{W}_{ij}^t = \frac{W_{ij}^t}{1 - \ell_i^t} \mathbf{1}\{c(i) = c(j)\}, \quad \boldsymbol{W}_{\text{in}}^t = \boldsymbol{D}_{1-\ell^t} \overline{\boldsymbol{W}}^t, \quad \boldsymbol{D}_{1-\ell^t} = \text{diag}(1 - \ell_1^t, \ldots, 1 - \ell_N^t) \preceq \boldsymbol{I}. \tag{104}$$

Thus

$$\boldsymbol{W}^t = \boldsymbol{D}_{1-\ell^t} \overline{\boldsymbol{W}}^t + \boldsymbol{W}_{\text{out}}^t. \tag{105}$$

Let $\rho_{\text{in}}^t$ denote the worst within-block disagreement contraction factor of $\overline{\boldsymbol{W}}^t$, and assume $\sup_{t \geq t_0} \rho_{\text{in}}^t \leq \overline{\rho}_{\text{in}} < 1$ as in the theorem statement. Then for any block-disagreement vector $\boldsymbol{S}$ with $\boldsymbol{P}_{\mathcal{C}}^t \boldsymbol{S} = \boldsymbol{0}$, the block version of Lemma B.1 yields

$$\|(\overline{\boldsymbol{W}}^t \otimes \boldsymbol{I}_d)\boldsymbol{S}\|_{\pi^t} \leq \rho_{\text{in}}^t \|\boldsymbol{S}\|_{\pi^t}, \tag{106}$$

up to a norm-equivalence constant.

**Step 5: operator-norm bound for the leakage component.** Recall the leakage mass $\ell_i^t \triangleq \sum_{j \notin C_{c(i)}} W_{ij}^t$ and define $\varepsilon_t \triangleq \max_i \ell_i^t$. Then the leakage operator is uniformly small under the $\pi^t$-weighted geometry:

$$\left\|(\boldsymbol{W}_{\text{out}}^t \otimes \boldsymbol{I}_d)\right\|_{\pi^t \to \pi^t} \leq c_\pi \varepsilon_t, \quad c_\pi \triangleq \frac{1}{\sqrt{\pi_{\text{min}}^t}}, \tag{107}$$

where $\pi_{\text{min}}^t \triangleq \min_i \pi_i^t$. For any stacked vector $\boldsymbol{S} = [\boldsymbol{s}_1^\top, \ldots, \boldsymbol{s}_N^\top]^\top$, each block satisfies $\|((\boldsymbol{W}_{\text{out}}^t \otimes \boldsymbol{I}_d)\boldsymbol{S})_i\|_2 \leq \ell_i^t \max_j \|\boldsymbol{s}_j\|_2 \leq \varepsilon_t \max_j \|\boldsymbol{s}_j\|_2$, and the max block norm is controlled by the weighted norm via $\max_j \|\boldsymbol{s}_j\|_2 \leq \|\boldsymbol{S}\|_{\pi^t}/\sqrt{\pi_{\text{min}}^t}$. Therefore (107) holds. When $\sup_t(\pi_{\text{max}}^t/\pi_{\text{min}}^t) < \infty$, the constant $c_\pi$ is uniformly bounded across rounds.

**Step 6: bounded-iterate constants.** We work in the same local regime as in the proof of Theorem 3.2 and reuse the bounded-iterate notation. In particular, assume the stacked iterates are uniformly bounded:

$$\sup_{t \geq 0} \|\boldsymbol{S}^t\|_2 =: \Theta_{\text{max}} < \infty. \tag{108}$$

This is a standard local-regime assumption in linearized stability analyses. Consequently, using $\|\boldsymbol{G}\|_2 \leq \rho_G < 1$ in (88), we have the uniform bound on the mixed update:

$$\sup_{t \geq 0} \|\boldsymbol{G}\boldsymbol{S}^t + \boldsymbol{b}\|_2 \leq \rho_G \Theta_{\text{max}} + \|\boldsymbol{b}\|_2 =: B_{\text{mix}} < \infty. \tag{109}$$

**Step 7: main recursion on within-block disagreement.** Let $x_t \triangleq \|\boldsymbol{M}_{\mathcal{C}}^t \boldsymbol{S}^t\|_{\pi^t}$ so that $x_t^2 = \Delta_{\text{in}}(t)$. Apply $\boldsymbol{M}_{\mathcal{C}}^t$ to (88):

$$\boldsymbol{M}_{\mathcal{C}}^t \boldsymbol{S}^{t+1} = \boldsymbol{M}_{\mathcal{C}}^t (\boldsymbol{W}^t \otimes \boldsymbol{I}_d)(\boldsymbol{G}\boldsymbol{S}^t + \boldsymbol{b}). \tag{110}$$

Use decomposition (105) to split the right-hand side into an in-block part and a leakage part.

*(i) Historical within-block disagreement contraction.* Decompose $\boldsymbol{G}\boldsymbol{S}^t = \boldsymbol{G}\boldsymbol{M}_{\mathcal{C}}^t \boldsymbol{S}^t + \boldsymbol{G}\boldsymbol{P}_{\mathcal{C}}^t \boldsymbol{S}^t$. For the historical term, using $\boldsymbol{D}_{1-\ell^t} \preceq \boldsymbol{I}$, non-expansiveness (102), the block contraction (106), and the local-training contraction $\|\boldsymbol{G}\|_2 \leq \rho_G$ ((50)), we obtain

$$\left\|\boldsymbol{M}_{\mathcal{C}}^t (\boldsymbol{D}_{1-\ell^t} \overline{\boldsymbol{W}}^t \otimes \boldsymbol{I}_d)\boldsymbol{G}\boldsymbol{M}_{\mathcal{C}}^t \boldsymbol{S}^t\right\|_{\pi^t} \leq \rho_{\text{in}}^t \rho_G \, x_t. \tag{111}$$

*(ii) In-block forcing.* We control $\boldsymbol{G}\boldsymbol{P}_{\mathcal{C}}^t \boldsymbol{S}^t$ using the same coupling argument as in (70)–(71) (proof of Theorem 3.2). Accordingly, define the within-block analogue:

$$\delta_{\text{in}}^t \triangleq \max_{k \in [K]} \Big(\sum_{i \in C_k} \pi_{k,i}^t \|\boldsymbol{G}_i - \overline{\boldsymbol{G}}_k^t\|_2^2\Big)^{1/2}, \quad \overline{\boldsymbol{G}}_k^t \triangleq \sum_{i \in C_k} \pi_{k,i}^t \boldsymbol{G}_i. \tag{112}$$

By direct expansion (identical to (71) but block-restricted),

$$\|\boldsymbol{M}_{\mathcal{C}}^t \boldsymbol{G} \boldsymbol{P}_{\mathcal{C}}^t \boldsymbol{S}^t\|_{\pi^t} \leq \delta_{\mathrm{in}}^t \|\boldsymbol{P}_{\mathcal{C}}^t \boldsymbol{S}^t\|_{\pi^t} \leq \delta_{\mathrm{in}}^t \|\boldsymbol{S}^t\|_{\pi^t} \leq \delta_{\mathrm{in}}^t \Theta_{\max}. \tag{113}$$

For the bias term, we reuse Lemma B.4 to obtain a valid uniform bound $\kappa_b H_\star$ on the injected disagreement scale (in the same local regime). Moreover, the in-block forcing induced by $\boldsymbol{G} \boldsymbol{P}_{\mathcal{C}}^t \boldsymbol{S}^t$ is controlled by $\delta_{\mathrm{in}}^t$ together with the bounded-iterate constant $\Theta_{\max}$. Accordingly, define the (enlarged) forcing constant

$$B_{\mathrm{het}} \triangleq \kappa_b H_\star + \left(\sup_{t \geq t_0} \delta_{\mathrm{in}}^t\right) \Theta_{\max}, \tag{114}$$

so that (using again non-expansiveness and (106))

$$\left\|\boldsymbol{M}_{\mathcal{C}}^t (\boldsymbol{D}_{1-\ell^t} \overline{\boldsymbol{W}}^t \otimes \boldsymbol{I}_d) (\boldsymbol{G} \boldsymbol{P}_{\mathcal{C}}^t \boldsymbol{S}^t + \boldsymbol{b})\right\|_{\pi^t} \leq \rho_{\mathrm{in}}^t B_{\mathrm{het}}. \tag{115}$$

*(iii) Leakage perturbation.* By (107), non-expansiveness, and (109),

$$\left\|\boldsymbol{M}_{\mathcal{C}}^t (\boldsymbol{W}_{\mathrm{out}}^t \otimes \boldsymbol{I}_d) (\boldsymbol{G} \boldsymbol{S}^t + \boldsymbol{b})\right\|_{\pi^t} \leq \kappa_\pi \, \varepsilon_t \, B_{\mathrm{mix}}. \tag{116}$$

Combining (110), (111), (115), and (116) yields the desired 1D recursion for all $t \geq t_0$:

$$x_{t+1} \leq \rho_{\mathrm{in}}^t \rho_G \, x_t + \rho_{\mathrm{in}}^t B_{\mathrm{het}} + \kappa_\pi \, \varepsilon_t \, B_{\mathrm{mix}}. \tag{117}$$

**Step 8: recursion.** Recall $x_t \triangleq \|\boldsymbol{M}_{\mathcal{C}}^t \boldsymbol{S}^t\|_{\pi^t}$ so that $\Delta_{\mathrm{in}}(t) = x_t^2$. From (117), we have for all $t \geq t_0$,

$$x_{t+1} \leq \rho_{\mathrm{in}}^t \rho_G \, x_t + \rho_{\mathrm{in}}^t B_{\mathrm{het}} + \kappa_\pi \, \varepsilon_t \, B_{\mathrm{mix}}. \tag{118}$$

Squaring (118) and using $(u + v + w)^2 \leq 2u^2 + 4v^2 + 4w^2$ yields

$$\Delta_{\mathrm{in}}(t + 1) \leq 2(\rho_{\mathrm{in}}^t \rho_G)^2 \, \Delta_{\mathrm{in}}(t) + 4(\rho_{\mathrm{in}}^t)^2 \, B_{\mathrm{het}}^2 + 4\kappa_\pi^2 \, \varepsilon_t^2 \, B_{\mathrm{mix}}^2. \tag{119}$$

Using $\rho_{\mathrm{in}}^t \leq \overline{\rho}_{\mathrm{in}}$ for all $t \geq t_0$ and $\varepsilon_t^2 \leq \sup_{s \geq t_0} \varepsilon_s^2$, define

$$\alpha \triangleq 2(\overline{\rho}_{\mathrm{in}} \rho_G)^2 \in (0, 1). \tag{120}$$

Then (119) implies the uniform recursion, for all $t \geq t_0$,

$$\Delta_{\mathrm{in}}(t + 1) \leq \alpha \, \Delta_{\mathrm{in}}(t) + 4\overline{\rho}_{\mathrm{in}}^2 B_{\mathrm{het}}^2 + 4\kappa_\pi^2 B_{\mathrm{mix}}^2 \sup_{s \geq t_0} \varepsilon_s^2. \tag{121}$$

Unrolling (121) gives, for all $t \geq t_0$,

$$\Delta_{\mathrm{in}}(t) \leq \alpha^{t-t_0} \Delta_{\mathrm{in}}(t_0) + \frac{4\overline{\rho}_{\mathrm{in}}^2 B_{\mathrm{het}}^2}{1 - \alpha} + \frac{4\kappa_\pi^2 B_{\mathrm{mix}}^2}{1 - \alpha} \sup_{s \geq t_0} \varepsilon_s^2. \tag{122}$$

Noting that $\alpha^{t-t_0} = (\sqrt{2} \, \overline{\rho}_{\mathrm{in}} \rho_G)^{2(t-t_0)}$ and $1 - \alpha = 1 - 2(\overline{\rho}_{\mathrm{in}} \rho_G)^2$, we obtain

$$\Delta_{\mathrm{in}}(t) \leq (\sqrt{2} \, \overline{\rho}_{\mathrm{in}} \rho_G)^{2(t-t_0)} \Delta_{\mathrm{in}}(t_0) + \frac{4\overline{\rho}_{\mathrm{in}}^2 B_{\mathrm{het}}^2}{1 - 2(\overline{\rho}_{\mathrm{in}} \rho_G)^2} + C \sup_{s \geq t_0} \varepsilon_s^2, \tag{123}$$

where we define

$$C \triangleq \frac{4\kappa_\pi^2 B_{\mathrm{mix}}^2}{1 - 2(\overline{\rho}_{\mathrm{in}} \rho_G)^2}. \tag{124}$$

**Step 9: multi-point collapse.** By definition (101), $\Delta_{\text{in}}(t)$ is the $\pi^t$-weighted mean squared deviation of client models from the $K$ block representatives $\{\bar{\boldsymbol{\theta}}_k^t\}_{k=1}^K$:

$$\Delta_{\text{in}}(t) = \sum_{k=1}^{K} \sum_{i \in C_k} \pi_i^t \|\boldsymbol{\theta}_i^t - \bar{\boldsymbol{\theta}}_k^t\|_2^2. \tag{125}$$

Consequently, if $\varepsilon_t \to 0$ and the forcing term remains bounded, the bound in (123) implies that the within-block dispersion $\Delta_{\text{in}}(t)$ is uniformly small for large $t$ (up to a constant floor). Equivalently, for each block $C_k$, all client iterates $\{\boldsymbol{\theta}_i^t\}_{i \in C_k}$ concentrate in a neighborhood of the block representative $\bar{\boldsymbol{\theta}}_k^t$, where the neighborhood radius is controlled by $B_{\text{het}}$ and the residual leakage level. Meanwhile, the vanishing leakage $\varepsilon_t \to 0$ eliminates cross-block mass transport, so different block representatives remain decoupled. Therefore the dynamics exhibits a multi-point collapse and reduces to a $K$-prototype clustered FL. The case $K = 1$ recovers the one-point consensus collapse. This completes the proof of Theorem 3.3. $\square$

We emphasize that consensus or clustering is not inherently undesirable. If clients are genuinely homogeneous, a consensus solution can be appropriate; if clients naturally form a few well-separated groups, a clustered solution can also match the underlying data structure. In this paper, we regard them as degeneration modes only when they arise from optimization-induced collapse rather than from true client heterogeneity, especially under fine-grained or continuous heterogeneity where richer client-specific collaboration is needed.

# E. Experiment

## E.1. Implementation Details

All experiments are conducted on a single NVIDIA A800 GPU with 80GB of memory and CUDA support. The code is available at https://github.com/YinHonb/pFedCCG.

### E.1.1. IMAGE TASK

Our image classification experiments are conducted on the CIFAR-10 dataset (Krizhevsky et al., 2009), which contains 60,000 RGB images of size $32 \times 32$ spanning 10 object categories, with 50,000 images for training and 10,000 for testing.

We employ ResNet-10 (He et al., 2016) as the backbone network due to its lightweight architecture and favorable balance between classification accuracy and computational efficiency. Under the standard federated learning setting, the training data are distributed among 50 clients in a label-skewed non-IID fashion using a Dirichlet distribution. To examine different degrees of data heterogeneity, the Dirichlet concentration parameter is varied over $\beta \in 0.2, 0.5, 1$, where smaller values indicate more severe label distribution skew.

Local model optimization is carried out using stochastic gradient descent (SGD) with a learning rate of 0.01 and a batch size of 64. Each client performs 10 local training epochs per communication round, and the total number of communication rounds is fixed to 50. To alleviate overfitting, an $\ell_2$ regularization term with a weight decay coefficient of $1 \times 10^{-5}$ is applied during training. All experiments are conducted with a fixed random seed of 42 to ensure reproducibility. The collaboration-scheduling hyperparameters are fixed within each task and are not tuned separately for each heterogeneity level. For CIFAR-10, we use $\alpha_{\min} = 0.1$, $\alpha_{\max} = 0.4$, $t_s = 25$, and $\tau = 7$ across all heterogeneity settings.

For evaluation, the CIFAR-10 test set is partitioned among clients according to the same client-specific label distribution used for the training split. Model performance is reported as the mean and standard deviation of personalized classification accuracy over all clients on their corresponding local test sets.

### E.1.2. NLP TASK

For the natural language processing task, we conduct experiments on the Yahoo! News dataset introduced in (Zhang et al., 2015), which consists of documents organized into ten categories. To facilitate the experiments, we construct a balanced subset by randomly selecting 1,000 samples from each category to form the training set and 100 samples from each category to form the test set.

We adopt TextCNN as the base text classification model, which effectively captures local semantic patterns through convolutional filters of multiple kernel sizes. Each input word is represented by a 50-dimensional dense vector generated from the pre-trained GloVe embedding model, specifically the glove.6B.50d model, to encode semantic information.

Under the federated learning setting, the global training set is distributed among 50 clients and partitioned in a label-skewed non-IID manner according to a Dirichlet distribution. By varying the concentration parameter of the Dirichlet distribution among 0.2, 0.5, and 1, we systematically control the degree of label heterogeneity across clients, where smaller values correspond to more severe label imbalance. This partitioning strategy reflects realistic federated scenarios in which data distributions differ significantly among participants while maintaining the overall class diversity of the dataset. For Yahoo, we use $\alpha_{\min} = 0.3$, $\alpha_{\max} = 0.6$, $t_s = 25$, and $\tau = 7$ across all heterogeneity settings.

During training, each client performs five local epochs per communication round with a batch size of 64 and a learning rate of 0.01. The total number of communication rounds is fixed at 50, and the random seed is set to 42 to ensure reproducibility.

### E.1.3. RECOMMEND TASK

For recommendation experiments, we employ the MovieLens-1M dataset (Harper & Konstan, 2015), which contains approximately one million ratings from 6,040 users on 3,706 movies. Ratings with a value of 4 or higher are treated as positive interactions. Following a global temporal split, we construct a training set of approximately 800,000 ratings, among which about 480,000 are positive interactions, and a test set of approximately 200,000 ratings, including around 120,000 positive interactions.

To simulate federated learning scenarios, the training set is partitioned among 50 clients along the user dimension. The partitioning is performed in a label-skewed non-IID manner using a Dirichlet distribution over the primary genre of each movie. The Dirichlet concentration parameter $\beta$ is varied to control the degree of statistical heterogeneity across clients, with smaller values corresponding to more severe imbalance. Each client is assigned at least 200 training interactions, which ensures sufficient local data for model updates while reflecting realistic differences in data distributions among participants. For MovieLens, we use $\alpha_{\min} = 0.4$, $\alpha_{\max} = 0.6$, $t_s = 25$, and $\tau = 10$ across all heterogeneity settings.

We adopt the Neural Collaborative Filtering model as the base recommendation model. Each client performs five local training epochs per communication round using a batch size of 64 and a learning rate of 0.01, with a total of 50 communication rounds. Model performance is evaluated by computing the average Recall at 20 and Hit Ratio at 20 across all clients. This evaluation captures both the effectiveness of federated optimization and the ability of each client to learn personalized preferences.

### E.1.4. LLM-SFT TASK

We conduct supervised fine-tuning of large language models using the Aya dataset(Singh et al., 2024), a human-curated multilingual instruction-following corpus covering 65 languages. The dataset is expanded through templating and translation to enable comprehensive multilingual and cross-lingual evaluation, suitable for federated LLM training.

To simulate realistic language heterogeneity, the top 10 languages are selected based on sample frequency. For each language, 1000 tokenized samples are retained for training and 50 for testing, with oversampling compensating for tokenization loss. Clients are randomly assigned $k$ languages each, ensuring all languages appear in at least one client. For Aya, we use $\alpha_{\min} = 0.4$, $\alpha_{\max} = 0.6$, $t_s = 15$, and $\tau = 7$ across all heterogeneity settings.

Clients fine-tune a LoRA-adapted LLM (Qwen2.5-1.5B-Instruct), updating only LoRA parameters with rank 8, alpha 16, and dropout 0.05, while the backbone model remains frozen. Local training uses the AdamW optimizer with learning rate $5e - 5$, weight decay $1e - 5$, batch size 1, gradient accumulation 1, and gradient clipping with maximum norm 1.

### E.2. Baselines

We compare the proposed method with a set of representative baselines, including one local training method, two classical federated learning methods, two clustered federated learning methods, and seven personalized federated learning methods. These baselines are selected to systematically evaluate the effectiveness of different learning paradigms in handling data heterogeneity, client collaboration, and personalization.

The selected baselines are categorized as follows.

*Local training* serves as a non-collaborative lower bound, providing a reference for performance without any client interaction.

*Classical federated learning methods* include FedAvg-FT and FedProx-FT, which represent global aggregation strategies

with and without heterogeneity-aware regularization. These methods provide insight into standard FL performance under non-IID data conditions.

*Clustered federated learning methods* include CFL and FedGWC, which group clients with similar behaviors to mitigate negative transfer caused by data heterogeneity. These approaches aim to improve training efficiency and stability by leveraging natural client clusters.

*Personalized federated learning methods* include FedAMP, pFedGraph, FedCollab, GraphRx, CFLGP, pFedCS, and FedAGHN. These methods introduce client-specific collaboration or aggregation mechanisms to enhance personalized performance in non-IID settings, highlighting the potential benefits of tailored federated strategies.

**Local.** Local training optimizes models independently on each client using only local data. No information is shared across clients, serving as a lower bound for collaborative learning performance.

**FedAvg-FT(McMahan et al., 2017).** FedAvg-FT aggregates local model updates into a single global model through iterative averaging. The aggregated model is then fine-tuned locally to provide limited personalization for each client.

**FedProx-FT(Li et al., 2020).** FedProx-FT extends FedAvg by introducing a proximal regularization term to restrict local updates. This design improves training stability under statistical and system heterogeneity, followed by local fine-tuning.

**FedAMP(Huang et al., 2021).** FedAMP enables personalized federated learning via attention-based message passing among clients. Clients with similar data distributions are encouraged to collaborate more strongly during training.

**CFL(Sattler et al., 2020).** Clustered Federated Learning (CFL) partitions clients into clusters based on optimization similarity. Each cluster trains a shared model to reduce the impact of data heterogeneity across clients.

**pFedGraph(Ye et al., 2023).** pFedGraph explicitly learns a collaboration graph that models pairwise relationships among clients. Personalized models are obtained by aggregating information from connected clients in the learned graph.

**FedCollab(Bao et al., 2023).** FedCollab constructs non-overlapping collaboration coalitions according to data distribution similarity and data quantity. Clients primarily collaborate within their coalition to mitigate negative transfer.

**GraphRx(Wang et al., 2025).** GraphRx formulates client collaboration as a graph-based personalized federated learning problem. Both model parameters and collaboration weights are jointly optimized to balance generalization and personalization.

**FedGWC(Song et al., 2025).** FedGWC is a clustered federated learning approach that identifies homogeneous client groups using a Gaussian-weighted clustering strategy. Cluster-specific models are trained to improve robustness under heterogeneous data distributions.

**CFLGP(Kim et al., 2024).** CFLGP performs client clustering by periodically collecting and clustering client gradients. This gradient-based strategy enables reliable grouping without accessing local data.

**pFedCS(Wu et al., 2025)** pFedCS selects collaborating clients based on classifier similarity. A distance-constrained aggregation mechanism is employed to generate personalized models for local training.

**FedAGHN(Song et al., 2025).** FedAGHN employs attentive graph hypernetworks to dynamically model client-specific collaboration structures. Personalized models are generated by aggregating information over learned collaboration graphs.

### E.3. Additional Experimental Results

**Robustness to noisy affinity templates.** Since the affinity template $\Psi$ is used as a stable collaboration prior, we examine whether pFedCCG is sensitive to template perturbations. We inject noise into the affinity template with different total variation (TV) levels and report the personalized accuracy in Table 6. The results show that pFedCCG remains robust under moderate perturbations. As the TV level increases, the performance decreases gradually rather than collapsing abruptly.

*Table 6.* Robustness to noisy affinity templates on CIFAR-10. These experiments are conducted on CIFAR-10 with 50 clients under Dirichlet heterogeneity levels $\beta \in \{0.2, 0.5, 1\}$.

| Template $\Psi$ TV | $\beta = 0.2$ | $\beta = 0.5$ | $\beta = 1$ |
| --- | --- | --- | --- |
| 0 | 85.20 | 80.36 | 78.35 |
| 0.05 | 85.20 | 80.35 | 78.34 |
| 0.10 | 85.16 | 80.12 | 78.25 |
| 0.15 | 84.85 | 79.90 | 77.88 |
| 0.20 | 84.53 | 79.49 | 77.48 |
| 0.25 | 84.39 | 79.29 | 77.00 |
| 0.30 | 84.21 | 78.98 | 76.57 |
| Best baseline | 83.97 | 78.56 | 76.27 |

*Table 7.* Robustness to different prescribed stationary distribution variants on CIFAR-10. These experiments are conducted on CIFAR-10 with 50 clients under Dirichlet heterogeneity levels $\beta \in \{0.2, 0.5, 1\}$.

| $q$ variant | $\beta = 0.2$ | $\beta = 0.5$ | $\beta = 1$ |
| --- | --- | --- | --- |
| Permutation | 84.51 | 80.01 | 78.14 |
| Noise | 85.23 | 80.22 | 78.50 |
| w/o $q$ | 85.01 | 80.26 | 78.24 |
| Best baseline | 83.97 | 78.56 | 76.27 |

Even under relatively large perturbations, pFedCCG still outperforms the best baseline under all three heterogeneity settings. This indicates that pFedCCG does not require an exact or oracle-quality template; a reasonably informative and stable affinity prior is sufficient to maintain strong personalized performance.

**Robustness to stationary distribution variants.** We also evaluate the effect of changing the prescribed stationary distribution $q$ in the projection-based kernel construction. Specifically, we consider three variants: *Permutation*, which permutes the original client weights; *Noise*, which perturbs the original weights; and *w/o $q$*, which removes the prescribed data-proportion geometry. As shown in Table 7, pFedCCG consistently outperforms the best baseline under all tested variants. This suggests that the proposed projection-based collaboration geometry is not overly sensitive to the exact specification of $q$. Nevertheless, using the original $q$ remains principled because it aligns the stationary collaboration geometry with the $q$-weighted personalization objective.

**Sensitivity to Collaboration Scheduling Hyperparameters** We evaluate the sensitivity of pFedCCG to the scheduling hyperparameters in Eq. (19), including $\alpha_{\max}$, $\alpha_{\min}$, and $\tau$. The experiments are conducted on CIFAR-10 with 50 clients under Dirichlet heterogeneity levels $\beta \in \{0.2, 0.5, 1\}$. Unless otherwise specified, we vary one hyperparameter at a time while keeping the others fixed.

As shown in Table 8, pFedCCG remains stable across a broad range of scheduling configurations. The performance changes only mildly when $\alpha_{\max}$, $\alpha_{\min}$, or $\tau$ varies within a reasonable range. These results indicate that pFedCCG does not require per-heterogeneity tuning of the schedule hyperparameters, and that the proposed collaboration geometry control is robust to moderate hyperparameter variations.

**Sensitivity to Projection Accuracy** We further examine whether pFedCCG requires an accurately solved projection subproblem. In implementation, the projection accuracy is controlled by the number of alternating projection steps, the stopping tolerance, and the maximum number of PGD iterations. As shown in Table 9, relaxing the projection solver settings only causes marginal performance changes. This suggests that pFedCCG does not rely on a highly accurate inner solve, and the projection step can be implemented efficiently with relatively loose solver configurations.

**Geometry-Level Diagnostics** Finally, we provide direct geometry-level diagnostics to quantify collapse mitigation and expressivity preservation. We report the intra-cluster model distance at rounds 5, 10, and 15, the average number of selectable aggregation models per client, row diversity, and effective rank of the collaboration structure. Larger values

*Table 8.* Sensitivity to collaboration scheduling hyperparameters on CIFAR-10. We report the mean personalized accuracy across 50 clients under different Dirichlet heterogeneity levels.

<table>
<tr><th colspan="4">(a) Sensitivity to $\alpha_{max}$</th><th colspan="4">(b) Sensitivity to $\alpha_{min}$</th><th colspan="4">(c) Sensitivity to $\tau$</th></tr>
<tr><td>$\alpha_{max}$</td><td>$\beta = 0.2$</td><td>$\beta = 0.5$</td><td>$\beta = 1$</td><td>$\alpha_{min}$</td><td>$\beta = 0.2$</td><td>$\beta = 0.5$</td><td>$\beta = 1$</td><td>$\tau$</td><td>$\beta = 0.2$</td><td>$\beta = 0.5$</td><td>$\beta = 1$</td></tr>
<tr><td>0.2</td><td>85.20</td><td>80.23</td><td>78.15</td><td>0.0</td><td>84.92</td><td>80.19</td><td>78.31</td><td>2</td><td>85.21</td><td>80.30</td><td>78.34</td></tr>
<tr><td>0.3</td><td>85.21</td><td>80.33</td><td>78.48</td><td>0.1</td><td>85.20</td><td>80.36</td><td>78.35</td><td>5</td><td>85.22</td><td>80.35</td><td>78.36</td></tr>
<tr><td>0.4</td><td>85.20</td><td>80.36</td><td>78.35</td><td>0.2</td><td>85.24</td><td>80.45</td><td>78.43</td><td>7</td><td>85.20</td><td>80.36</td><td>78.35</td></tr>
<tr><td>0.5</td><td>85.22</td><td>80.29</td><td>78.28</td><td>0.3</td><td>85.16</td><td>80.30</td><td>78.45</td><td>10</td><td>85.25</td><td>80.42</td><td>78.41</td></tr>
<tr><td>0.6</td><td>85.12</td><td>80.25</td><td>78.19</td><td>0.4</td><td>85.12</td><td>80.21</td><td>78.41</td><td>15</td><td>85.34</td><td>80.56</td><td>78.44</td></tr>
<tr><td></td><td></td><td></td><td></td><td></td><td></td><td></td><td></td><td>20</td><td>85.42</td><td>80.45</td><td>78.41</td></tr>
</table>

*Table 9.* Sensitivity to projection accuracy on CIFAR-10. Solver setting denotes (projection steps, stopping tolerance, maximum PGD iterations).

| Solver setting | $\beta = 0.2$ | $\beta = 0.5$ | $\beta = 1$ |
|---|---|---|---|
| $(5, 10^{-10}, 2000)$ | 85.20 | 80.36 | 78.35 |
| $(1, 10^{-4}, 2000)$ | 85.20 | 80.32 | 78.35 |
| $(1, 10^{-3}, 2000)$ | 85.18 | 80.40 | 78.36 |
| $(1, 10^{-2}, 200)$ | 85.16 | 80.26 | 78.30 |

generally indicate weaker collapse, richer collaboration geometry, and stronger preservation of collaborative expressivity. As shown in Table 10, pFedCCG consistently maintains the largest intra-cluster distance, full support size, and the highest row diversity and effective rank, confirming that it better mitigates collapse while preserving richer collaboration patterns.

**Robustness to Local Learning Rates**   We evaluate the robustness of pFedCCG to local optimization hyperparameters by varying the local learning rate on CIFAR-10 with 50 clients. The experiments are conducted under the highly heterogeneous setting $\beta = 0.2$. As shown in Table 11, pFedCCG achieves the best performance under all learning rate settings. This suggests that the performance gain of pFedCCG is not tied to a specific local learning rate and remains stable under different local optimization configurations.

**Scalability with Different Numbers of Clients**   We further evaluate the scalability of pFedCCG by varying the number of clients. Specifically, we compare the performance under 20 and 100 clients on CIFAR-10 with Dirichlet heterogeneity levels $\beta \in \{0.2, 0.5, 1\}$. As shown in Table 12, pFedCCG consistently achieves the best performance across different federation scales and heterogeneity levels. These results demonstrate that pFedCCG remains effective when the number of clients changes, suggesting that the proposed controlled collaboration geometry is scalable in larger federations.

**2D Spatial Visualization of Model Representations**   To investigate the dynamic evolution of collaborative relationships among clients during model training, we employ Multidimensional Scaling (MDS) to project the model relation matrices of 20 clients at different communication rounds ($t \in 0, 1, 5, 10, 15$) into a 2D Euclidean space for intuitive visualization. All experiments are conducted on the CIFAR-10 dataset, where an extreme non-IID environment is constructed via Dirichlet distribution ($\beta = 0.2$) to simulate complex label-shift scenarios in practical cross-silo FL deployments.

Evolution trajectory analysis reveals that the proposed pFedCCG algorithm achieves prominent advantages in both structural representation and evolutionary stability. As shown in Figure 3, pFedCCG maintains a well-separated and discriminative distribution of client representations in the MDS space throughout the entire training process, effectively mitigating the model homogenization issue prevalent in conventional federated learning. In the early training stages, pFedCCG establishes collaborative topologies tailored to the local data characteristics of each client. With the progression of communication rounds, its topology exhibits high robustness: it preserves effective intra-cluster collaboration while decoupling tasks across distinct data distributions through rational spatial separation. This enables each client to capture global shared knowledge without sacrificing the precise modeling capability for local heterogeneous data.

FedAMP (f–j) and pFedGraph (k–o) suffer from pronounced "spatial collapse" in the late training stages, where all client representations rapidly converge toward the spatial center. This reflects the high risk of model homogenization caused by ineffective personalization constraints in these methods. FedCollab (p–t) presents an extremely sparse distribution

*Table 10.* Geometry-level diagnostics on CIFAR-10 with 20 clients. Dist@5/10/15 denote the intra-cluster model distance at different rounds. Selectable denotes the average number of selectable aggregation models per client. RowDiv and EffRank denote row diversity and effective rank, respectively.

| Method | Dist@5 | Dist@10 | Dist@15 | Selectable | RowDiv | EffRank |
|---|---|---|---|---|---|---|
| FedAMP | 0.84 | 0.75 | 0.51 | 7 | 0.22 | 1.93 |
| pFedGraph | 0.53 | 0.18 | 0.06 | 10 | 0.02 | 1.06 |
| GraphRx | 0.57 | 0.35 | 0.27 | 20 | 0.04 | 1.33 |
| pFedCCG | 1.47 | 1.59 | 1.65 | 20 | 0.35 | 2.53 |

*Table 11.* Sensitivity to local learning rates on CIFAR-10 with $\beta = 0.2$.

| Method | lr=0.05 | lr=0.01 | lr=0.005 |
|---|---|---|---|
| FedAvg-FT | 84.04 | 83.61 | 82.93 |
| FedProx-FT | 84.21 | 83.79 | 82.69 |
| FedAMP | 78.83 | 76.00 | 59.89 |
| CFL | 83.65 | 81.36 | 77.65 |
| pFedGraph | 83.48 | 82.64 | 80.04 |
| FedCollab | 82.45 | 82.48 | 80.65 |
| GraphRx | 83.92 | 83.39 | 80.85 |
| FedGWC | 84.11 | 83.88 | 82.64 |
| CFLGP | 82.74 | 82.68 | 82.04 |
| pFedCS | 83.28 | 82.95 | 80.67 |
| FedAGHN | 84.07 | 83.97 | 83.18 |
| pFedCCG | **84.55** | **85.20** | **85.52** |

in the same coordinate system, as the Euclidean distances between client representations exceed the visualization range, leading to incomplete display of partial client points. By comparison, the client representations of pFedCCG consistently converge stably and compactly within the core observation region, demonstrating superior numerical stability and spatial representation consistency under extreme non-IID conditions relative to other baseline algorithms.

The comparison is further extended to another set of baseline clustered FL algorithms (CFL, FedAHGN, FedCFLP, FedCS, and FedGWC) under the same experimental setting ($\beta = 0.2$), with their topological evolution visualized via MDS in Figure 4. The baseline methods generally exhibit either insufficient stability or poor discriminative power when handling strongly heterogeneous data. CFL (a–d) and FedGWC (q–t) show over-sparse and continuously expanding distributions, with Euclidean distances among client representations diverging rapidly as training proceeds. Such unconstrained dispersed structures hinder the formation of stable local collaborative consensus and can induce oscillations in later training stages.

FedAHGN (e–h) and FedCS (m–p) fall into an opposite extreme of overly cohesive distributions, where client representations are excessively compact and lack spatial discriminability. This indicates a failure to effectively decouple heterogeneous tasks across clients, which may cause personalized models to degenerate into quasi-global models with compromised local performance. FedCFLP (i–l), although relatively better distributed, exhibits blurred cluster boundaries and ambiguous convergence directions, resulting in suboptimal collaborative topology formation.

In contrast to the divergence, collapse, or disordered distributions observed in baseline algorithms, pFedCCG maintains clear cluster boundaries to ensure effective decoupling of heterogeneous tasks, while its stable and compact convergence trajectories demonstrate robustness and adaptability under extreme non-IID label-shift conditions.

To quantitatively evaluate the effectiveness of the proposed pFedCCG algorithm in heterogeneous environments, we analyzed the personalized test accuracy of 50 clients on the CIFAR-10 dataset. The experiments were conducted under an extreme Non-IID setting ($\beta = 0.2$) with 50 communication rounds. Figure 5 presents violin plots of the test accuracy for all compared algorithms, with mean (Mean) and standard deviation (Std) indicated.

The results show that pFedCCG achieves the best performance among all compared methods, with an average accuracy of 85.20%. This is significantly higher than the average accuracy of pFedGraph, which is 82.64%, and also exceeds that of the

*Table 12.* Scalability with different numbers of clients on CIFAR-10. We report the mean personalized accuracy under 20 and 100 clients.

| Method | 20@0.2 | 20@0.5 | 20@1 | 100@0.2 | 100@0.5 | 100@1 |
|---|---|---|---|---|---|---|
| FedAvg-FT | 87.26 | 85.37 | 84.04 | 81.43 | 73.73 | 68.91 |
| FedProx-FT | 87.30 | 85.19 | 83.81 | 81.49 | 73.63 | 68.80 |
| FedAMP | 74.84 | 67.53 | 61.92 | 77.47 | 68.14 | 59.77 |
| CFL | 81.13 | 77.17 | 73.38 | 80.97 | 72.97 | 67.66 |
| pFedGraph | 86.42 | 84.08 | 83.60 | 80.41 | 73.15 | 68.76 |
| FedCollab | 83.00 | 80.44 | 80.28 | 81.92 | 74.05 | 68.88 |
| GraphRx | 85.91 | 83.67 | 81.61 | 81.60 | 73.90 | 69.12 |
| FedGWC | 86.44 | 84.71 | 82.86 | 81.44 | 73.97 | 68.86 |
| CFLGP | 85.35 | 82.66 | 79.93 | 82.08 | 73.59 | 69.21 |
| pFedCS | 86.34 | 84.81 | 81.98 | 80.89 | 71.31 | 63.68 |
| FedAGHN | 87.32 | 85.22 | 83.60 | 81.92 | 74.16 | 68.41 |
| pFedCCG | 87.96 | 86.01 | 84.69 | 83.61 | 77.53 | 72.62 |

recently competitive FedAGHN, which reaches 83.97%. Under highly biased data distributions, pFedCCG effectively aligns semantically similar clients, enhancing knowledge transfer across clients. In addition, pFedCCG exhibits the lowest standard deviation ($\sigma = 6.94$) among all algorithms. Compared with FedAMP ($\sigma = 10.77$) and CFL ($\sigma = 8.90$), the distribution of pFedCCG is more concentrated, and the lower whisker of the box plot is noticeably higher, indicating its ability to suppress underperforming "tail clients" and significantly reduce performance disparities caused by data heterogeneity.

Under extreme non-IID conditions, pFedCCG simultaneously improves average performance and reduces variability. These results demonstrate that, by preserving client-specific features while fully leveraging cross-client collaboration, pFedCCG achieves optimal performance in both overall classification accuracy and client-level performance balance.

**Algorithm Sweet zone.** In Table 13, we present the performance of pFedCCG under various data distributions, ranging from highly heterogeneous to fully IID, including a moderately heterogeneous setting with a Non-IID label shift of 2. The experimental results demonstrate that pFedCCG exhibits outstanding adaptability across all scenarios and achieves the highest average accuracy in both heterogeneous and IID settings, highlighting its significant "algorithm sweet spot" advantage. When confronted with the typical strong heterogeneity of the CIFAR-10 dataset, pFedCCG attains the highest accuracy of 85.20%, significantly outperforming the collaborative benchmark methods FedAGHN at 83.97% and pFedGraph at 82.64%. This advantage remains stable as the degree of heterogeneity varies: even in the extremely shifted non-IID scenario, pFedCCG maintains an accuracy of 89.43%, while similar algorithms like FedAMP and CFL show a significant decline. The experiments indicate that the proposed collaborative strength scheduling and distribution-aware projection mechanism can adaptively adjust according to the degree of data heterogeneity, achieving the best balance between local personalization and global collaboration. Additionally, the results in the table show that pFedCCG not only improves the average performance but also effectively reduces the performance fluctuations among clients.

*Table 13.* Performance on Cifar10

| Setting | nonidskew | 0.05 | 0.1 | 0.2 | 0.5 | 1 | 2 | 5 | 10 | iid |
|---|---|---|---|---|---|---|---|---|---|---|
| Local | 89.76 (4.52) | 92.67 (8.80) | 86.77 (12.85) | 77.40 (10.33) | 65.97 (9.41) | 59.97 (7.43) | 55.82 (4.83) | 50.66 (3.22) | 48.51 (2.52) | 46.85 (1.34) |
| FedAvg-FT | 89.73 (4.44) | 93.73 (7.18) | 88.81 (10.89) | 83.61 (7.83) | 78.38 (5.76) | 76.20 (3.58) | 75.27 (2.60) | 70.69 (1.57) | 76.63 (2.16) | 77.40 (0.39) |
| FedProx-FT | 90.03 (4.42) | 93.81 (7.06) | 88.92 (10.78) | 83.79 (7.72) | 78.39 (5.76) | 76.27 (3.60) | 74.95 (2.71) | 75.92 (1.53) | 76.02 (1.22) | 76.78 (0.36) |
| FedAMP | 86.12 (5.80) | 91.81 (9.52) | 85.87 (13.15) | 76.00 (10.77) | 65.51 (8.37) | 59.83 (6.20) | 55.83 (4.27) | 51.17 (2.72) | 49.33 (2.13) | 48.32 (1.02) |
| CFL | 94.54 (4.04) | 93.43 (7.99) | 87.98 (11.86) | 81.36 (8.90) | 74.81 (6.84) | 68.07 (5.74) | 65.18 (3.82) | 61.75 (3.10) | 60.21 (1.95) | 61.98 (2.44) |
| pFedGraph | 94.58 (3.94) | 93.03 (8.45) | 87.77 (12.28) | 82.64 (8.42) | 77.32 (5.92) | 75.40 (4.17) | 74.33 (3.27) | 74.91 (2.37) | 76.06 (1.16) | 77.04 (0.35) |
| FedCollab | **94.60 (3.96)** | 93.03 (8.58) | 87.82 (12.08) | 82.48 (8.40) | 77.51 (5.85) | 76.21 (3.35) | 75.29 (2.65) | 76.22 (1.55) | 76.52 (1.15) | 77.38 (0.36) |
| GraphRx | 92.12 (3.86) | 93.98 (6.81) | 88.53 (11.22) | 83.39 (7.91) | 78.45 (5.73) | 76.14 (3.55) | 75.40 (2.73) | 76.08 (1.59) | 75.82 (1.16) | 76.68 (0.37) |
| FedGWC | 90.76 (4.57) | 93.60 (7.64) | 88.84 (10.96) | 83.88 (8.08) | 78.25 (5.58) | 75.73 (4.06) | 75.28 (2.77) | 75.41 (1.55) | 75.71 (1.19) | 76.68 (1.16) |
| CFLGP | 93.56 (3.44) | 93.90 (7.17) | 88.44 (11.16) | 82.68 (8.17) | 77.00 (6.40) | 74.38 (4.76) | 72.71 (3.77) | 73.63 (1.85) | 73.50 (1.26) | 74.26 (0.85) |
| pFedCS | 91.27 (4.51) | 94.05 (6.92) | 88.81 (11.15) | 82.95 (8.16) | 76.92 (6.51) | 72.03 (5.11) | 69.49 (3.35) | 76.02 (2.15) | **76.64 (1.82)** | 76.31 (0.51) |
| FedAGHN | 93.28 (4.15) | 93.22 (7.81) | 87.91 (11.73) | 83.97 (7.62) | 78.56 (5.78) | 75.39 (4.41) | 73.56 (2.91) | 71.45 (1.80) | 70.78 (1.62) | 70.59 (0.73) |
| pFedCCG | 94.52 (4.01) | **94.21 (6.83** | **89.43 (10.09)** | **85.20 (6.94)** | **80.36 (4.97)** | **78.35 (3.44)** | **77.00 (2.58)** | **76.95** (1.55) | 76.63 (1.05) | **77.40 (0.36)** |

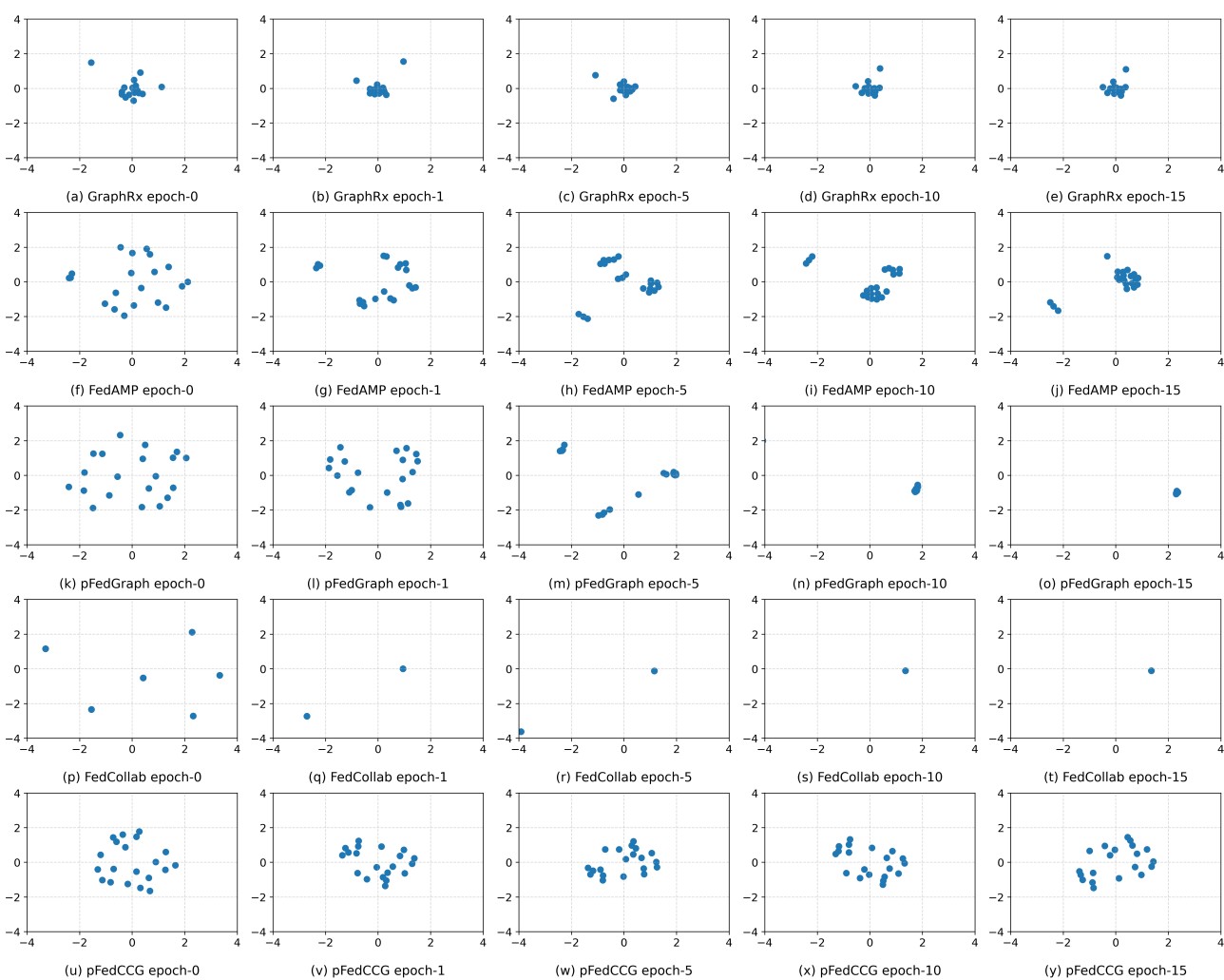

*Figure 3.* Euclidean Distance Scatter Plots of Client Representations: Evolution of GraphRx, FedAMP, pFedGraph, FedCollab and pFedCCG from Epoch 0 to 15

## F. Illustration

With the iteration of epochs, it will ultimately lead to two outcomes, as shown in Fig. 6: collapse to one point (Scenario 1 and 2) or collapse to multiple points (Scenario 3).

- Scenario 1: In cases of relatively low heterogeneity, all models will gradually collapse to one point.

- Scenario 2: In cases of moderate heterogeneity, the model will gradually cluster. Due to the existence of a "bridge model" between clusters, clusters are constantly approaching each other, resulting in multi hop connected nodes gradually becoming directly connected, ultimately leading to clusters constantly merging and collapsing to one point.

- Scenario 3: In cases of high heterogeneity, the model will gradually cluster and eventually collapse to multiple points.

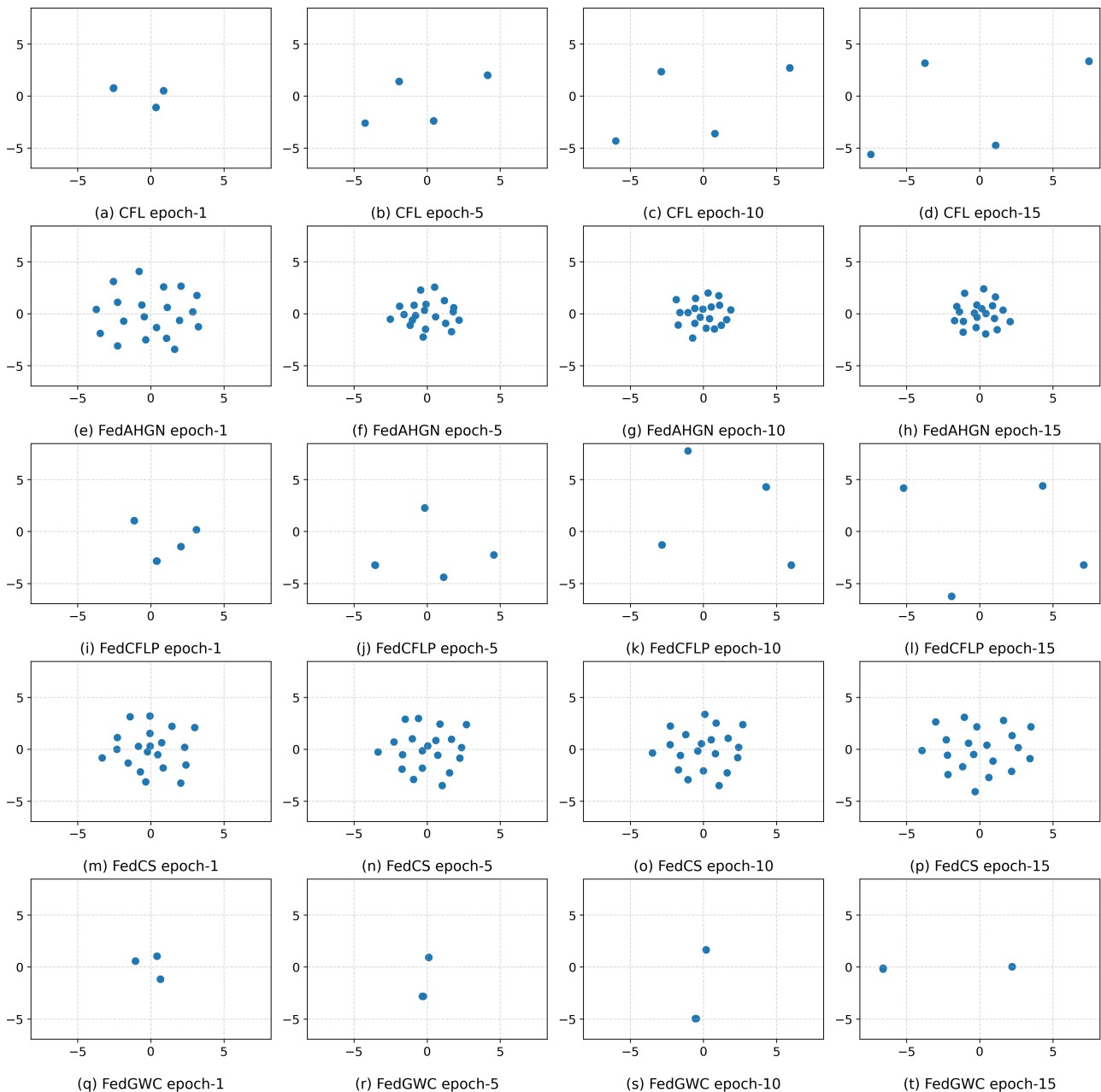

*Figure 4.* Euclidean Distance Scatter Plots of Client Representations: Evolution of CFL, FedAHGN, FedCFLP, FedCS and FedGWC from Epoch 1 to 15

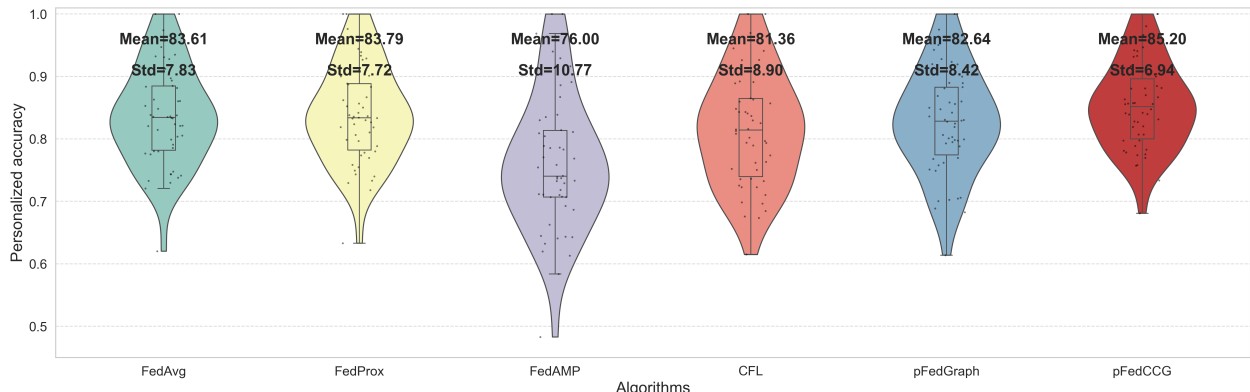

*(a)* Personalized Accuracy Distribution (FedCollab, GraphRx, FedGWC, CFLGP, pFedCS, FedAGHN and pFedCCG)

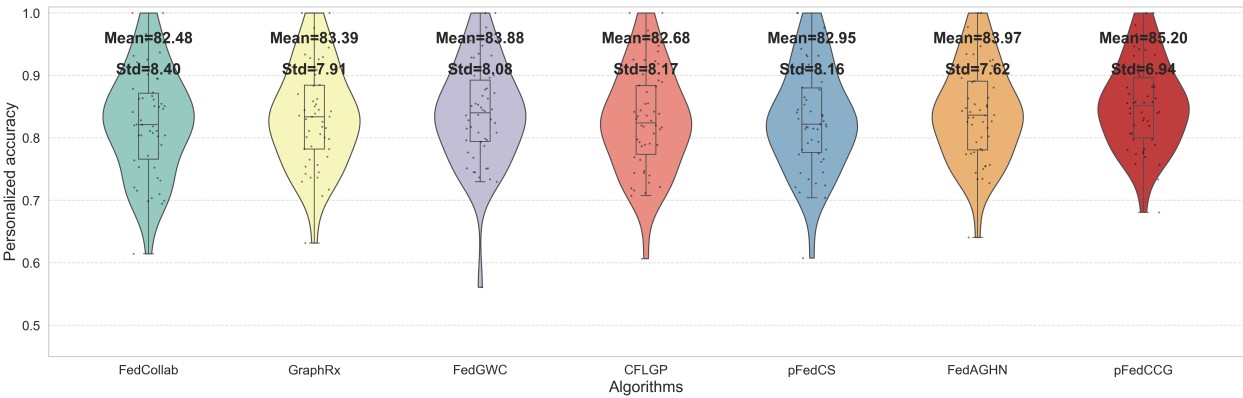

*(b)* Personalized Accuracy Distribution (FedAvg, FedProx, FedAMP, CFL, pFedGraph and pFedCCG)

*Figure 5.* Comparison of Personalized Accuracy Distributions Across Different FL Algorithms: Violin Plots Show Mean, Standard Deviation, and Distribution Density

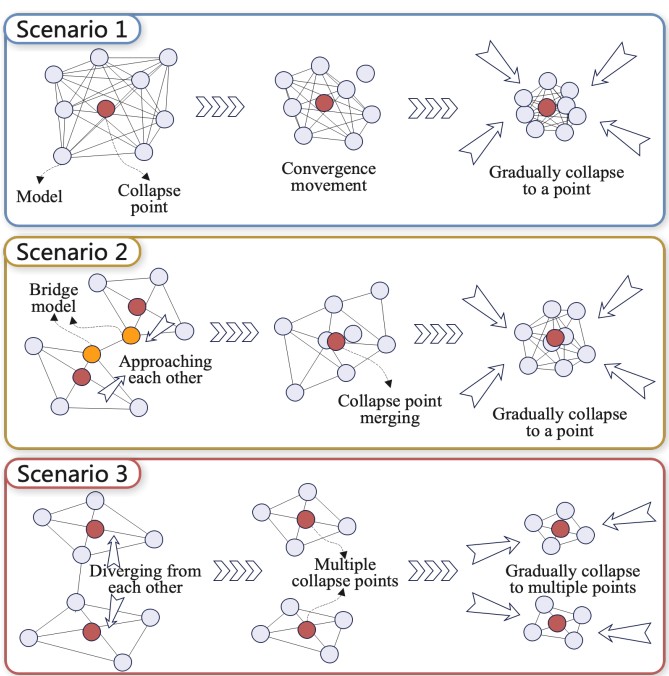

*Figure 6.* Similarity guided collapse dynamics in graph-based PFL: (Scenario 1) low-heterogeneity single-point collapse; (Scenario 2) bridge-induced cluster merging to one point; (Scenario 3) high-heterogeneity multi-point collapse.

