# OpenReview forum: "Controlled Collaboration Geometry for Personalized Federated Learning"
_ICML.cc/2026/Conference — ICML 2026 regular_

### Official Review · Reviewer_PgpC · 2026-03-03

**Soundness:** 2
**Presentation:** 2
**Significance:** 3
**Originality:** 3
**Overall Recommendation:** 4
**Confidence:** 3

**Summary:**

This paper studies the two main paradigms in personalized federated learning (PFL)—cluster-based and collaboration-based approaches—and theoretically shows that collaboration-based aggregation is more expressive than clustering under the same prototype budget (Theorem 3.1). At the same time, it identifies two failure modes that can arise from the coupled training-and-aggregation dynamics: (i) overly strong inter-client mixing can drive the system toward consensus, effectively degenerating into standard FL (Theorem 3.2), and (ii) updating the collaboration matrix via model-similarity feedback can trigger self-clustering, leading to degeneration into clustered FL (Theorem 3.3). Motivated by these analyses, the authors propose pFedCCG, which removes model-dependent feedback by using a static affinity template, enforces a stationary distribution aligned with clients’ data proportions, and constructs valid Markov collaboration kernels via projection. Across multiple tasks—including image classification, text classification, recommendation, and LLM supervised fine-tuning—under non-IID settings, pFedCCG consistently improves average personalization performance.

**Compliance With Llm Reviewing Policy:**

Affirmed.

**Ethical Review Concerns:**

I could not find an explicit Limitations section, even in the appendix. The paper includes Impact statement section; however, it is essentially non-informative and does not discuss concrete potential risks etc.

**Final Justification:**

Most of my concerns have been mitigated through the authors’ rebuttal, especially by the additional experimental results. The new evidence improves the empirical support for the paper’s main claims and strengthens the overall soundness of the work. Overall, the rebuttal positively changed my evaluation rather than merely reinforcing my prior assessment. Therefore, I am increasing my score from 3 to 4 (positive).

**Key Questions For Authors:**

Notation concern (overloaded $K$): It appears to overload the notation $K$, using it both for the number of anchors/prototypes and for a (symmetric) flow matrix.

**Limitations:**

I could not find an explicit Limitations section, even in the appendix. The paper includes an impact statement section; however, it is essentially non-informative and does not discuss concrete potential risks, etc.

**Strengths And Weaknesses:**

**Strengths**

(S1) Theoretical analysis of the expressivity of collaboration-based personalized FL (Theorem 3.1).

(S2) Theoretical analyses of two failure modes in coupled training-and-aggregation dynamics (Theorems 3.2 and 3.3).

(S3) Consistently strong empirical effectiveness of the proposed method (pFedCCG) across several tasks (Section 5).


**Weaknesses**

**(W1) Lack of theoretical evaluations of the proposed method’s effect on the failure modes and expressivity.**

I agree with the motivation and the overall design rationale: the method is clearly inspired by the failure modes identified in Section 3, and it is reasonable to expect that pFedCCG will make the collapse mechanisms less likely in practice. However, the paper does not clearly quantify how much the proposed design mitigates these failure modes, nor how well it preserves the expressivity advantage highlighted in Theorem 3.1.

**(W2) Potential performance degradation under misspecified affinity template $\Psi$.**

The proposed method fixes an affinity template $\Psi$ a priori based on Eq. (9). In Section 5, the authors instantiate $\Psi$ using the JS divergence of client data distributions and the cosine similarity of initial model updates. While these are plausible proxies, it is unclear whether they reliably reflect closeness between clients’ optimal solutions in general. This paper does not discuss the risk of performance degradation when $\Psi$ is misspecified or noisy.

**(W3) Missing experimental analyses.**

(a) Hyperparameter sensitivity. This paper seems to lack sensitivity studies for key hyperparameters, including: (i) local optimization hyperparameters (e.g., local learning rate), and (ii) pFedCCG-specific parameters such as $\alpha_{\min}, \alpha_{\max}, t_s, \tau$, etc.

(b) Robustness to partial participation and scalability. It would be important to evaluate robustness under partial participation and to study scaling behavior with different numbers of clients (e.g., 20, 50, 100).

(c) Computational overhead. The paper does not provide investigations of additional computation/runtime overhead introduced by the newly introduced projection step.

---

> ### Author Rebuttal · Authors · 2026-03-31
>
> Thank you for your insightful evaluation of our work. We address each point as follows:
>
> Weaknesses:
>
> **(W1)** We have added quantitative diagnostics on CIFAR-10 with 20 clients to measure both failure-mode mitigation and expressivity preservation. For failure modes, we report the intra-cluster model distance (Eq. (8)) at rounds 5/10/15 and the average number of selectable aggregation models per client. Larger values indicate weaker collapse and less self-clustering. For expressivity, we report the row diversity and effective rank of the within-cluster submatrix. Larger values indicate richer collaboration geometry.  The results show that pFedCCG consistently maintains the largest intra-cluster distance, full support size, and the highest row diversity/effective rank, which quantitatively confirms that it better mitigates collapse while preserving stronger expressivity than the competing methods.
>
> |Method|Dist@5|Dist@10|Dist@15|Selectable|RowDiv|EffRank|
> |---|---:|---:|---:|---:|---:|---:|
> |FedAMP|0.84|0.75|0.51|7|0.22|1.93|
> |pFedGraph|0.53|0.18|0.06|10|0.02|1.06|
> |GraphRx|0.57|0.35|0.27|20|0.04|1.33|
> |pFedCCG|**1.47**|**1.59**|**1.65**|**20**|**0.35**|**2.53**|
>
> **(W2)**
> To address this concern, we added additional robustness experiments by explicitly perturbing the similarity template with noise. We put the detailed results in the rebuttal for **Reviewer 2mhk (Weakness 3)**. The results show that pFedCCG is competitive under moderate noise. We will clarify this limitation and the corresponding robustness evidence in the revised version.
>
> **(W3)**
>
> (a) We added a sensitivity study on the **local learning rate** (CIFAR-10, 50 clients, $\beta=0.2$). The results show that pFedCCG remains strong under both settings and still achieves the best performance. This suggests that pFedCCG is not overly sensitive to the local optimization hyperparameter.
>
> |lr|0.05|0.005|
> |---|---:|---:|
> |FedAvg-FT|84.04|82.93|
> |FedProx-FT|84.21|82.69|
> |FedAMP|78.83|59.89|
> |CFL|83.65|77.65|
> |pFedGraph|83.48|80.04|
> |FedCollab|82.45|80.65|
> |GraphRx|83.92|80.85|
> |FedGWC|84.11|82.64|
> |CFLGP|82.74|82.04|
> |pFedCS|83.28|80.67|
> |FedAGHN|84.07|83.18|
> |pFedCCG|**84.55**|**85.52**|
>
> We also added an ablation on **pFedCCG-specific hyperparameters**. We put the detailed results in the rebuttal for **Reviewer rayy (Weakness 2)**.
>
>
> (b) We added experiments on both scalability and partial participation. The results show that pFedCCG remains the best method when scaling from **20 to 100 clients**, and it also achieves the best performance under **40%–60% client participation**. This suggests that pFedCCG is robust to reduced participation and scales well as the federation grows.
>
> **Scalability: different numbers of clients**
>
> |Method| 20@$\beta$0.2 |20@$\beta$0.5|20@$\beta$1|100@$\beta$0.2|100@$\beta$0.5|100@$\beta$1|
> |---|--------------:|---:|---:|---:|---:|---:|
> |FedAvg-FT|  87.26 |85.37|84.04|81.43|73.73|68.91|
> |FedProx-FT|  87.30 |85.19|83.81|81.49|73.63|68.80|
> |FedAMP|  74.84 |67.53|61.92|77.47|68.14|59.77|
> |CFL|  81.13 |77.17|73.38|80.97|72.97|67.66|
> |pFedGraph|  86.42 |84.08|83.60|80.41|73.15|68.76|
> |FedCollab|  83.00 |80.44|80.28|81.92|74.05|68.88|
> |GraphRx|  85.91 |83.67|81.61|81.60|73.90|69.12|
> |FedGWC|    86.44 |84.71|82.86|81.44|73.97|68.86|
> |CFLGP|    85.35 |82.66|79.93|82.08|73.59|69.21|
> |pFedCS|         86.34 |84.81|81.98|80.89|71.31|63.68|
> |FedAGHN|         87.32 |85.22|83.60|81.92|74.16|68.41|
> |pFedCCG|     **87.96** |**86.01**|**84.69**|**83.61**|**77.53**|**72.62**|
>
> **Partial participation** (40%–60% clients sampled each round, 100 clients, 50 rounds, $\beta=0.5$)
>
> |Method|Acc|
> |---|---:|
> |FedAvg-FT|72.69|
> |FedProx-FT|72.36|
> |FedAMP|64.87|
> |CFL|65.15|
> |pFedGraph|70.21|
> |FedCollab|61.82|
> |GraphRx|70.94|
> |FedGWC|69.92|
> |CFLGP|63.04|
> |pFedCS|68.46|
> |FedAGHN|69.49|
> |pFedCCG|**73.63**|
>
> (c) We added a runtime analysis of the **additional collaboration overhead** introduced by pFedCCG (excluding local training and model aggregation, Unit: s). The results show that the projection step incurs **relatively low overhead** compared with most baselines, especially against adaptive graph-based methods. We will include this analysis in the revised version.
>
> | Clients num |20|50|100|150|200|300|500|
> |----|---:|---:|---:|---:|---:|---:|---:|
> | FedAMP|0.10|0.61|2.42|5.42|9.71|21.77|60.16|
> | CFL|0.11|0.65|2.51|5.57|9.87|21.80|60.78|
> | pFedGraph |46.83|281.60|349.71|-|-|-|-|
> | FedCollab |1.23|43.21|697.20|-|-|-|-|
> | GraphRx |12.27|77.95|237.63|512.35|-|-|-|
> | FedGWC |0.01|0.02|0.16|0.25|0.45|1.10|3.05|
> | CFLGP |1.37|4.14|9.23|18.34|26.16|31.58|63.45|
> | pFedCS |0.11|0.47|1.21|2.59|4.28|9.52|27.93|
> | FedAGHN |106.14|121.05|163.65|239.91|332.83|-|-|
> | pFedCCG |0.02|0.04|0.15|1.23|5.27|7.58|13.34|
>
> **Key Questions:**
> We agree that this notation is overloaded and may cause confusion, and we will revise it in the final version to use distinct symbols.
>
> **Limitations:**
> We will add an explicit Limitations section in the final version,

---

> > ### Author Rebuttal · Reviewer_PgpC · 2026-04-03
> >
> > I appreciate your rebuttal. Most of my concerns have been resolved. I will improve my score.

---

> > > ### Author Response · Authors · 2026-04-05
> > >
> > > We sincerely appreciate your careful reading and constructive comments, which have helped improve the quality and clarity of the paper.

---

### Official Review · Reviewer_rayy · 2026-03-11

**Soundness:** 3
**Presentation:** 4
**Significance:** 3
**Originality:** 3
**Overall Recommendation:** 5
**Confidence:** 4

**Summary:**

This paper addresses a critical vulnerability in personalized federated learning, that training naturally drifts into one of two degenerate regimes - overly strong mixing leads to FL like global consensus and similarity driven feedback results in block consensus analogous to clustered FL. The authors argue that collaboration is mathematically more expressive than clustering and that controlled collaboration is needed to avoid the two degenerate regimes and learn a better personalized model. They thus propose pFedCCG that replaces similarity driven feedback with a static similarity indicator learned from client descriptors, and dynamically schedule the aggregation intensity over time.Their work is grounded in a rigorous mathematical analysis. Ultimately, their extensive empirical evaluations across image classification, NLP, recommendation, and LLM fine-tuning tasks demonstrate that pFedCCG consistently achieves superior average personalized accuracy while effectively preventing both consensus collapse and network fragmentation across diverse non-IID settings.

**Compliance With Llm Reviewing Policy:**

Affirmed.

**Final Justification:**

The rebuttal response (particularly the one with different proxies) completely addresses all of my concerns. The paper very accurately articulates an important dynamic in collaborative ML that is rarely talked about - that similarity based protocols lead to a degenerate mode analogous to clustered FL. The math is clean and intuitive and sets a good foundation for further work. Hence this paper deserves and acceptance.

**Key Questions For Authors:**

Static similarity indicators act as anchor that prevents nodes from drifting away as they mix, as they are pulled towards their original friends as captured in the prior. This seems to be an unfair advantage that the protocol has. One ablation experiment (cos variant) does make use of the initial model updates and fixes them and is claimed to have performance as good. Does it mean that using static similarity indicators, fixing them before the nodes have started mixing is an important principle to be followed for the personalization task?

**Limitations:**

-

**Strengths And Weaknesses:**

Strengths:
- The authors provide a highly articulate explanation of the coupled evolution of optimization and aggregation in Section 1. Identifying how this dynamic leads to the two degenerate regimes is a subtle but critical insight for self-organizing graphs. This establishes a compelling and clear motivation for their proposed control mechanisms.
- The theoretical framework excellently formalizes their intuition. Establishing that collaboration achieves a strictly faster approximation rate under continuous heterogeneity provides a solid mathematical justification for pursuing graph-based PFL over traditional clustered FL
- The empirical results show their protocol is not task specific and outperforms prior work across vision, nlp, recommendation, and large language tasks.
- Fig 2 is incredibly expressive and effectively communicates the results.
- Table 4 in Appendix E.3 shows they achieve best results even in iid cases

Weaknesses:
- The paper strongly argues against similarity driven feedback because it leads to self-clustering. However their solution relies on a client local descriptor to build a fixed affinity template. Sec 5.1 says that the JS divergence of data distribution and cosine similarity of initial model updates were used as descriptors. Utilizing global knowledge of the underlying data distributions in the main experiments to construct a static graph fundamentally violates the privacy constraints of federated learning. This gives the algorithm an oracle-like advantage over prior work.
- While the theoretical motivation to control the aggregation intensity is sound, the practical implementation relies on a highly parameterized surrogate. It is not described how fast the schedule must decay, how sensitive the algorithm's success is to these tuning choices - \alpha_{min}, \alpha_{max}, \tau, and how they are chosen.
- The primary results tables only report the mean and standard deviation of local personalized accuracies. It is important to compare the min accuracy to ensure that the protocol does not do injustice to any nodes while making the rich richer.

---

> ### Author Rebuttal · Authors · 2026-03-31
>
> Thank you for your valuable insights and suggestions, we address each point as follows:
>
> Weaknesses:
> 1. Thank you for this important comment. We agree that our method relies on a similarity template, but it does not require direct access to the true global data distributions. What is needed is only a pairwise similarity proxy, which can be obtained in a privacy-preserving manner without exposing raw local data, e.g., via secure similarity computation or by perturbing local statistics before similarity estimation.
> To further address this concern, we added robustness experiments with noisy affinity templates (see Rebuttal for **Reviewer 2mhk Key Questions 3**). The results show that pFedCCG remains competitive and still outperforms the strongest baseline under moderate perturbation. This suggests that the benefit of pFedCCG does not come from an oracle-like template, but from the controlled collaboration geometry itself, which is reasonably robust to imperfect or privacy-preserving similarity estimates. We will clarify this point and discuss privacy-preserving implementations in the final version.
>
> 2. We clarify that, to demonstrate the robustness of pFedCCG, we use **fixed schedule hyperparameters across different heterogeneity levels on the same dataset**, rather than tuning them separately for each run. For CIFAR-10, we use $\alpha_{\min}=0.1$, $\alpha_{\max}=0.4$, $t_s=25$, and $\tau=7$ for all reported heterogeneity settings. Here, $t_s$ is typically set to about half of the total communication rounds. We have also added a hyperparameter ablation on CIFAR-10. The results show that pFedCCG is **not sensitive to $\alpha_{\min}$, $\alpha_{\max}$, or $\tau$ within a reasonable range**. For the competing methods, we follow the hyperparameter settings in the original papers and official open-source implementations. More detailed tuning information will be added in the final version.
>
> **$\alpha_{\max}$ ablation** ($\alpha_{\min}=0.1,\tau=7$)
>
> |  $\alpha_{\max}$ |   0.2 |   0.5 |     1 |
> |-----------------:|------:|------:|------:|
> |              0.2 | 85.20 | 80.23 | 78.15 |
> |              0.3 | 85.21 | 80.33 | 78.48 |
> |              0.4 | 85.20 | 80.36 | 78.35 |
> |              0.5 | 85.22 | 80.29 | 78.28 |
> |              0.6 | 85.12 | 80.25 | 78.19 |
>
> **$\alpha_{\min}$ ablation** ($\alpha_{\max}=0.4,\tau=7$)
>
> | $\alpha_{\min}$ |   0.2 |  0.5 |1|
> |----------------:|------:|-----:|---:|
> |             0.0 | 84.92 |80.19 |78.31|
> |             0.1 | 85.20 |80.36 |78.35|
> |             0.2 | 85.24 |80.45 |78.43|
> |             0.3 | 85.16 |80.30 |78.45|
> |             0.4 | 85.12 |80.21 |78.41|
>
> **$\tau$ ablation** ($\alpha_{\max}=0.4,\alpha_{\min}=0.1$)
>
> |    $\tau$ |   0.2 |   0.5 |     1 |
> |----------:|------:|------:|------:|
> | 2 | 85.21 | 80.30 | 78.34 |
> | 5 | 85.22 | 80.35 | 78.36 |
> | 7 | 85.20 | 80.36 | 78.35 |
> | 10 | 85.25 | 80.42 | 78.41 |
> | 15 | 85.34 | 80.56 | 78.44 |
> | 20 | 85.42 | 80.45 | 78.41 |
>
> 3. We report the minimum personalized accuracy across clients. The results show that **pFedCCG achieves the best minimum accuracy under all heterogeneity levels ($\beta=0.2,0.5,1$) on CIFAR-10**, indicating that its gains are not obtained by improving only strong clients while hurting weaker ones.
>
> |Method|0.2|0.5|1|
> |---|---:|---:|---:|
> |FedAvg-FT|62.03|67.94|70.00|
> |FedProx-FT|63.31|67.78|71.14|
> |FedAMP|48.30|50.48|47.70|
> |CFL|61.49|56.65|59.46|
> |pFedGraph|61.39|64.48|66.39|
> |FedCollab|61.45|67.27|69.19|
> |GraphRx|63.17|68.43|70.09|
> |FedGWC|56.10|65.19|62.03|
> |CFLGP|60.67|64.78|66.19|
> |pFedCS|60.77|62.03|60.82|
> |FedAGHN|64.06|66.91|67.02|
> |pFedCCG|**68.09**|**69.54**|**71.44**|
>
> **Key Questions:**
> Thank you for this insightful comment. We agree that fixing the similarity indicator before mixing starts is an important design principle in our method, but its necessity depends on the heterogeneity regime. In extreme non-IID or pathological settings with clear natural cluster structure, where clients are already well separated into a few groups, controlled collaboration geometry may be less necessary, since a clustered or strongly localized collaboration pattern can already be appropriate. However, in more complex non-IID settings with fine-grained or continuous heterogeneity (e.g., Dirichlet-style heterogeneous partitions), the client relationships are not well described by a few rigid groups. In such cases, updating collaboration weights from current mixed models can create the harmful positive-feedback loop identified in our theory, causing self-clustering and loss of expressivity. From this perspective, the role of the static similarity indicator is not to provide an unfair advantage but to serve as an exogenous anchor that breaks this loop. Therefore, as also discussed in our rebuttal to Weakness 1, controlled collaboration geometry is not universally necessary in every scenario, but it is an important design choice for preventing personalization collapse in complex heterogeneous settings.

---

> > ### Author Rebuttal · Reviewer_rayy · 2026-04-04
> >
> > Thank you for showing that the performance is less sensitive to alpha and that the gains are present even in the minimum accuracy.
> > Regarding the pairwise similarity, perturbing the perfect template is not convincing enough. It would be more insightful to see what happens when the template is derived from a fundamentally weaker proxy (rather than a noisy version of the ideal data distribution as I believe a slight noise still largely preserves the macroscopic graph topology).
> > Further, there appears to be a logical tension. In your response - "This suggests that the benefit of pFedCCG does not come from an oracle-like template, but from the controlled collaboration geometry itself" - does this contradict the low sensitivity to alpha and tau? If the benefit does not come from the template, and the explicit control schedule has minimal impact, what exactly is driving the performance?

---

> > > ### Author Response · Authors · 2026-04-05
> > >
> > > Thank you for this helpful follow-up. To address this concern, we added a broader proxy study on CIFAR-10 using several qualitatively weaker template constructions, all under the same schedule setting $(\alpha_{\min}=0.1, \alpha_{\max}=0.4, t_s=25)$.
> > >
> > > Specifically, beyond the oracle-like **pFedCCG-JS** template built from the true local label distributions, we evaluated:
> > > (1) **cos-JS**, which uses cosine similarity of initial update directions as a proxy for local data characteristics and then computes JS similarity on those proxies;
> > > (2) **full-cos** and **full-cos-normalization**, which directly use cosine similarity of the full-model initial update directions;
> > > (3) **classifier-cos**, which only uses the classifier-layer initial update directions; and
> > > (4) **softmax-JS** and **softmax-cos**, which derive similarity from model softmax outputs on a small public anchor set of 256 samples.
> > >
> > > These proxies are fundamentally weaker than the true data-distribution JS template, since they only access coarse optimization or prediction behavior rather than the underlying client distributions themselves. The results are shown below:
> > >
> > > | Method | $\beta=0.2$ | $\beta=0.5$ | $\beta=1$ |
> > > |---|---:|---:|---:|
> > > | pFedCCG-JS | 85.20 | 80.36 | 78.35 |
> > > | cos-JS | 85.08 | 79.52 | 77.15 |
> > > | full-cos | 85.22 | 80.19 | 77.73 |
> > > | full-cos-normalization | 85.07 | 80.08 | 77.78 |
> > > | classifier-cos | 85.24 | 79.90 | 77.43 |
> > > | softmax-JS | 84.86 | 79.88 | 77.89 |
> > > | softmax-cos | 84.59 | 79.38 | 77.49 |
> > > | Best baseline | 83.97 | 78.56 | 76.27 |
> > >
> > > These results support two conclusions. First, pFedCCG does not rely on an oracle-quality template: even substantially weaker proxies remain consistently above the strongest baseline across all three heterogeneity levels. Second, proxy quality still matters: stronger proxies such as JS and full-cos generally perform better than weaker output-based proxies, especially when $\beta$ increases and client relationships become more intricate. Thus, the template is not irrelevant, but it only needs to provide a coarse and stable collaboration prior rather than an oracle-level estimate.
> > >
> > > We agree that our previous wording was too strong and may have created a logical tension. Our intended claim was not that the template is irrelevant, nor that the schedule is unimportant. Rather, the empirical picture is:
> > >
> > > (i) pFedCCG does not require an oracle-like template; a coarse but stable proxy is sufficient,
> > >
> > > (ii) the gain mainly comes from breaking the similarity-feedback loop that drives the degeneration in Theorem 3.3, and from converting that proxy into a valid, stable collaboration kernel through controlled geometry,
> > >
> > > (iii) the explicit schedule provides an additional but relatively modest refinement, which is why performance is not highly sensitive to $\alpha$ and $\tau$ within a reasonable range. In other words, low sensitivity to $\alpha$ and $\tau$ indicates robustness of the control mechanism, rather than the absence of a scheduling effect.

---

### Official Review · Reviewer_bvHB · 2026-03-11

**Soundness:** 3
**Presentation:** 2
**Significance:** 2
**Originality:** 3
**Overall Recommendation:** 4
**Confidence:** 4

**Summary:**

The paper provides a theoretical description of collaboration in personalized federated learning. The authors show that the best achievable risk for collaborative learning always exceeds the risk for clustered learning and provide lower and upper bounds, respectively. The paper presents a theoretical investigation of consensus collapse and multi-point collapse in collaborative algorithms. Motivated by the developed theory, the authors propose their own personalized algorithm and validate it against a wide range of baselines on image classification, NLP, recommendation, and LoRA fine-tuning, demonstrating that the proposed approach scales on various problems.

**Compliance With Llm Reviewing Policy:**

Affirmed.

**Final Justification:**

I still find the reported quality improvements somewhat questionable. However, given additional experiments provided in the rebuttal, I raise my score to 4.

**Key Questions For Authors:**

1) For the reported runs, could the authors provide information on the number of clients that experienced an improvement in performance? If personalized performance degraded for some clients, how severe was the worst-case drop?

2) Was $\alpha_{min}$ and $\alpha_{max}$ tuned for each level of heterogeneity?

3) Could the authors provide information on the computational overhead associated with the use of projected gradient descent at each iteration of the algorithm? How accurately does the subproblem need to be solved?

I would be glad to reconsider my evaluation if all concerns are addressed during the discussion.

**Limitations:**

Yes

**Strengths And Weaknesses:**

Strengths:

1) The paper provides both intuition and theoretical description of the phenomenon encountered by personalized algorithms in practice. This offers insights for designing new collaborative methods that are resilient to degradation.

2) Theoretical section is well-structured. I checked the proofs and they seem correct.

3) Experimental section covers multiple domains and includes a comprehensive set of baselines, encompassing both traditional federated and clustered/collaborative personalized approaches.

Weaknesses:

1) My major concern relates to the presentation of empirical results. Tables 1-4 report means and deviations of personalized qualities across clients. Since the intervals overlap substantially, it can be possible that some local models perform worse when trained with pFedCCG. Therefore, a comparison of the number of winning/losing clients should be conducted.

2) One of the main algorithmic insights of pFedCCG is the projection of the feedback-free template onto the Markov set with pre-described stationary distribution. The mixing strength depends on the magnitude of $\alpha_{t}$, with a schedule determined by hyperparameters $\alpha_{\min}$ and $\alpha_{\max}$. Since the presented results show that pFedCCG consistently outperforms the baselines across different levels of heterogeneity (which clearly require different mixing strengths according to the text), it can be assumed that $\alpha_{\min}$ and $\alpha_{\max}$ were tuned for every run. However, I did not find any mention of how these hyperparameters were selected. Furthermore, a numerical ablation study on tuning these hyperparameters would be useful for both researchers and practitioners. It should also be clarified how the competing methods were tuned.

3) At each iteration, pFedCCG requires solving a constrained convex optimization problem to compute the projection. In Line 288, the authors state that this procedure converges rapidly in practice. However, this claim should be supported with a numerical analysis of the computational overhead. Since competing methods do not utilize inner minimization, this could turn out to be a bottleneck of the proposed procedure.

4) Proofs of Theorems 3.2 and 3.3 are based on the linearization of local objectives around their optima. Consequently, there is a concern that the developed theory may only be applicable in the final stages of training. The authors should discuss whether this effect is observed from the very beginning of training.

5) In some places, the paper is difficult to read. Some notations introduced in Sections 3.3 and 3.4 are defined only in Appendix. Additionally, assumptions underlying the analysis are either stated in Appendix (e.g., local quadratic growth) or appear directly within the proofs (e.g., boundedness of the parameter set).

6) The paper contains some typos that affect the perception of the work. For example, in the TextCNN experiment with $\beta=0.2$, pFedCCG performed worse than GraphRx, contrary to what is marked by the authors.

---

> ### Author Rebuttal · Authors · 2026-03-31
>
> Thank you for your valuable feedback, which helps us improve the clarity and rigor of our work. We address each point as follows:
>
> Weaknesses:
> 1. We have added a client-level winning/losing comparison to complement the mean and deviation results (including ties). The new results show that pFedCCG is better than all baselines. This indicates that the improvement of pFedCCG is not driven by only a few clients, but is broadly shared across clients without causing observable local performance degradation.
>
> |Method|W@1|W@3|L@1|L@3|
> |---|---:|---:|---:|---:|
> |FedAvg-FT|4|15|0|1|
> |FedProx-FT|3|16|0|2|
> |FedAMP|2|2|46|48|
> |CFL|2|3|0|31|
> |pFedGraph|2|8|0|16|
> |FedCollab|2|7|2|18|
> |GraphRx|2|10|0|1|
> |FedGWC|10|27|0|1|
> |CFLGP|2|4|0|15|
> |pFedCS|2|5|0|11|
> |FedAGHN|4|25|0|0|
> |pFedCCG|37|46|0|0|
>
> 2. We clarify that, to demonstrate the robustness of pFedCCG, we use **fixed schedule hyperparameters across different heterogeneity levels on the same dataset**, rather than tuning them separately for each run. For CIFAR-10, we use $\alpha_{\min}=0.1$, $\alpha_{\max}=0.4$, $t_s=25$, and $\tau=7$ for all reported heterogeneity settings. Here, $t_s$ is typically set to about half of the total communication rounds. We have also added a hyperparameter ablation on CIFAR-10. The results show that pFedCCG is **not sensitive to $\alpha_{\min}$, $\alpha_{\max}$, or $\tau$ within a reasonable range**. For the competing methods, we follow the hyperparameter settings in the original papers and official open-source implementations. More detailed tuning information will be added in the final version.
>
> Due to character limitations, please refer to the rebuttal for **Reviewer rayy (Weakness 2)** for ablation experiments.
>
> 3. We added a runtime analysis of the **additional collaboration overhead** introduced by pFedCCG (excluding local training and model aggregation, Unit: s), see **Reviewer PgpC (Weakness (W3) (c))**. The results show that the projection step incurs **relatively low overhead** compared with most baselines, especially against adaptive graph-based methods. We will include this analysis in the revised version.
>
> 4. As shown in **Fig. 3 in the appendix**, this effect is not only observed in the final stage. After local training first pulls client models apart, the collapse behaviors described by the theory emerge very early in training, typically within the first few rounds (e.g., rounds 1–5). We will clarify this point in the final version.
>
> 5. We will revise the presentation to make the paper clearer and easier to follow by improving readability and moving key notations and assumptions closer to where they are used.
>
> 6. Thank you for pointing this out. This is a typo: for the TextCNN experiment at $\beta=0.2$, the correct result of pFedCCG is 79.76 (8.95), which is already correctly reported in Table 3. We will carefully proofread the full paper to avoid such errors in the final version. We have also released the code for reproducibility (Line 1228).
>
> Key Questions:
> 1. We have added a client-level comparison of whether performance improves under pFedCCG. On CIFAR-10, all 50/50 clients improve under each heterogeneity level ($\beta=0.2,0.5,1$). Therefore, no client experiences degradation, and the worst-case drop is 0. This indicates that the gain of pFedCCG is not concentrated on a small subset of clients, but is shared broadly across the population. This trend is also consistent with the violin plots in the appendix, which show an overall upward shift of the client-wise performance distribution.
> 2. **These hyperparameters were not tuned for each heterogeneity level.** We use fixed schedule hyperparameters within each task to demonstrate the robustness of pFedCCG. The settings are: CIFAR-10 $(\alpha_{\min}=0.1,\alpha_{\max}=0.3,t_s=25,\tau=7)$, Yahoo $(0.3,0.3,25,7)$, MovieLens $(0.4,0.2,25,10)$, and Aya $(0.4,0.2,15,7)$. We will clarify this explicitly in the final version, and provide the ablation details in xxx.
> 3. We added an explicit runtime analysis of the extra overhead introduced by the PGD-based projection step; due to the rebuttal space limit, we refer the detailed overhead results to **Reviewer PgpC (Weakness (W3) (c))**.
>
> We also evaluated how the **subproblem accuracy** affects final performance. In implementation, the subproblem accuracy is controlled by three solver parameters: $I$ (maximum PGD iterations), $J$ (number of alternating projection steps per PGD iteration), and $\varepsilon$ (stopping tolerance). The results below show that pFedCCG does **not** require a highly accurate subproblem solve: even with much looser settings, the final performance changes only marginally.
>
> |$J,\varepsilon,I$|    $\beta$0.2 |  $\beta$0.5 |  $\beta$1 |
> |---|--------------:|------------:|----------:|
> |5,1e-10,2000|         85.20 |       80.36 |     78.35 |
> |1,1e-4,2000|         85.20 |       80.32 |     78.35 |
> |1,1e-3,2000|         85.18 |       80.40 |     78.36 |
> |1,1e-2,200|         85.16 |       80.26 |     78.30 |

---

> > ### Author Rebuttal · Reviewer_bvHB · 2026-04-04
> >
> > Thank you to the reviewers for their efforts in preparing the response! I still find the reported quality improvements somewhat questionable. However, given additional experiments provided in the rebuttal, I raise my score to 4.

---

> > > ### Author Response · Authors · 2026-04-05
> > >
> > > We sincerely appreciate your time, thoughtful feedback, and continued engagement with our work.

---

### Official Review · Reviewer_2mhk · 2026-03-12

**Soundness:** 2
**Presentation:** 3
**Significance:** 3
**Originality:** 3
**Overall Recommendation:** 3
**Confidence:** 3

**Summary:**

This paper studies personalized federated learning from the perspective of collaboration geometry. The main claim is that collaborative PFL is more expressive than clustered PFL, but this advantage can be lost through two degeneration modes: collapse toward global consensus and similarity-driven self-clustering. To address this, the paper proposes pFedCCG, which controls the collaboration kernel using a fixed affinity template, projection onto a valid Markov geometry, and collaboration scheduling. Experiments on several tasks show improved personalized performance over multiple baselines.

**Compliance With Llm Reviewing Policy:**

Affirmed.

**Key Questions For Authors:**

1.	The paper characterizes both consensus and clustered solutions as degeneration modes because they reduce the expressivity of collaborative PFL. However, in some scenarios these regimes may actually match the underlying data structure. Could the authors clarify under what conditions these regimes should be considered undesirable?
2.	On Theorem 3.1 , the claim that “collaboration strictly dominates clustering” is central. Does this comparison only hold for hard-assignment clustered FL with one prototype per client group, or does it also cover stronger clustered baselines with soft assignment or additional client-specific adaptation? Please clarify exactly what class is ruled out by the theorem.
3.	The proposed method relies on a fixed affinity template and a projection-based kernel construction. How sensitive is pFedCCG to the quality of the affinity template and the choice of the prescribed stationary distribution qqq? In particular, if the template is noisy or qqq is mispecified, does the method still outperform adaptive baselines?

**Limitations:**

No. The paper does not explicitly discuss the main limitations of the proposed approach.

**Strengths And Weaknesses:**

Strength:
The paper addresses an important problem in personalized federated learning and proposes an interesting geometry-based perspective. The theoretical framing around two collapse modes is useful, and the experiments are fairly broad and show consistent gains across multiple tasks.

Weakness:
The concern lies in the interpretation and grounding of the theory. In particular, the paper treats both consensus and clustered solutions as degeneration modes, but in some settings these may simplify the true data structure. In addition, while the theory motivates controlling collaboration geometry, the connection to the specific pFedCCG design could be better justified. Finally, the empirical results are strong, but they mainly show accuracy improvements; more direct evidence for the claimed collapse dynamics would strengthen the paper.

---

> ### Author Rebuttal · Authors · 2026-03-31
>
> Thank you for your insightful comments and suggestions, which help strengthen the rigor of our work. We have made the following responses to the main concerns of the reviewers：
>
> Weakness:
> 1. We agree that consensus or clustered solutions are not always undesirable. They are only problematic when they arise as optimization-induced collapse rather than from the true data structure. If the clients are genuinely homogeneous, consensus is appropriate; if they naturally form a few well-separated groups, clustering is also appropriate. Our goal is only to prevent artificial collapse when richer collaboration is needed.
>
> 2. We agree that the connection from theory to the specific pFedCCG design should be stated more explicitly. Our claim is not that the theory uniquely derives pFedCCG, but that it identifies the key **design requirements**, and pFedCCG is a principled instantiation of them **(a simple and practical solution)**. More specifically:
>
> - **Similarity-feedback self-clustering** $\rightarrow$ **fixed feedback-free affinity template**
> - **Over-contractive mixing / consensus collapse** $\rightarrow$ **stage-wise control of mixing strength scheduling)**
> - **Need for valid and objective-aligned collaboration geometry** $\rightarrow$ **projection-based Markov kernel construction with prescribed stationary distribution $q$**
>
> 3. Following this suggestion, we added direct diagnostics beyond accuracy. We put the detailed results in the rebuttal for **Reviewer PgpC (Weakness 1)**. In addition, Fig. 3 in the appendix visualizes the claimed collapse dynamics process.
>
> Key Questions:
> 1. Thank you for this important comment. We agree that consensus or clustered solutions are not inherently undesirable: consensus is appropriate when client distributions are truly homogeneous, and clustering is appropriate when clients naturally form a few well-separated groups. They become undesirable only when they arise as optimization-induced collapse rather than from the true data structure, especially under fine-grained or continuous heterogeneity where richer collaboration is needed. pFedCCG is compatible with consensus or clustering when the data truly support them; for example, if clients naturally form clusters (e.g., pathological partitions), the similarity template can still induce a cluster-like collaboration pattern. Our goal is only to prevent collapse caused by optimization dynamics.
> 2. Thank you for this important question. **Theorem 3.1 is formally stated for the K-prototype clustered class**, i.e., methods whose client models are restricted by a clustering structure with only K effective anchors/prototypes. Hard assignment is the clearest special case, but the limitation is **not only about hard assignment**. It also applies to stronger clustered variants **as long as** their personalized models remain confined to **cluster-restricted representations**—for example, soft assignment or within-cluster adaptation that still reduces cross-cluster collaboration and limits each client to a cluster-specific prototype set. In this sense, the theorem applies more broadly to methods whose personalization remains fundamentally constrained by a K-cluster partition. By contrast, if a method allows essentially unrestricted client-specific mixing across clusters, then it is no longer clustered in the sense of Theorem 3.1, but is closer to the collaborative class itself. We will clarify this scope more explicitly in the final version.
> 3. We added robustness experiments by perturbing both the affinity template $\Psi$ and the prescribed stationary distribution $q$. For $\Psi$, we inject noise with different total variation (TV) levels; for $q$, we test permutation, noisy perturbation, and a uniform variant (w/o $p$).
>
> The results show that pFedCCG is **robust to moderate template noise and $q$ misspecification**. As the quality of $\Psi$ degrades, performance decreases gradually, but even under substantial perturbation (TV$=0.3$), pFedCCG still remains above the best adaptive baseline. Similarly, under all tested variants of $q$, pFedCCG consistently outperforms the adaptive baseline. These results indicate that the method does not rely on a highly accurate template or perfectly specified $q$ to remain effective.
>
> | Template $\Psi$ TV |0.2|     0.5 |     1 |
> |--------------------|---:|--------:|------:|
> | 0                  |85.20|   80.36 | 78.35 |
> | 0.05               |85.20|   80.35 | 78.34 |
> | 0.1                |85.16|   80.12 | 78.25 |
> | 0.15               |84.85|   79.90 | 77.88 |
> | 0.2                |84.53|   79.49 | 77.48 |
> | 0.25               |84.39|   79.29 | 77.00 |
> | 0.3                |84.21|   78.98 | 76.57 |
> | Best baseline               |83.97|   78.56 | 76.27 |
>
> |$q$ variant|   0.2 |0.5|1|
> |---|------:|---:|---:|
> |Permutation| 84.51 |80.01|78.14|
> |Noise| 85.23 |80.22|78.50|
> |w/o $p$| 85.01 |80.26|78.24|
> |Best baseline| 83.97 |78.56|76.27|

---

### Decision · Program_Chairs · 2026-04-30

**Decision:**

Accept (regular)

**Comment:**

The paper presents two degenerate extreme regimes that can occur in personalized federated learning: a collapse toward near-consensus and a collapse to block-separated collaboration. The authors show that the best achievable risk for collaborative learning always exceeds the risks for these 2 extreme regimes. Based on this observation, they propose pFedCCG and conduct intensive experiments to validate it.

The paper exposes a sound and convincing theoretical analysis.  The rebuttal phase mainly addressed questions about the experimental part. A comparison of the number of winning/losing clients was given to show the effective gains of the approach beyond a simple calculation of the average accuracy. Additional results were also provided about hyperparameters tuning. An empirical computational complexity analysis was presented for the more time consuming part that relies on  constrained convex optimization.

The main weakness of pFedCCG is that it takes advantage from a preprocessing step that computes a fixed affinity template from feature values. This shared knowledge cannot be done very efficiently and privately in decentralized/federated settings. However, the authors have shown that the results are robust against approximate evaluations of this matrix.